**EMBO** *reports*

# Functional BRI2-TREM2 interactions in microglia: implications for Alzheimer's and related dementias

Tao Yin [ID] ✉, Metin Yesiltepe & Luciano D'Adamio [ID] ✉

## Abstract

*ITM2B/BRI2* **mutations cause Alzheimer's Disease (AD)-related dementias. We observe heightened** *ITM2B/BRI2* **expression in microglia, a pivotal cell type in AD due to risk-increasing variants in the microglial gene** *TREM2*. **Single-cell RNA-sequencing demonstrates a Trem2/Bri2-dependent microglia cluster, underscoring their functional interaction. α-secretase cleaves TREM2 into TREM2-CTF and sTREM2. As BRI2 hinders α-secretase cleavage of the AD-related Aβ-Precursor-Protein, we probed whether BRI2 influences TREM2 processing. Our findings indicate a BRI2-TREM2 interaction that inhibits TREM2 processing in heterologous cells. Recombinant BRI2 and TREM2 proteins demonstrate a direct, cell-free BRI2-TREM2 ectodomain interaction. Constitutive and microglial-specific** *Itm2b-Knock-out* **mice, and** *Itm2b-Knock-out* **primary microglia provide evidence that Bri2 reduces Trem2 processing, boosts Trem2 mRNA expression, and influences Trem2 protein levels through α-secretase-independent pathways, revealing a multifaceted BRI2-TREM2 functional interaction. Moreover, a mutant** *Itm2b* **dementia mouse model exhibits elevated Trem2-CTF and sTrem2, mirroring sTREM2 increases in AD patients. Lastly, Bri2 deletion reduces phagocytosis similarly to a pathogenic TREM2 variant that enhances processing. Given BRI2's role in regulating Aβ-Precursor-Protein and TREM2 functions, it holds promise as a therapeutic target for AD and related dementias.**

**Keywords** *ITM2B*; microglia; Trem2; Alzheimer Disease; Phagocytosis
**Subject Categories** Molecular Biology of Disease; Neuroscience

## Introduction

*ITM2B* mutations have been linked to four autosomal dominant neurodegenerative diseases, the Familial British Dementia (FBD) (Vidal et al, 1999), the Familial Danish Dementia (FDD) (Vidal et al, 2000) and the newly discovered Familial Chinese (Liu et al, 2021) and Familial Korean Dementias (Rhyu et al, 2023). *ITM2B* encodes for a type II membrane protein called BRI2. BRI2 is synthesized as a precursor protein that is cleaved at the C-terminus by proprotein convertase into mature BRI2 and a 23 amino acid-long (Bri23) soluble C-terminal fragment (Choi et al, 2004). All pathogenic *ITM2B* mutations lead to changes in the C-terminal region of BRI2, resulting in the production of longer C-terminal fragments, which are processed into amyloidogenic peptides.

FDD and FBD share similarities with Alzheimer's disease (AD) in terms of their histopathological features, such as, neuroinflammation, neurodegeneration, the presence of extracellular amyloid plaques and intraneuronal neurofibrillary tangles. However, the composition of the amyloid plaques in FDD and FBD is different from that of AD. In AD, the amyloid plaques are primarily composed of amyloid-β (Aβ) peptides, which derive from the proteolytic processing of Amyloid-β Precursor protein (APP), whereas in FDD and FBD, the plaques are composed of the cleavage products of the mutant BRI2 proteins, the ADan peptide, or the ABri peptide, respectively (Garringer et al, 2010). Of note, in patients with FDD, Aβ deposition was observed either in combination with ADan or alone (Vidal et al, 2000). These differences in the composition of the plaques have led to the classification of FDD and FBD as distinct Alzheimer's disease-related dementias (ADRD) and to the conclusion that *ITM2B* mutations cause accumulation of amyloidogenic peptides aggregates, which lead to neuronal damage and dementia.

Studies on *Itm2b-Knock-out* (*Itm2b-KO*) and conditional *Itm2b-KO* mice, have shown that BRI2 has a cell autonomous physiological function in synaptic transmission and plasticity in glutamatergic neurons at both presynaptic and postsynaptic termini (Yao et al, 2019). FDD and FBD knock-in rodents show synaptic plasticity deficits like those observed in *Itm2b-KO* mice (Yin et al, 2021a). In addition, in FDD and FBD knock-in (KI) animal models, the mutant forms of Bri2 protein have been found to be unstable and rapidly degraded (Tamayev et al, 2010a, b; Yin et al, 2021b). These findings suggest that the pathogenesis of FDD and FBD may be more complex than originally thought and that both the accumulation of amyloidogenic peptides and a loss of BRI2 protein function may contribute to the development of these diseases.

BRI2 also has a dual anti-amyloidogenic function, reducing both Aβ production and Aβ aggregation. It has been found that BRI2 binds to APP in cis, thereby reducing APP cleavage and Aβ production (Matsuda et al, 2005, 2008, 2011a; Tamayev et al, 2011). In addition, the extracellular domain of BRI2 includes a BRICHOS domain that inhibits or delays Aβ aggregation (Chen et al, 2020;

Department of Pharmacology, Physiology & Neuroscience New Jersey Medical School, Brain Health Institute, Jacqueline Krieger Klein Center in Alzheimer's Disease and Neurodegeneration Research, Rutgers, The State University of New Jersey, 205 South Orange Ave, Newark, NJ 07103, USA. ✉E-mail: ty183@rutgers.edu; luciano.dadamio@rutgers.edu

Tambaro et al, 2017). These activities of BRI2 on APP processing and Aβ aggregation are supported by evidence that APP and APP processing play a role in long-term synaptic plasticity and memory deficits in FDD and FBD knock-in mice (Matsuda et al, 2011b; Tamayev and D'Adamio, 2012a, b; Tamayev et al, 2012b, 2011).

While BRI2 function has been primarily studied in neuronal cells, BRI2 may also have biological roles also in other Central nervous system (CNS) cell types. Analysis of mouse and human nervous system single cell RNAseq (scRNAseq) (Zeisel et al, 2018) and single nuclei RNAseq (Li et al, 2018) (snRNAseq) data showed that *ITM2B* expression in the CNS is highest in microglia (Fig. 1). This finding is significant as increasing evidence link neuroinflammation to AD (Akiyama et al, 2000; Tarkowski et al, 2003). For instance, variants of the TREM2 gene, which is exclusively expressed in microglia in the CNS (Schmid et al, 2002), have been shown to increase the risk of developing sporadic, late-onset AD (Guerreiro et al, 2013). TREM2 also undergoes regulated intramembrane proteolysis (Wunderlich et al, 2013), similar to APP, in which α-secretase cleaves TREM2 (Thornton et al, 2017) resulting in the release of soluble TREM2 ectodomain (sTREM2) and the membrane-tethered C-terminal fragment (TREM2-CTF). Levels of sTREM2 are increased in the Cerebrospinal fluid and CNS soluble fraction of AD patients (Heslegrave et al, 2016; Piccio et al, 2016; Suárez-Calvet et al, 2019), suggesting a potential role of TREM2 processing by α-secretase in AD pathology. TREM2-CTF is

subsequently cleaved in the transmembrane region by γ-secretase (Wunderlich et al, 2013).

Based on the above evidence, in the present study, we investigated the role of BRI2 in microglia, with a focus on potential BRI2-TREM2 physiological interactions mirroring those observed between BRI2 and the other AD-related secretases' substrate, APP. Understanding the functions of BRI2 in microglia and its interactions with other AD-related proteins may provide further insight into the complex pathogenic mechanisms underlying AD and related dementias.

# Results

## In the CNS, microglia express the highest levels of *Itm2b* mRNA

Unbiased clustering with high resolution (1.0) of mouse scRNAseq (Zeisel et al, 2018) and human snRNAseq (Li et al, 2018) data sets revealed 40 clusters in the mouse hippocampus and 31 clusters in the human dorso-lateral prefrontal cortex (DFC). Similar clusters were then manually grouped together when visualizing with uniform manifold approximation and projection (UMAP) (McInnes et al, 2018) to simplify the cell-types annotation (Fig. 1A, left panel, and Fig. 1B, left panel). Specifically, we combined

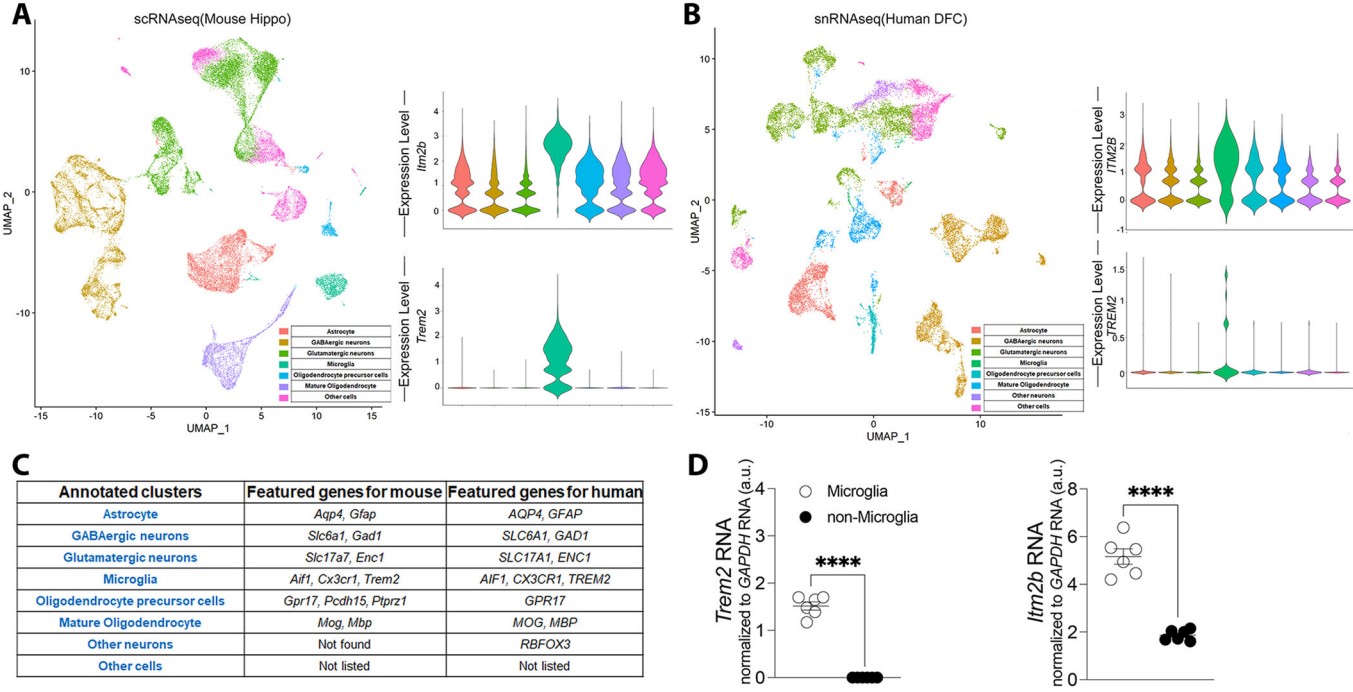

**Figure 1. Microglia express the highest levels of *ITM2B* mRNA in the CNS.**

(A) UMAP visualization of mouse hippocampal cell clusters classified by cell type based on DEG identified by Seurat v4. Violin plots represent the log-normalized expression of *Itm2b* and *Trem2* across cell populations in mouse hippocampal cell clusters. (B) Human DFC cell clusters, classified by cell type as in (A). Violin plots represent the log-normalized expression of *ITM2B* and *TREM2* across cell populations in human DFC cell clusters. (C) List of cell-type- specific marker genes used to annotate major brain populations. (D) *Itm2b* and *Trem2* mRNA expression in mouse microglia and non-microglia cells analyzed by quantitative RT-PCR. Data information: The data sets analyzed are publicly available and are described in the two following papers: mouse scRNAseq data set (Zeisel et al, 2018), hippocampus $n = 5$ females, $n = 5$ males; human snRNAseq data set (Li et al, 2018), $n = 10$ (sex not specified). More information about these datasets can be found in the cited paper. Statistical comparisons between the groups shown in (D) was conducted using two-tailed unpaired $t$ test ****$P < 0.0001$. The data are derived from are from 15-month-old *w/w* control animal, male $n = 3$, females $n = 3$; the letter "n" indicates biological replicates. All data are expressed as means $+/-$ SEM. Source data are available online for this figure.

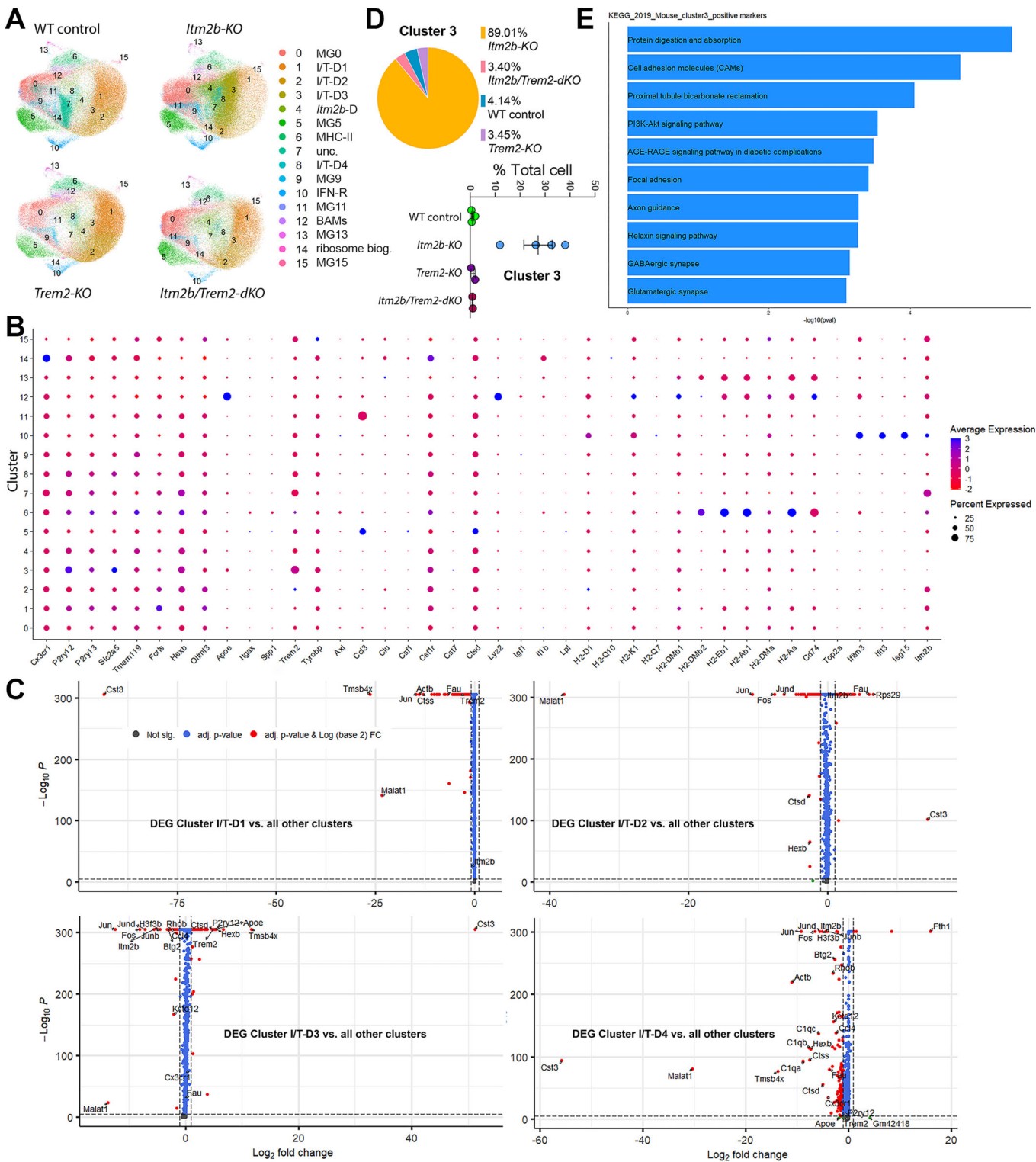

clusters corresponding to (1) Astrocytes; (2) GABAergic neurons; (3) glutamatergic neurons, (4) microglia, (5) oligodendrocyte precursor cells, (6) mature oligodendrocytes and (7) other nonspecific neurons or (8) other cells. The major populations were annotated based on differential expression of known cell-type-specific marker genes (listed in Fig. 1C). *Itm2b* and *Trem2* mean

mRNA expression was upregulated in the cluster identified as microglia compared to the rest of the cells in both mouse and human datasets (Fig. 1A, right panel, and Fig. 1B, right panel).

To validate these findings, CD11b+ cells were isolated employing the microglia isolation protocol (Tambini and D'Adamio, 2020) using the Adult Brain Dissociation Kit and the CD11b magnetic

◄ **Figure 2. *Itm2b* modulates microglial transcriptome and clustering in a *Trem2*-dependent manner.**

(A) UMAPs of microglia grouped by genotype. (B) Average scaled expression levels of selected signature genes per cluster and cluster's annotation based on expression of signature genes. (C) Volcano plots showing differentially expressed genes in clusters I/T-D1, 2, 3 and 4. (D) Proportional contribution of each genotype and proportional contribution of individual samples of each genotype to cluster 3. (E) KEGG pathway enrichment analysis of pathways upregulated in cluster 3. Data information: The data presented in this analysis are the result of two experiments, namely Data 1 and Data 2. To combine specific sample datasets from both Data 1 and Data 2, we employed the integration feature within the Seurat package. By utilizing the first 20 principal components, we integrated these datasets into a single entity referred to as "Object1," which encapsulates information from a total of 297,215 cells. Volcano plots in (C) were obtained using Fast Wilcoxon rank sum test and auROC. These cells derive from: *Trem2-KO*, n = 1 male and n = 1 female; *Itm2b-KO*, n = 2 males and n = 2 females; WT controls, n = 1 male and n = 2 females; *Itm2b/Trem2-dKO*, n = 1 male and n = 1 female. The scRNAseq data are deposited at https://www.ncbi.nlm.nih.gov/geo/info/seq.html, GSE233601 to allow public access once the data are published. All data are expressed as means +/− SEM.

microbeads from Miltenyi. Prior to brain harvesting, we removed peripheral myeloid cells and blood from brain tissue via intracardiac catheterization and perfusion. Quantitative RT-PCR analysis showed that the microglia-specific marker *Trem2* mRNA was expressed in the purified microglia but not in the unbound flow-through cells (referred to as non-microglia), indicating the purity and efficiency of the microglia isolation, and that *Itm2b* mRNA levels are significantly higher in microglia than non-microglia (Fig. 1D). Taken together, these findings indicate that *Itm2b* expression in the CNS is predominantly in microglia.

## *Itm2b* modulates microglial transcriptome in a *Trem2*-dependent manner

Next, we examined the impact of *Itm2b*, *Trem2* and combined *Itm2b-Trem2* deficiency on microglial transcriptome by scRNAseq. Single/live CD11b⁺ cells were isolated from WT control, *Itm2b-KO*, *Trem2-KO* and *Itm2b/Trem2-dKO* (double KO) brains, and single-cell transcriptomes were generated using the 10x Genomics platform in two independent experiments. After quality control, cells were plotted on UMAP dimensions for visualization (Fig. EV1A, left panels). To specifically select microglia for further analysis, we performed cell type annotation using a single cell dataset published by Van Hove (Van Hove et al, 2019) as a reference. Cells predicted to be of the type "microglia" with >95% confidence were retained for further analysis (Fig. EV1B). As the samples were sequenced in two different experiments (Data 1 and Data 2), the scRNAseg dataset integration functionality of the Seurat package was used to perform the joint analysis (Stuart et al, 2019). Select sample datasets indicated above from Data 1 and Data 2 were integrated using the first 20 principal components into Object1 containing information on 297,215 cells (Fig. EV1C). Unsupervised clustering of microglia revealed a total of 16 distinct microglia clusters across all mice (Figs. 2A, EV1C, EV2A and EV2B). Based on expression of specific marker genes (Chen and Colonna, 2021), we identified cluster 6 as MHC-II microglia (upregulation of genes such as *H2-Aa, H2-Ab1, H2-DMb1, H2-DMb2, H2-DMa* and *Cd74*, Figs. 2B and EV2E). Cluster 10 was identified as IFN-R microglia based on the upregulation of genes such as *Ifitm3, Isg15* and *Ifit3* (Fig. 2B). Cluster 12 was identified as brain-associated macrophages (BAMs), distinguished by the upregulation of *Mrc1, Cd163*, and *Lyve1* (Fig. EV2C). Similarly, cluster 14 exhibited strong resemblances to a recently characterized ribosome biogenesis cluster (Sun et al, 2023), marked by the upregulation of ribosomal genes (Fig. EV2C).

We next examined the representation of these microglia clusters in each genotype (Fig. EV2D and Table 1). We focused on clusters 0 to

10, which account for the large fraction of microglia, and excluded cluster 7 because ~93% of the cells assigned to this cluster derived from one WT control animal (the male WT control, Fig. EV2B). Therefore, the observed expansion of cluster 7 is attributed to animal-specific factors rather than genotype-specific factors. Several of these 10 clusters exhibited distinct representations across different genotypes. Cluster 4 displayed overrepresentation in the *Itm2b-KO* samples (and, to a somewhat lesser extent, in *Itm2b/Trem2-dKO* samples) and has thus been designated as the *Itm2b*-dependent (*Itm2b*-D) cluster. Differences in representation can be observed in Table 1.

Clusters 1, 2, 3, and 8 exhibit alterations in *Itm2b-KO* and/or *Trem2-KO* samples, yet do not display significant changes in *Itm2b/Trem2-dKO* samples. These findings strongly imply an interaction between *Itm2b* and *Trem2* in the formation of these clusters. Consequently, we designate these clusters as *Itm2b/Trem2*-dependent clusters (I/T-D) 1, 2, 3, and 4, respectively. Volcano plots showing the DEG between clusters I/T-D1, I/T-D2, I/T-D3, and I/T-D4 and all other clusters are shown in Fig. 2C. Among these clusters, I/T-D3 displayed the most distinctive pattern, with an almost exclusive representation in *Itm2b-KO* mice (89% of the microglia in this cluster originated from *Itm2b-KO* mice, Figs. 2D and EV2D). Notably, microglia assigned to I/T-D3 demonstrated high prevalence in all four *Itm2b-KO* mice investigated, ranging from 12.1 to 38.3% of the total microglia population (Figs. 2D and EV2B), thus underscoring the reproducibility across different subjects (unlike cluster 7). KEGG pathway enrichment analysis indicated that several neuronal function-related pathways were upregulated in cluster 3 relative to all other clusters. These include pathways related to axon guidance, GABAergic and Glutamatergic synapses (Fig. 2E and Table 2). Overall, these data demonstrate that the observed effect is not related to animal-specific factors, but rather to genotype-specific factors.

The enzymatic dissociation protocol utilized for microglia isolation from the brains carries the potential of activating microglia. Consequently, the scRNAseq data might not offer a completely accurate representation of the actual microglial populations in the brains of the mutant mice under investigation. Nevertheless, our findings provide robust evidence supporting the existence of a functional interaction between *Itm2b* and *Trem2*. Notably, *Itm2b* may act epistatically to *Trem2* in the regulation of cluster I/T-D3. This regulation could arise from the inhibition of Trem2 function by Bri2. This inhibition appears to be relieved in the absence of Bri2, leading to the expansion of cluster 3. It is noteworthy that the deletion of *Trem2* or both *Trem2* and *Itm2b* does not result in an increase in clusters 3, which emphasizes the essential role of *Trem2* in this pathway and underscores the role of *Itm2b* as an inhibitory regulator.

**Table 1.  Clusters where representation is affected by the genotype are highlighted.**

|  | *Itm2b-KO* | *I/T-DKO* | **WT** | *Trem2-KO* |
|---|---|---|---|---|
| Cluster 0 | 23.6 | 33.2 | 23 | 20.1 |
| Cluster 1 | 7.4 | 28 | 26 | 38.6 |
| Cluster 2 | 11.6 | 24.4 | 22.5 | 41.5 |
| Cluster 3 | 89 | 3.4 | 4.1 | 3.4 |
| Cluster 4 | 40.6 | 28.6 | 12.7 | 18.1 |
| Cluster 5 | 23.3 | 30.1 | 29.3 | 17.2 |
| Cluster 6 | 16.8 | 29.3 | 22.2 | 31.7 |
| *Cluster 7* | *9.2* | *0.1* | *90.1* | *0.6* |
| Cluster 8 | 42.6 | 21.9 | 15.6 | 19.9 |
| Cluster 9 | 36.2 | 23 | 18.7 | 22.1 |
| Cluster 10 | 29.4 | 18.3 | 28.2 | 24.1 |

Changes in cluster representation are noted marking them in blue when reduced and black when increased.

## BRI2 and TREM2 interact in transiently transfected cells

Like APP, TREM2 is processed by α- and γ-secretases. Although the functional consequences of TREM2 processing are not well understood, TREM2 cleavage has been suggested to play a role in regulating the activity of microglia in the brain as well as in AD pathogenesis (Heslegrave et al, 2016; Lichtenthaler et al, 2022; Piccio et al, 2016; Suárez-Calvet et al, 2019). BRI2 interacts with APP via its membrane-proximal region, which contains the secretases' cleavage sites, and inhibits APP processing (Matsuda et al, 2008; Tamayev et al, 2012b). If APP and TREM2 share structural similarities in these regions, it is possible that BRI2 also interacts with TREM2 and inhibit its processing in a similar manner.

To test these hypotheses, we utilized HEK293, derived from human embryonic kidney cells, and N2a cells, a mouse neuroblastoma cell line. The utilization of heterologous cells, where a functional interaction does not naturally happen, presents both advantages and disadvantages. One drawback is that it may not fully represent functions that rely on specialized, cell-specific, multi-molecular, complex pathways. On the other hand, it could more effectively reveal direct outcomes of protein-protein interactions, like that between BRI2 and TREM2, by isolating them from other biological effects. Thus, we transfected HEK293 and N2a cells with constructs coding for BRI2 FLAG-tagged at the $NH_2$-terminal cytoplasmic tail (F-BRI2) and rat Trem2 (Trem2-Miα isoform, UniProtKB - A0A6G8MV71), and then immunoprecipitated the lysates with an anti-FLAG antibody to pull down BRI2. The immunoprecipitants were analyzed using a Trem2-specific antibody to detect any interaction between BRI2 and Trem2. The results showed that Trem2 was precipitated by the anti-FLAG antibody only when BRI2 and Trem2 were co-expressed (Fig. 3A). This suggests an interaction between BRI2 and Trem2.

Trem2 is highly glycosylated, resulting in heterogeneous sizes (Fig. 3A,C). Deglycosylation of Trem2 leads to a homogeneous protein of about 22 kDa, which is efficiently immunoprecipitated by the anti-FLAG antibody in cells co-expressing F-BRI2 and Trem2 (Fig. 3C, middle panel). The anti-Trem2 NT1 antibody (the NT1 epitope is depicted in Fig. 3B) used in the study does not recognize Trem2-CTF, the membrane-bound product of Trem2 processing by

α-secretase. However, the anti-Trem2 CT antibody (the epitope of CT is depicted in Fig. 3B), detected both Trem2 and Trem2-CTF in both total lysates and immunoprecipitants (Fig. 3C), indicating that F-BRI2 interacts with both Trem2 and Trem2-CTF.

Next, we performed a reverse immunoprecipitation by using antibodies against Trem2 to pull down BRI2. We found that both anti-Trem2 CT and anti-Trem2 NT1 antibodies were able to pull down F-BRI2 only when Trem2 was co-expressed with BRI2 (Fig. 3D). However, a different antibody (anti-Trem2 NT2, see epitope in Fig. 3B) that did not immunoprecipitated Trem2 was not able to pull down BRI2 (Fig. 3D), indicating that the interaction between BRI2 and Trem2 is specific. There is a notable trend suggesting that F-BRI2 may preferentially interact with highly glycosylated mature Trem2. In Fig. 3A,C,D, we observe a BRI2-NTF band, likely representing the membrane-bound products resulting from BRI2 processing by ADAM10 (Martin et al, 2008). While the precise cleavage site remains unidentified, based on the comparison with BRI2-deletion mutants in Fig. 3F, we can estimate that this fragment may encompass amino acids approximately 1–105 of BRI2. Notably, it is important to mention that in these experiments, the presence of F-BRI2-NTF is not evident in lysates from cells co-transfected with F-BRI2 and Trem2.

To define the domain(s) of BRI2 that bind(s) to Trem2, BRI2 fragments progressively deleted from the COOH-terminus (scheme in Fig. 3E) were co-transfected with Trem2 in HEK293 cells. Trem2 was expressed at similar levels in all transfections (Fig. 3F, upper panel). Deletion of the BRI2-BRICHOS domain (F-BRI2$_{1-131}$) reduced binding of Trem2. F-BRI2$_{1-117}$, F-BRI2$_{1-105}$, and F-BRI2$_{1-93}$ bound Trem2 while F-BRI2$_{1-80}$ did not (Fig. 3F), suggesting the presence of two Trem2-binding domains in BRI2; one probably contained in the BRI2-BRICHOS domain and the other between amino acids 81 and 93 of BRI2. This second domain partially overlaps with the APP-binding domain of BRI2 (Fig. 3E). The Trem2-CTF is also co-immunoprecipitated in the M2-IP (Fig. 3C). The nature of this interaction, whether it is direct and mediated by the second Trem2-binding domain of BRI2, requires further investigation. Notably, BRI2 has been shown to interact with both APP and the APP C-terminal metabolite CTF-β (Matsuda et al, 2005) and can inhibit APP-CTF-β processing by the γ-secretase. In future studies, it would be intriguing to

**Table 2. KEGG pathway enrichment analysis indicated that several neuronal function-related pathways were upregulated in cluster 3 relative to all other clusters.**

| Term | Overlap | P value | Adjusted P value | Odds ratio | Combined Score | log10pval |
|---|---|---|---|---|---|---|
| Protein digestion and absorption | 11/90 | 3.61504E−06 | 0.000777234 | 6.477175916 | 81.16165051 | 5.441886621 |
| Cell adhesion molecules (CAMs) | 14/170 | 1.93491E−05 | 0.002080029 | 4.188178008 | 45.45372721 | 4.713339039 |
| Proximal tubule bicarbonate reclamation | 5/22 | 8.73655E−05 | 0.006261197 | 13.53148789 | 126.4572973 | 4.058659837 |
| PI3K-Akt signaling pathway | 19/357 | 0.000288145 | 0.013977797 | 2.630386271 | 21.44303049 | 3.540388695 |
| AGE-RAGE signaling pathway in diabetic complications | 9/101 | 0.000329076 | 0.013977797 | 4.526024992 | 36.29519912 | 3.482703857 |
| Focal adhesion | 13/199 | 0.000390078 | 0.013977797 | 3.248910549 | 25.50123066 | 3.408848471 |
| Axon guidance | 12/180 | 0.000539795 | 0.014585831 | 3.31544771 | 24.94649214 | 3.267771 |
| Relaxin signaling pathway | 10/131 | 0.000542729 | 0.014585831 | 3.8270366 | 28.77510996 | 3.265417287 |
| GABAergic synapse | 8/90 | 0.00071199 | 0.017008656 | 4.505375101 | 32.65246415 | 3.14752594 |
| Glutamatergic synapse | 9/114 | 0.00080004 | 0.017200862 | 3.963013234 | 28.25964781 | 3.096888246 |

| Term | Genes |
|---|---|
| Protein digestion and absorption | COL1A1;COL18A1;COL1A2;COL4A2;COL4A1;ELN;COL6A1;SLC1A1;ATP1A2;ATP1B2;SLC38A2 |
| Cell adhesion molecules (CAMs) | SELPLG;CADM1;SDC2;NRXN1;CLDN11;CLDN10;MAG;NFASC;CDH2;PECAM1;CNTN2;NCAM2;CD34;MPZL1 |
| Proximal tubule bicarbonate reclamation | CAR2;ATP1A2;ATP1B2;SLC38A3;SLC4A4 |
| PI3K-Akt signaling pathway | CSF1R;NTRK2;LAMA2;ANGPT1;LAMA4;VEGFC;IL2RG;GNG11;EGFR;COL1A1;COL1A2;CCND2;COL4A2;YWHAQ;COL4A1;COL6A1;GNB4;FGFR3;FGFR2 |
| AGE-RAGE signaling pathway in diabetic complications | COL1A1;COL1A2;COL4A2;COL4A1;PLCE1;VEGFC;PLCB1;F3;AGT |
| Focal adhesion | LAMA2;ACTN1;LAMA4;VEGFC;ARHGAP5;EGFR;COL1A1;COL1A2;CCND2;COL4A2;COL4A1;COL6A1;PAK7 |
| Axon guidance | SEMA5A;EPHA4;SEMA6A;EFNB3;UNC5B;PARD3;PLXNA2;PAK7;PLXNB1;EPHB1;GNAI1;RGMA |
| Relaxin signaling pathway | COL1A1;COL1A2;COL4A2;COL4A1;GNB4;VEGFC;PLCB1;GNG11;EGFR;GNAI1 |
| GABAergic synapse | GNB4;SLC6A11;SLC6A1;SLC38A3;SLC38A2;GNG11;GNAI1;GABRG1 |
| Glutamatergic synapse | SLC1A1;GNB4;SLC1A3;SLC38A3;GRIN2C;PLCB1;SLC38A2;GNG11;GNAI1 |

determine if the Trem2-BRI2 interaction follows a similar pattern and whether BRI2 can also influence the cleavage of TREM2-CTF by γ-secretase.

The interactions investigated in the previous experiments involved human BRI2 and rat Trem2 proteins. To explore potential interactions between human BRI2 and TREM2 and to refine our understanding of the interacting domains, we conducted transfections in HEK293 cells using bicistronic constructs encoding various combinations of human BRI2 and human TREM2 (UniProtKB-Q9NZC2). In the first set of constructs, we co-expressed full-length TREM2 tagged with a 3xFLAG at the C-terminus alongside various BRI2 variants, including full-length BRI2, BRI2$_{1-131}$, BRI2$_{1-80}$, and BRI2δ$_{80-131}$, which lacks amino acids 80-131. The BRI2 constructs in the second open reading frames of the bicistronic plasmids were tagged with a Myc epitope at the N-terminus (Fig. 4A). The two coding regions were separated by an Internal Ribosomal Entry Site sequence. The results from three independent transfection series are depicted in Figs. 4B and EV3A. The expression levels of these constructs exhibited variation across experiments, making precise quantitative comparisons challenging. Nevertheless, the data unequivocally establish that, under these experimental conditions, TREM2 interacts with BRI2. Furthermore, an observable band corresponding to BRI2-NTF is evident in these immunoprecipitants (Fig. EV3A). This band is produced through ADAM10 processing of BRI2 (Martin et al, 2008). Since BRI2-NTF appears to

encompass approximately amino acids 1-105 of BRI2 (see above), the fact that BRI2-NTF co-precipitates with TREM2 aligns with our findings that both BRI2$_{1-105}$ and BRI2$_{1-93}$ interact with Trem2 (Fig. 3F). In addition, the evidence confirming the interaction between TREM2 and BRI2$_{1-131}$, in contrast to BRI2$_{1-80}$, provides additional support for the presence of a TREM2-binding domain spanning amino acids 80–131 of BRI2, as initially indicated in the experiments shown in Fig. 3F. Lastly, the observation that BRI2δ$_{80-131}$, lacking this TREM2-binding domain, still interacts with TREM2, solidifies the existence of a second TREM2-binding domain located in the extracellular C-terminus of BRI2, as initially postulated based on the experiment in Fig. 3F.

In the second set of constructs, we co-expressed full-length BRI2 tagged with a 3xFLAG at the N-terminus along with various variants of TREM2, including full-length TREM2, TREM2-δIg-like (lacking the Ig-like domain of TREM2 spanning amino acids 29-112), TREM2-CTF (comprising amino acids 158-230), TREM2-W198Ter (encoding the TREM2 W198Ter variant, a truncated TREM2 protein lacking the intracellular domain associated with FTD when homozygous (Giraldo et al, 2013), and Trem2-δ/α-site (lacking amino acids 152–163, including the α-secretase cleavage site). The TREM2 constructs in the second open reading frames of the bicistronic plasmids were tagged with a Myc epitope at the C-terminus (Fig. 4C). However, the Myc epitopes were not efficiently detected for TREM2 proteins in Western blots, and

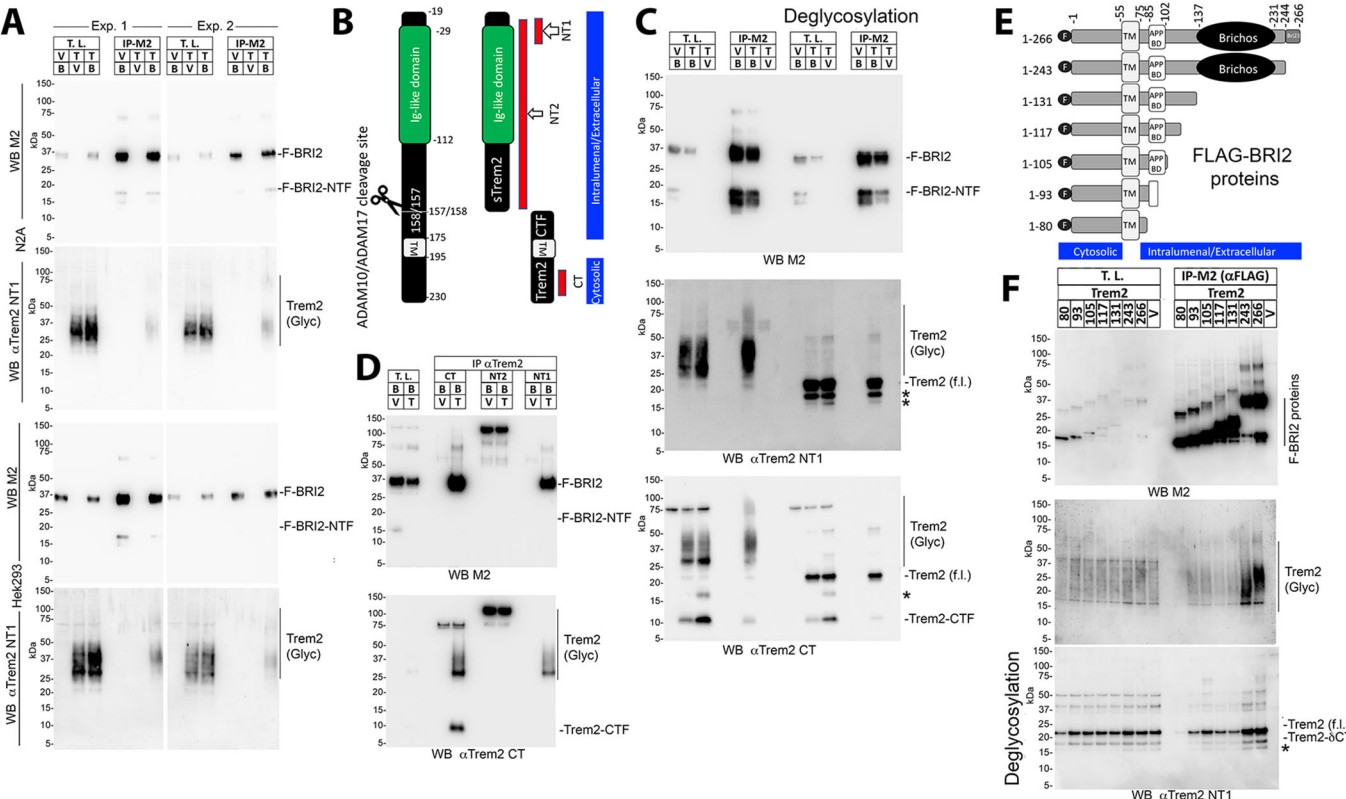

**Figure 3. Human BRI2 binds rodent Trem2 in transfected cells.**

(A) N2A or HEK293 cells were transfected with F-BRI2 (B) and Trem2 (T), either alone (V=empty pcDNA3.1vector) or in combination and analyzed by Western blot with anti-FLAG (M2) and anti-Trem2 (NT1) on total lysates (T.L.) and M2 immunoprecipitants (IP-M2). Immunoprecipitants bound to M2-Agarose beads were specifically eluted using the 3xFLAG peptide. For each cell line, two independent transfections were performed (Exp. 1 and Exp. 2). (B) Schematic representation of Trem2 and the two products of α-secretase cleavage, sTrem2, and Trem2-CTF. TM indicates the transmembrane region of Trem2. Red bars point to the antigenic regions used to produce the anti-Trem2 antibodies CT, NT1 and NT2. The cytosolic and intralumenal/extracellular regions of Trem2 are indicated. (C) Western blot analysis with anti-FLAG, anti-Trem2 NT1, and anti-Trem2 CT antibodies of T.L. and IP-M2 from HEK293 cells transfected with F-BRI2 and Trem2, either alone or in combination, with or without deglycosylation. *Indicates Trem2 species of unclear primary structure. Trem2 (f.l.) indicates full length Trem2. (D) Western blot analysis with anti-FLAG and anti-Trem2 CT antibodies of immunoprecipitants obtained with CT, NT1, and NT2 antibodies from HEK293 cells expressing either F-BRI2 alone or F-BRI2 plus Trem2. The nature of the bands migrating above 100 kDa in the NT2 IP samples is unknow. (E) Schematic representation of the F-BRI2 constructs used in (F). The Bri23 region, transmembrane region (TM), Brichos domain, APP-binding domain (APP BD), FLAG tag (F), cytosolic and intralumenal/extracellular regions are indicated. (F) WB analysis with anti-FLAG and anti-Trem2 antibodies of lysates and immunoprecipitants from HEK293 cells expressing F-BRI2 deletion mutants plus Trem2 or Trem2 alone (V). The * indicates Trem2 species of unclear primary structure. Data information: This figure encompasses the comprehensive dataset employed for these specific experiments. We have included the images of the complete membranes used for Western blot analyses, without any cropping of information above or below the targeted signals.

depending on the mutant, we used either the anti-human TREM2-CT antibody or the anti-human-TREM2-NT antibody for detection in Western blot. Like for the first series, expression levels of the constructs varied across experiments, which made accurate quantitative assessments challenging (Figs. 4D and EV3B). Yet, the data show that BRI2 is capable of immunoprecipitating TREM2. It is worth noting that we observed a distinct preference for the highly glycosylated forms of TREM2 in this interaction, emphasizing an affinity for fully mature TREM2 molecules. Furthermore, we detected TREM2-CTF within the immunoprecipitants (Figs. 4D and EV3B). These findings are consistent with the data presented in Fig. 3. Interestingly, the highly glycosylated forms of the TREM2-δ/α-site mutant also exhibited enrichment in the BRI2 immunoprecipitants (Figs. 4D and EV3B), resembling the pattern observed for TREM2, suggesting that the deletion of the 12 amino acids did not significantly alter the binding characteristics. However, TREM2-W198Ter, TREM2-δIg-like, and TREM2-CTF

transfectants produced more intricate results (Figs. 4D and EV3B). All three TREM2 mutants exhibited binding to BRI2, but the binding patterns differed. BRI2 displayed minimal binding to TREM2-CTF, while it exhibited binding with both immature and highly glycosylated mature forms of TREM2-W198Ter, and it bound primarily bound to immature forms of TREM2-δIg-like. It is important to note that these experiments with mutant proteins come with a caveat: these mutations may impact the trafficking of mutant TREM2 proteins, potentially influencing their interaction with BRI2 independently of the presence or absence of BRI2 binding domains. To fully elucidate the effects of these two phenomena on the interaction between BRI2 and TREM2, as well as the impact of pathogenic TREM2 mutations such as TREM2-W198Ter on this binding, further experiments are necessary. In summary, the analysis of TREM2-deletion mutants suggests a predominant interaction of mature TREM2 with BRI2. However, it does not conclusively identify two distinct BRI2 binding domains

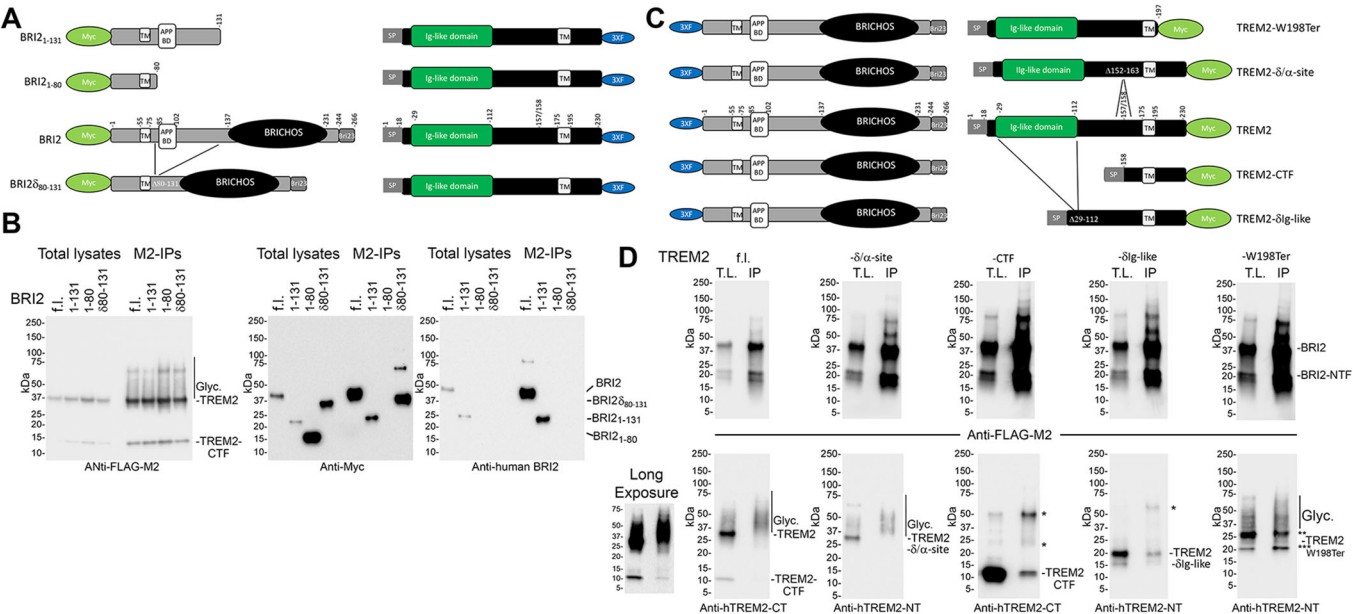

**Figure 4. Human BRI2 binds human TREM2 in transfected cells.**

(A) Schematic representation of the bicistronic expression plasmids used for HEK293 cell transfections in Panel (B): TREM2-3xFLAG + Myc-BRI2, 3xFLAG + Myc-BRI2$_{1-131}$, 3xFLAG + Myc-BRI2$_{1-80}$, and 3xFLAG + Myc-BRI2δ$_{80-131}$. The TREM2-3xFLAG is expressed by the 5′ cistron, while BRI2 proteins are expressed by the 3′ cistron. (B) Western blot analysis using anti-FLAG, anti-Myc, and anti-human BRI2 antibodies of Total Lysate and Immunoprecipitation (M2-IP) samples from transfected HEK293 cells. "Glyc." indicates glycosylated TREM2. The anti-human BRI2 antibody exhibits reactivity toward Myc-BRI2 and Myc-BRI2$_{1-131}$ suggesting recognition of an epitope located within the amino acids 80-131 region of BRI2. (C) Schematic representation of the bicistronic expression plasmids employed for HEK293 cell transfections in Panel (D): 3xFLAG-BRI2 + TREM2-Myc, 3xFLAG-BRI2 + TREM2-δIg-like-Myc, 3xFLAG-BRI2 + TREM2-CTF-Myc, 3xFLAG-BRI2 + TREM2-W198Ter-Myc, and 3xFLAG-BRI2 + TREM2-δ/α-site-Myc. The 3xFLAG-BRI2 is expressed by the 5′ cistron, while TREM2 proteins are expressed by the 3′ cistron. (D) Western blot analysis with anti-FLAG, anti-human TREM2-NT, and anti-human TREM2-CT antibodies of Total Lysate (T.L.) and Immunoprecipitation (IP) samples from transfected HEK293 cells. "Glyc." indicates glycosylated TREM2. * Indicates protein signals of unclear nature. ** and *** indicate TREM2W198Ter signals of unclear primary structure. A longer exposure (Long Exposure) of the Anti-hTREM2-CT Western blot for the 3xFLAG-BRI2 + TREM2-Myc transfection revealed traces of TREM2-CTF precipitating with BRI2. Data information: This figure represents one of three independent experiments conducted. Data from the other two experiments are presented in Fig. EV3. We have included the images of the complete membranes used for Western blot analyses, without any cropping of information above or below the targeted signals.

that may interact with the two TREM2-binding domains of BRI2 identified earlier.

To explore the endogenous interaction between Bri2 and Trem2, we conducted immunoprecipitation assays using the Trem2 CT antibody on primary macrophages. We chose macrophages over microglia due to their higher cell yield. Importantly, a band similar in size to Bri2 was precipitated with Trem2 CT from WT primary macrophages, but not *Itm2b-KO* macrophages, providing evidence for the physiological relevance of this interaction (Fig. EV3C). Furthermore, we observed a Bri2 band in the immunoprecipitation, consistent in size with Bri2-NTF. It is worth noting that a faint band of similar size was also observed in the immunoprecipitation from *Itm2b-KO* primary macrophages, introducing a caveat to this finding.

## Evidence of a direct interaction between BRI2 and TREM2 ectodomains

The experiments shown in Figs. 3 and 4 suggest the presence of two TREM2-binding domains within the extracellular domain of BRI2: one located within the BRICHOS domain and the other spanning amino acids 81 to 131 (or, potentially, 93) of BRI2. To confirm the existence of these two TREM2-binding domains and explore the possibility of a direct interaction between the extracellular domains

of human TREM2 and BRI2, we produced four recombinant proteins in CHO-S cells: sTREM2, TREM2-ECD, BRI2-BRICHOS (encompassing the most C-terminal TREM2-binding domain), and BRI2-ECD (containing both TREM2-binding domains) (Fig. 5A). We conducted experiments to assess whether recombinant BRI2-BRICHOS and BRI2-ECD directly interact with recombinant sTREM2 and TREM2-ECD. These four potential BRI2/TREM2 pairings were tested at a final concentration of 2 µM for each protein. The results demonstrate that both BRI2-BRICHOS and BRI2-ECD are capable of binding to both sTREM2 and TREM2-ECD (Fig. 5B).

Next, we tested whether sTREM2 and TREM2-ECD interact with the 3xFLAG-tag of BRI2-BRICHOS and BRI2-ECD. However, sTREM2 and TREM2-ECD were recovered in the eluates of 3xFLAG-tagged BRI2-BRICHOS and BRI2-ECD, but not in eluates from the 3xFLAG peptide alone (Fig. 5C), suggesting that the interaction is unlikely to be with the 3xFLAG portion of BRI2-BRICHOS and BRI2-ECD. Another improbable possibility is an interaction occurring at the junction between 3xFLAG and BRI2 proteins, given the differences in these junctions between BRI2-BRICHOS and BRI2-ECD.

Prior studies have shown diverse quaternary structures of recombinant BRI2-BRICHOS, driven by intermolecular disulfide bridges (Chen et al, 2017). Our findings are consistent with these

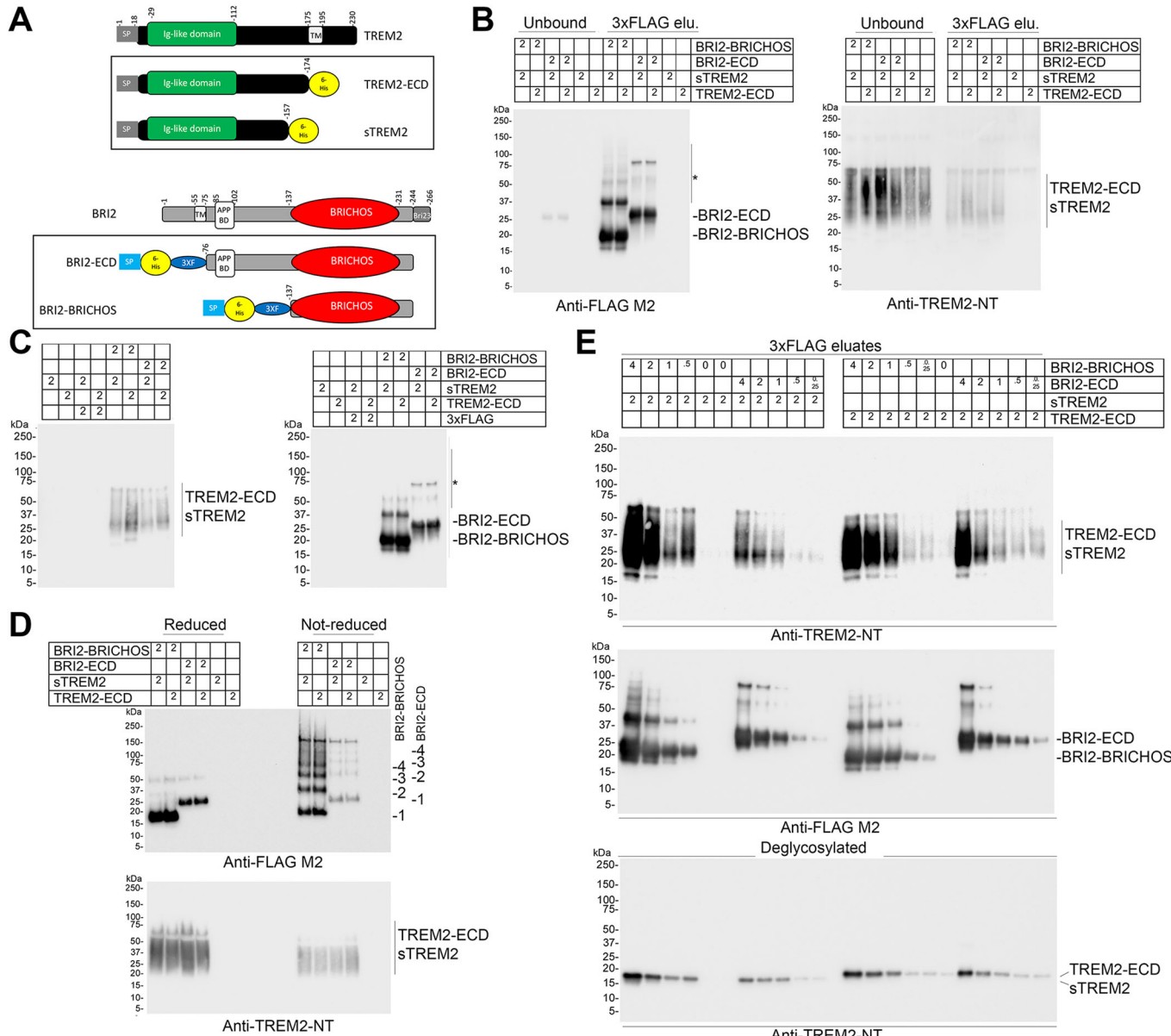

reports, demonstrating that both BRI2-BRICHOS and BRI2-ECD form dimers and oligomeric aggregates via intermolecular disulfide bridges (Fig. 5D). Moreover, we observe that sTREM2 and TREM2-ECD, co-eluted with BRI2-BRICHOS and BRI2-ECD, show no significant increase in molecular weight under non-reducing conditions compared to reducing conditions. This absence of a size difference implies that sTREM2 and TREM2-ECD do not form complexes with recombinant BRI2-BRICHOS and BRI2-ECD through disulfide bridges, which would result in an approximate 20 kDa increase in molecular weight on average (Fig. 5D). This finding underscores that these interactions are not artificially induced by covalent disulfide bonds between the recombinant proteins, further substantiating the physiological relevance of this interaction.

Interestingly, a single point mutation, R221E, within the BRI2 BRICHOS domain has been shown to favor the monomeric form

and exhibits greater efficacy than wild-type BRI2 BRICHOS in mitigating Aβ42-mediated neurotoxicity in mouse hippocampal slices (Chen et al, 2020; Manchanda et al, 2023). If monomers indeed possess the highest binding affinity for TREM2, it raises the possibility that our current recombinant proteins may underestimate the strength of interaction between the extracellular domains of BRI2 and TREM2, as well as the potential biological implications of the BRI2 extracellular domain, including its impact on TREM2 processing and functions.

To further assess these cell-free interactions, we conducted experiments with varying amounts of BRI2 recombinant proteins while maintaining the TREM2 recombinant protein concentration at 2 μM. In each of these iterations, the amount of recovered TREM2 recombinant proteins in the eluates was proportional to the quantity of BRI2 recombinant proteins (Fig. 5E). Recombinant sTREM2 and TREM2-ECD are glycosylated, as demonstrated by

**Figure 5. Evidence of a direct interaction between the ectodomain of BRI2 and TREM2.**

(A) Schematic representation of TREM2-ECD, sTREM2, BRI2-ECD and BRI2-BRICHOS recombinant proteins. TREM2-ECD encompasses the entire extracellular domain of TREM2, and BRI2-ECD encompasses the entire extracellular domain of BRI2, including the second putative TREM2-interacting domain. BRI2-BRICHOS and BRI2-ECD were fused with a 3xFLAG tag at their N-terminus, enabling immunoprecipitation using anti-FLAG M2-Agarose beads for the purification of protein complexes in a cell-free system via elution with a 3xFLAG peptide. The diagram highlights the signal peptides (SP), 7-Histidine tag (7-His, employed for protein purification), the 3XFLAG tag (3xF, utilized for complex purification), Ig-like domain (of TREM2), and BRICHOS domain (of BRI2). (B) BRI2-BRICHOS + sTREM2, BRI2-BRICHOS + TREM2-ECD, BRI2-ECD + sTREM2, BRI2-ECD + TREM2-ECD, sTREM2 alone, and TREM2-ECD alone were incubated overnight at 4 degrees Celsius with M2-Agarose beads at a concentration of 2 μM for each protein. Following extensive washing, complexes bound to M2-Agarose beads were specifically eluted using the 3xFLAG peptide. Unbound proteins and eluates (3xFLAG elu.) were analyzed by Western blot using either the anti-FLAG antibody M2 or an anti-human TREM2 N-terminal antibody (TREM2-NT). sTREM2 and TREM2-ECD were not recovered in the eluates when BRI2 recombinant proteins were absent. The * indicates residual BRI-BRICHOS and BRI2-ECD dimers and oligomers. (C) BRI2-BRICHOS + sTREM2, BRI2-BRICHOS + TREM2-ECD, BRI2-ECD + sTREM2, BRI2-ECD + TREM2-ECD, 3xFLAG + sTREM2, 3xFLAG + TREM2-ECD, sTREM2 alone, and TREM2-ECD alone were incubated as in B. Proteins eluted with the 3xFLAG peptide were analyzed by Western blot using either M2 or TREM2-NT. sTREM2 and TREM2-ECD were not recovered in the eluates when BRI2 recombinant proteins were absent. The * indicates residual BRI2-BRICHOS and BRI2-ECD dimers and oligomers. (D) Western blot analysis using M2 and TREM2-NT antibodies of a new experiment mirroring the setup in (B). Eluates were separated under reducing and non-reducing conditions. BRI2-BRICHOS and BRI2-ECD monomers, dimers, trimers, and tetramers are indicated by the numbers 1, 2, 3, and 4, respectively. Higher multimolecular complexes are present but not labeled. The sTREM2 and TREM2-ECD bound to BRI2-BRICHOS and BRI2-ECD analyzed under non-reducing conditions show no significant increase in molecular weight compared to those analyzed under reducing conditions. (E) Decreasing concentrations (4, 2, 1, 0.5, 0.25, and 0 μM) of BRI2-BRICHOS and BRI2-ECD were incubated with 2 μM of either sTREM2 or TREM2-ECD and analyzed as described in (C). The bottom panel displays a Western blot of deglycosylated eluates using the TREM2-NT antibody. Data information: This figure encompasses the comprehensive dataset employed for these specific experiments. We have included the images of the complete membranes used for Western blot analyses, without any cropping of information above or below the targeted signals.

their heterogeneous sizes (Fig. 5B–D) and the evidence that deglycosylation results in homogeneous proteins of ~17 and ~18 kDa, respectively (Fig. 5E). The significance of this glycosylation in the interaction between BRI2 and TREM2 remains to be determined. It's important to emphasize that these experiments did not yield conclusive evidence regarding the presence of two distinct TREM2 binding domains. This is because no appreciable differences were observed in the interactions of BRI2-BRICHOS and BRI2-ECD with sTREM2 and TREM2-ECD, making it challenging to definitively confirm the existence of two separate binding domains.

In summary, while an interaction between the 7 His tags (or at the junction of the 7 His and 3xFLAG tags) of the recombinant proteins cannot be formally excluded, and additional studies are needed to precisely identify the interaction regions within the BRI2 and TREM2 ectodomains, and to ascertain whether TREM2 interacts with monomeric or oligomeric forms of BRI2-BRICHOS and BRI2-ECD, the current findings strongly substantiate the presence of an interaction between BRI2 and TREM2, and suggest a direct interaction occurring within their extracellular domains.

## BRI2 reduces α-secretase-mediated processing of TREM2 in transiently transfected cells

Binding of BRI2 to APP has been shown to reduce secretases-mediated processing of APP (Matsuda et al, 2008). To test if BRI2 has a similar effect on Trem2 processing, HEK293 cells were co-transfected with Trem2 and either empty vector or F-BRI2. Co-transfection of Trem2 with F-BRI2 in HEK293 cells led to an increase in Trem2 levels and a decrease in Trem2-CTF in the cell lysates and sTrem2 in the tissue culture media (Fig. 6A,B). The observation that overexpression of BRI2 leads to elevated levels of the α-secretase substrates Trem2, and a simultaneous decrease in the α-secretase products Trem2-CTF and sTrem2, strongly supports the idea that BRI2 functions as an inhibitor of Trem2 processing by α-secretase.

To determine whether BRI2 binding is required for the effects on Trem2 processing, HEK293 cells were co-transfected with

Trem2 and either empty vector, F-BRI2, or F-BRI2$_{1-80}$ that does not bind Trem2 (Fig. 3F). WB analysis shows that F-BRI2 significantly increased Trem2 levels, reducing Trem2-CTF and sTrem2 amounts (Fig. 6C,D). In contrast, F-BRI2$_{1-80}$ did not alter levels of Trem2, Trem2-CTF and sTrem2 (Fig. 6C,D). This implies that BRI2's inhibition of α-secretase processing of Trem2 requires its binding to Trem2. Figure 6E provides additional support for the annotation of the sTrem2 band indicated in Fig. 6A,C. Specifically, it demonstrates that this band is detected by the antibody against the ectodomain of Trem2 (NT1) but not by the antibody against the intracellular domain of Trem2 (CT). In addition, the absence of Trem2 reactivity in the media with the CT antibody suggests that Trem2 is not appreciably secreted, at least under the tested conditions.

Since BRI2 is an α-secretase substrate, BRI2 overexpression might influence Trem2 processing through substrate competition. In addition, overexpression of BRI2 could impact the overall activity of α-secretases. However, prior evidence suggests that this is unlikely. BRI2 binds to APP but not to APP-Like Protein 1 (APLP1) and APP-Like Protein 2 (APLP2). Both APLP1 and APLP2 are also substrates for α-secretase, like APP (Scheinfeld et al, 2002a). Overexpression of F-BRI2 had no discernible impact on the processing of APLP1 and APLP2 (Matsuda et al, 2008). Nevertheless, when chimeric molecules of APLP1 and APLP2 were engineered, with the BRI2-binding domain of APP replacing the corresponding domains in APLP1 and APLP2, these chimeric proteins bound to BRI2 and were susceptible to processing inhibition by BRI2 (Matsuda et al, 2008). These results suggest that the inhibition of α-secretase processing by BRI2 is primarily driven by BRI2-substrate interaction rather than by substrate competition and/or increased activity of α-secretase.

## *Itm2b* deletion results in elevated CNS levels of Trem2-CTF and sTrem2 in mice

To confirm the in vitro findings, we measured the levels of Trem2, sTrem2, and Trem2-CTF in the CNS of approximately 245-day-old *Itm2b*-KO and WT mice. Due to the heterogeneity of Trem2 and sTrem2 caused by glycosylation, and their limited expression in

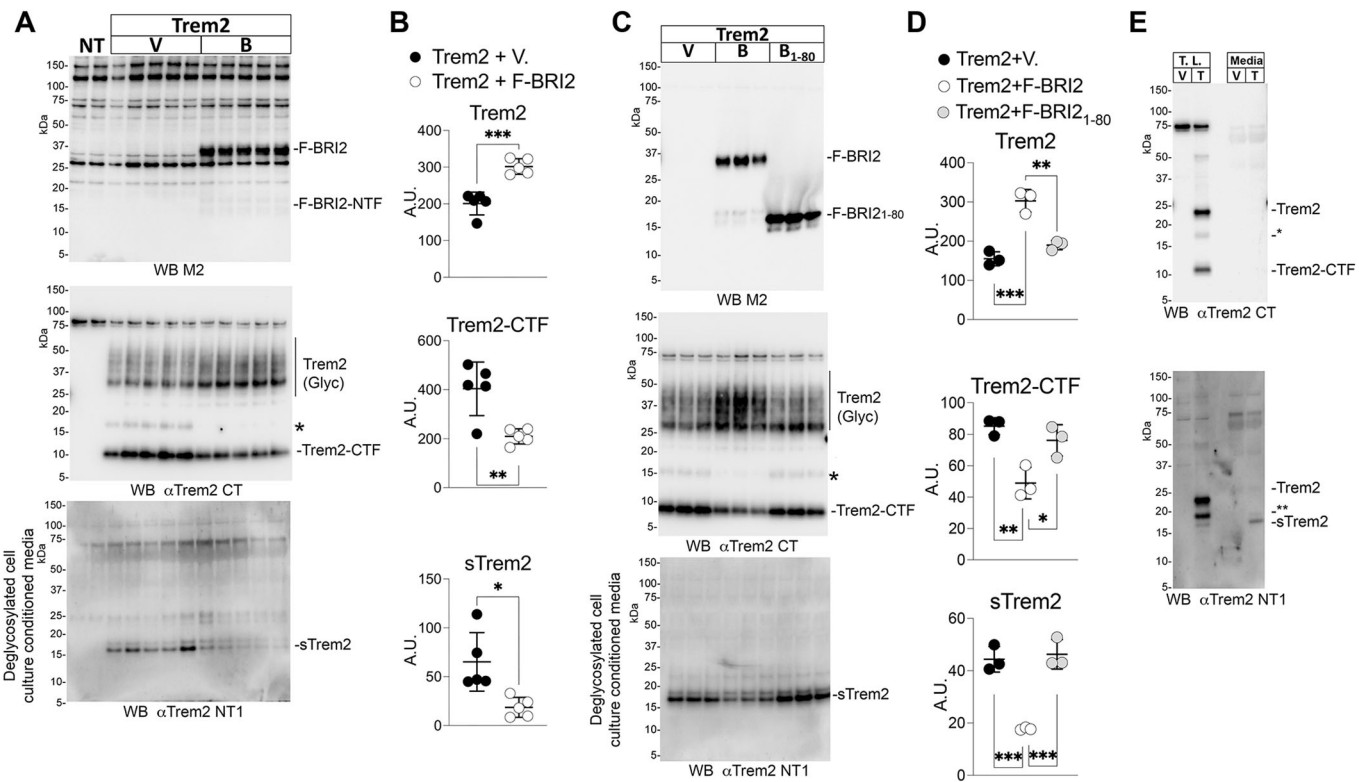

**Figure 6. BRI2 attenuates α-secretase cleavage of Trem2 in transfected cells.**

(A) HEK293 cells were transfected with Trem2 and either empty vector (V) or F-BRI2 (B). NT are non-transfected cells. Western blot of cell lysates with either the anti-FLAG antibody M2 or the anti-Trem2 antibody CT. Western blot of deglycosylated culture supernatants with the anti-Trem2 antibody NT1. *Indicates Trem2 species of unclear primary structure. (B) Quantification of Trem2, Trem2-CTF and sTrem2 levels detected by Western blot in (A). (C) HEK293 cells were transfected with Trem2 and either empty vector (V), F-BRI2 or deletion mutant F-BRI2$_{1-80}$. Western blot of cell lysates with either the anti-FLAG antibody M2 or the anti-Trem2 antibody CT. Western blot of deglycosylated culture supernatants with the anti-Trem2 antibody NT1 (lower panel). The * indicates Trem2 species of unclear primary structure. (D) Quantification of Trem2, Trem2-CTF and sTrem2 levels detected by Western blot in (C). (E) HEK293 cells were transfected with either Trem2 (T) or empty vector (V). Following transfection, lysates and media underwent deglycosylation and were subsequently analyzed by Western blot using anti-Trem2 antibodies CT and NT1. In the CT Western blot, the asterisk (*) indicates a Trem2-derived polypeptide that retains the CT epitope and is likely to lack part of the N-terminal Trem2 sequence, causing a reduction in size. In the NT1 Western blot, the double asterisk (**) highlights a Trem2-derived polypeptide that retains the NT1 epitope and is likely to lack part of the C-terminal sequence. The presence of this band primarily in cell lysates suggests potential retention of the transmembrane region and/or localization within intracellular compartments. Its low-level detection in the media further suggests intracellular origin. The band marked as sTrem2 is marked as sTrem2 because: (1) is of the expected size for deglycosilated sTrem2; (2) it is notably enriched in the media, consistent with the preferential localization of sTrem2 in extracellular fluids. Data information: This figure encompasses the comprehensive dataset employed for these specific experiments. We have included the images of the complete membranes used for Western blot analyses, without any cropping of information above or below the targeted signals. Statistical comparisons among the groups were conducted using two-tailed unpaired *t* test (B) and one-way ANOVA followed by post-hoc Tukey's multiple comparisons test when ANOVA showed significant differences (C, D). *$P < 0.05$, **$P < 0.01$, ***$P < 0.001$. The data presented are derived from are from: Trem2+Vector transfectant $n = 5$, Trem2+F-BRI2 transfectant $n = 5$ (A, B); Trem2+Vector transfectant $n = 3$, Trem2+F-BRI2 transfectant $n = 3$, Trem2+F-BRI2$_{1-80}$ $n = 3$ (C, D); the letter "n" indicates biological replicates. All data are expressed as means $+/-$ SEM. Source data are available online for this figure.

microglia (which represent only ~10% of CNS cells), we employed two ELISAs: ELISA 1, which detects only Trem2, and ELISA 2, which detects both Trem2 and sTrem2, to accurately measure their levels in vivo (Fig. 7A). Brain homogenates were separated into two fractions using centrifugation at $100,000 \times g$. The pellet fraction (P100) is enriched in cells-derived material, while the soluble fraction (S100) is enriched in soluble extracellular material. Using ELISA 1 (Fig. 7B, upper panels) we found that Trem2 is not detectable in *Trem2-KO* brains, demonstrating the specificity of the assay. Furthermore, Trem2 was detected only in the P100 fraction, indicating that it is cell-bound. We did not observe significant differences in the levels of Trem2 between *Itm2b-KO* and control WT animals. ELISA 2 (Fig. 7B, lower panels) revealed that Trem2

proteins are not detectable in *Trem2-KO* brains, confirming the specificity of the assay. We found that sTrem2 was significantly enriched in the S100 fraction of *Itm2b-KO* animals compared to control WT animals.

Although Trem2-CTF cannot be detected by ELISA 1 and 2 (Fig. 7A), we were able to detect it in the P100 fraction by WB analysis (Fig. 7C) as it is not glycosylated. Quantification of Trem2-CTF levels in the P100 brain fractions showed that both female and male *Itm2b-KO* brains contained significantly higher steady-state levels of Trem2-CTF compared to control WT animals (Fig. 7D). It is worth noting that in this experiment, the significance of the difference in Trem2-CTF levels is higher in females compared to males when tested against control animals of the same sex. While

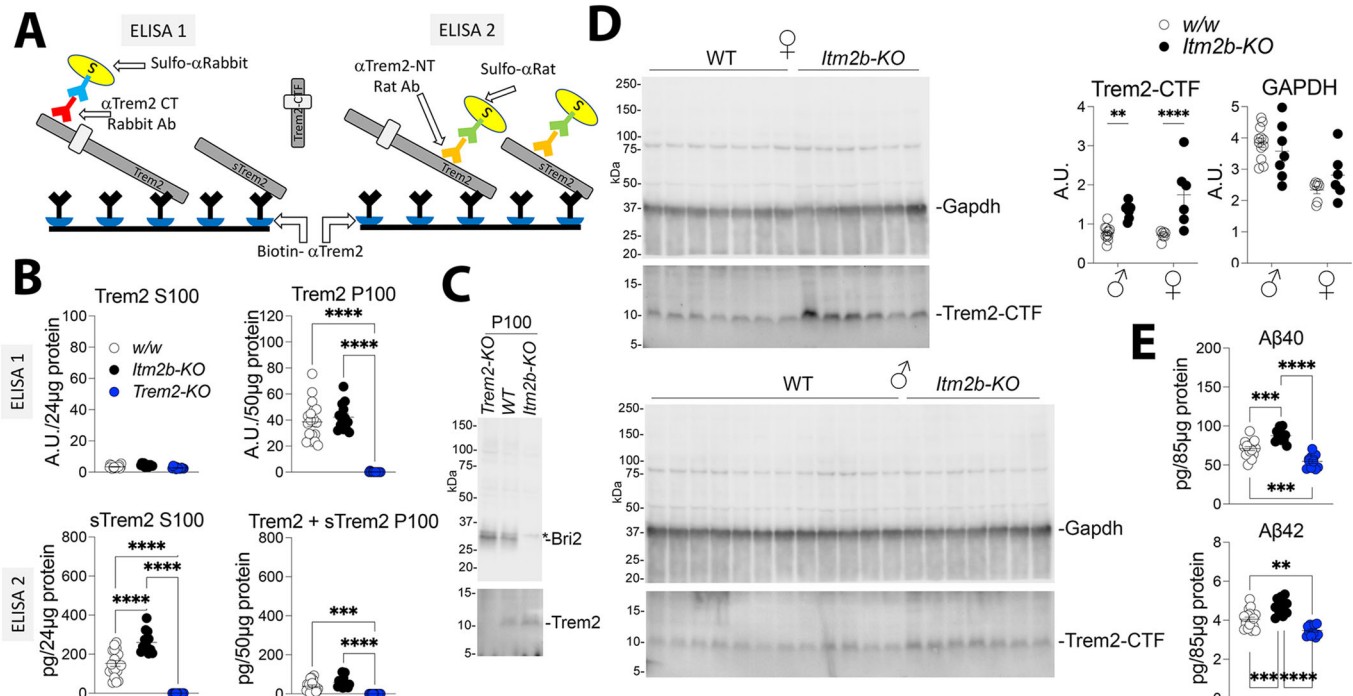

**Figure 7. Bri2 deletion results in elevated CNS levels of sTrem2 and Trem2-CTF.**

(A) Schematic representation of ELISA 1 and ELISA 2. Both ELISAs use the same Biotinylated-αTrem2 capture antibody (in black). ELISA 1 uses αTrem2-CT (red) + Sulfo-αRabbit (blue) detection antibodies. ELISA 2 uses αTrem2-NT (orange) + Sulfo-αRat (green) detection antibodies. Trem2 can be detected by both ELISAs, sTrem2 can be detected only by ELISA 2: neither ELISA can detect Trem2-CTF. (B) Quantification of Trem2 and sTrem2 in the P100 and S100 brain fractions of ~245 days old *w/w* control, *Itm2b-KO* and *Trem2-KO* mice. (C) Western blot analysis of P100 fractions from a representative *w/w, Trem2-KO* and *Itm2b-KO* P100 sample with αTrem2-CT and an αBri2 antibody. *Indicates a non-specific band. (D) Detection and quantification of Trem2-CTF in the P100 fraction by Western blot analysis and with Image Lab software; GAPDH was used as a loading control. (E) ELISA measurements of endogenous Aβ40 and Aβ42 in brain homogenates of *w/w, Trem2-KO* and *Itm2b-KO* animals. Data information: This figure encompasses the comprehensive dataset employed for these specific experiments. The membrane in (C) was cut at the 20 and 15 kDa molecular weight marker (MWM). The upper section was probed with the anti-Bri2 antibody, while the lower section was probed with the Trem2-CT antibody. Similarly, in (D), the two membranes were divided at the 20 and 15 kDa MWM. The upper portion was probed with the anti-Gapdh antibody, while the lower portion was probed with the Trem2-CT antibody. Statistical comparisons among the groups were conducted using one-way ANOVA followed by post-hoc Tukey's multiple comparisons test when ANOVA showed significant differences (B, E); two-way ANOVA followed by post-hoc Sidak's multiple comparisons test when ANOVA showed significant differences (D). **$P < 0.01$, ***$P < 0.001$, ****$P < 0.0001$. The data presented are derived from are from *w/w* control, females $n = 7$, males $n = 12$; *Itm2b-KO* females $n = 6$, males $n = 7$; *Trem2-KO*, females $n = 6$, males $n = 7$; the letter "n" indicates biological replicates. All data are expressed as means $+/-$ SEM. Source data are available online for this figure.

this observation suggests the possibility of sex differences in Trem2 processing and the impact of Bri2, further experiments are required to clarify this point. In summary, the absence of BRI2 in *Itm2b-KO* mice leads to increased levels of sTrem2 and Trem2-CTF, which suggests an increase in Trem2 processing by α-secretase. However, the levels of Trem2 itself remain unchanged in *Itm2b-KO* brains, which suggests that compensatory mechanisms are in play in vivo to maintain Trem2 levels.

Next, we measured Aβ40 and Aβ42 levels in *Itm2b-KO, Trem2-KO,* and control WT mice. Since rodent and human APP differ in the Aβ region by 3 amino acids (Aβ P5, P10, and P13), we modified the human Aβ40/Aβ42 ELISA system from Meso Scale Diagnostic. This system employs capture antibodies specific for the distinct COOH-termini of Aβ40 and Aβ42, which are 100% conserved between humans and rodents. Consequently, this ELISA can capture both human and rodent Aβ40 and Aβ42. For mouse Aβ detection, we replaced the anti-human Aβ antibody 6E10 included in the ELISA kit with M3.2, a mouse monoclonal antibody specific for rodent Aβ. The specificity of this assay has been previously reported (Pham et al, 2022). Consistent with previous reports that BRI2 reduces Aβ

production by inhibiting APP cleavage (Matsuda et al, 2008; Tamayev et al, 2011), Aβ40 and Aβ42 levels were increased in *Itm2b-KO* mice compared to controls (Fig. 7E). These findings suggest that Bri2 can inhibit both Trem2 and APP cleavage, and loss of Bri2 function leads to increased processing of both proteins. *Trem2-KO* mice, on the other hand, had significantly lower CNS Aβ40 and Aβ42 levels (Fig. 7E). This finding may appear contradictory to the notion that TREM2 mediates Aβ clearance (Lessard et al, 2018; Yeh et al, 2016). However, previous studies primarily investigated human Aβ, utilizing either transgenic models expressing human APP or in vitro oligomeric forms of human Aβ42. It is noteworthy that the 3-amino acid differences between rodent and human Aβ greatly influence the propensity of Aβ to form oligomers. Human Aβ species are known to have a heightened tendency to aggregate compared to their rodent counterparts. If Trem2 primarily facilitates the clearance of oligomeric or aggregated Aβ species, while Trem2 deletion enhances the efficiency of clearing soluble Aβ, *Trem2 KO* mice might exhibit reduced Aβ levels, particularly if the majority of mouse Aβ forms are monomeric. Moreover, Trem2 deletion could conceivably lower Aβ levels by influencing Aβ

generation, possibly through a trans-cellular mechanism. Although we currently lack data to definitively reconcile this apparent contradiction, these factors underscore the intricate role played by TREM2 in Aβ regulation.

In summary, loss of Bri2 expression in vivo causes an increase of Trem2-CTF and sTrem2, the two products of α-secretase-processing of Trem2, which suggests an increase in Trem2 processing by α-secretase. This data, together with the evidence that BRI2 overexpression in cells lines causes a decrease in Trem2-CTF and sTrem2 levels (Fig. 6), suggests that, physiologically, BRI2 dampens α-secretase-mediated processing of TREM2.

## Microglia-specific reduction of *Itm2b* increases CNS sTrem2 levels

The data in transfected cells suggest that the effect of BRI2 on Trem2 is cell autonomous. If this were the case, reducing Bri2 expression in vivo only in microglia should cause an increase in sTrem2 levels. To determine the cell-autonomous or non-cell autonomous nature of Bri2's effect on sTrem2 levels in vivo, we took advantage of *Cx3cr1^CreER/wt^* (Yona et al, 2013) and *Itm2b*-Floxed (*Itm2b^f/f^*) (Matsuda et al, 2008; Yao et al, 2019) mice. The *Cx3cr1^CreER/wt^* animals contain a modified version of the chemokine (C-X3-C) receptor 1 (Cx3cr1) gene, with an inserted CreERT2 sequence followed by an internal ribosome entry site and an enhanced yellow fluorescent protein (EYFP). This results in the expression of Cre-ERT2 and EYFP only in microglia in the brain. The Cre-ERT2 fusion protein requires the presence of tamoxifen to translocate from the cytosol to the nucleus, where it can mediate loxP-loxP recombination. In *Itm2b^f/f^* mice, exon 3 of the *Itm2b* gene is flanked by two loxP sites. Therefore, in *Itm2b^f/f^*:*Cx3cr1^CreER/wt^* animals, tamoxifen administration should induce Cre-ERT2-mediated conversion of *Itm2b^f^* alleles into *Itm2b-KO* alleles, leading to suppression of *Itm2b* expression specifically in microglia but not in other brain cell types.

To verify that Cre-ERT2 and EYFP are only expressed in microglia, we prepared cell suspensions from brain tissue isolated from ~380 days old *Cx3cr1^CreER/wt^* and *Cx3cr1^wt/wt^* animals after intracardiac catheterization and perfusion. Cells were stained with the microglia-specific anti-CD11b-APC-conjugated antibody and analyzed by Fluorescence-activated cell sorting (FACS). The vast majority of *Itm2b^wt/wt^*:*Cx3cr1^CreER/wt^* microglia (CD11b+) were EYFP+ and >99% of EYFP+ cells were CD11b+ and CD45^low^ (Fig. 8A), confirming that the Cre-ERT2 and EYFP expression is indeed restricted to microglia in the brain.

CreERT2 mouse lines exhibit some degree of leakiness, which causes tamoxifen independent Cre activity (Alvarez-Aznar et al, 2020). To test whether *Itm2b^f/f^*:*Cx3cr1^CreER/wt^* animals showed tamoxifen independent Cre recombinase activity, we perfused ~380 days old *Itm2b^f/f^*:*Cx3cr1^CreER/wt^* animals and prepared cell suspensions from brain tissue. After sorting the cells into EYFP+ (microglia) and EYFP- (non-microglia) cell populations (Fig. 8B), the genomic DNA was isolated and analyzed by PCR tests to amplify the *Itm2b^f^* and the recombined *Itm2b^KO^* alleles (Fig. 8C). All microglia samples (EYFP+) analyzed showed the presence of both the *Itm2b^f^* and *Itm2b^KO^* alleles (Fig. 8D), indicating tamoxifen independent Cre recombinase activity in microglia. Non-microglia samples (EYFP-) showed only the *Itm2b^f^* allele (Fig. 8D), indicating that Cre-ERT2 expression and partial *Itm2b* inactivation is

restricted to microglia. Since the PCR method used was not quantitative, the percentage of *Itm2b^f^* alleles that had undergone recombination-conversion to *Itm2b^KO^* alleles could not be determined.

Next, ~425 days old *Itm2b^f/f^*:*Cx3cr1^CreER/wt^* and *Itm2b^w/w^*:*Cx3cr1^CreER/wt^* animals were perfused and cell suspensions from brain tissue were sorted into EYFP+ microglia and EYFP- non-microglia cell populations. Quantitative RT-PCR analysis showed downregulation of *Itm2b* mRNA in *Itm2b^f/f^*:*Cx3cr1^CreER/wt^* microglia compared to *Itm2b^w/w^*:*Cx3cr1^CreER/wt^* microglia and confirmed that *Itm2b* mRNA levels are significantly higher in microglia compared to non-microglia (Fig. 8E). Interestingly, we also observed a downregulation of *Trem2* mRNA expression in *Itm2b^f/f^*:*Cx3cr1^CreER/wt^* microglia when compared to *Itm2b^w/w^*:*Cx3cr1^CreER/wt^* microglia (Fig. 8E).

As we observed a microglia-specific partial loss of *Itm2b* mRNA in *Itm2b^f/f^*:*Cx3cr1^CreER/wt^* mice without tamoxifen treatment, we measured sTrem2 levels in ~425 days old *Itm2b^f/f^*:*Cx3cr1^CreER/wt^* and *Itm2b^f/f^*:*Cx3cr1^wt/wt^* littermates, without tamoxifen treatment. We found that sTrem2 levels were significantly increased in *Itm2b^f/f^*:*Cx3cr1^CreER/wt^* as compared to *Itm2b^f/f^*:*Cx3cr1^wt/wt^* littermates (Fig. 8F). The increase in sTrem2 levels in *Itm2b^f/f^*:*Cx3cr1^CreER/wt^* mice without tamoxifen treatment indicates that the partial loss of Bri2 function in microglia alone is sufficient to increase sTrem2 levels. This data supports the idea that Bri2 inhibits α-secretase processing of Trem2 through a cell-autonomous mechanism, possibly mediated by the Bri2/Trem2 interaction. Nevertheless, it cannot be ruled out that Bri2 may also impact sTrem2 levels through a non-cell-autonomous mechanism or by other microglial-specific mechanisms.

## FDD-KI mice show elevated CNS levels of Trem2-CTF and sTrem2

Both FBD and FDD patients exhibit a noteworthy neuroinflammatory component. We have developed FDD-KI and FBD-KI mice. It is important to highlight that, unlike transgenic models where disease-associated minigenes are overexpressed under neuronal-specific promoters, potentially missing the impact of pathogenic mutations in other CNS cell types like microglia, KI animals are engineered to introduce pathogenic mutations into the host rodent genes via gene editing. This approach faithfully replicates the genetic basis of the disease. In KI models, the expression of mutant genes is regulated by the native control elements, ensuring that disease-related proteins are produced in a biologically relevant quantity, cell specificity, and following a natural spatial and temporal pattern (D'Adamio, 2023). Thus, in these KI animals, the pathogenic mutations are expected to be expressed in microglia in a manner that mirrors the physiological conditions. Interestingly, the Danish mutant BRI2 protein in FDD-KI rodents is unstable and undergo rapid degradation (Tamayev et al, 2010b; Yin et al, 2021a, b), mirroring the significant reduction in mature BRI2 levels, which is the functional form of BRI2, observed in brain lysates from FDD patients (Matsuda et al, 2011b; Tamayev et al, 2012a, 2010a, b). Building upon this observation, we hypothesized that pathogenic *ITM2B* mutations may lead to an upregulation of TREM2 processing. To test this hypothesis, we measured sTrem2 levels by ELISA 1 in the S100 fraction and Trem2-CTF by WB in the P100 fraction of brain homogenates from FDD-KI mice and WT control littermates. Our results revealed a significant enrichment of sTrem2 in the S100 fraction of FDD-KI mice compared to

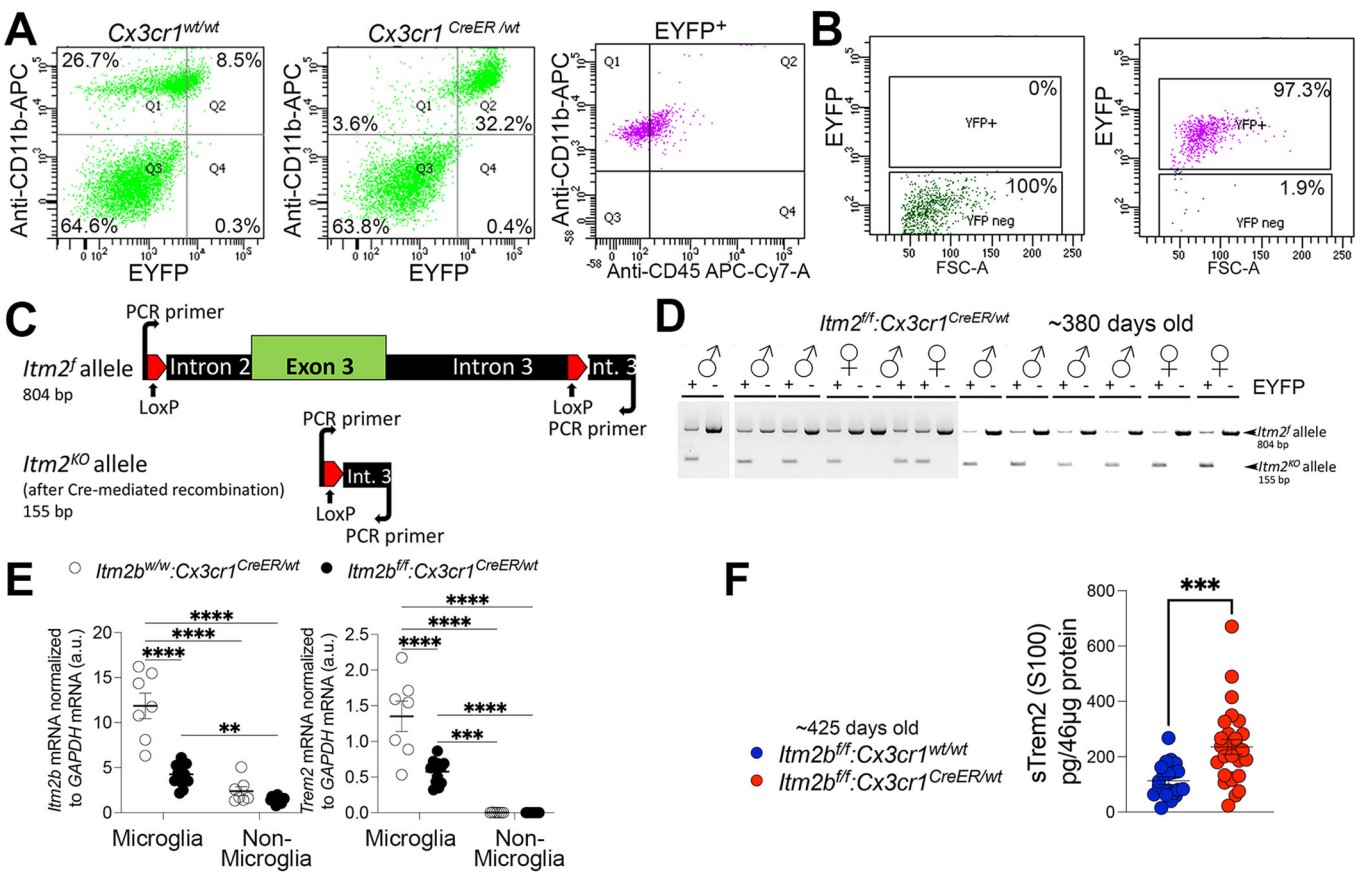

**Figure 8. Microglial-specific depletion of Bri2 increases CNS levels of sTrem2.**

(A) CD11b and CD45 staining, and FACS analysis of brain cells isolated from *Cx3cr1CreER/wt and Cx3cr1wt/wt* animals. (B) FACS analysis of sorted EYFP+ (microglia) and EYFP- (non-microglia) brain cell populations from *Itm2bf:Cx3cr1CreER/wt* animals. (C) Schematic representation of the PCR test used to identify the *Itm2bf* and *Itm2bKO* alleles. (D) PCR analysis of genomic DNA isolated from EYFP+ and EYFP- cells sorted from *Itm2bf/f:Cx3cr1CreER/wt* brains. (E) Analysis of *Itm2b* and *Trem2* mRNA expression in sorted EYFP+ (microglia) and EYFP- (non-microglia) brain cell populations from ~14 months-old *Itm2bf/f:Cx3cr1CreER/wt* and *Itm2bw/w:Cx3cr1CreER/wt* animals. (F) ELISA 2 was used to measure sTrem2 levels in *Itm2bf/f:Cx3cr1CreER/wt* and *Itm2bf/f:Cx3cr1wt/wt* littermates. Data information: Statistical comparisons among the groups were conducted two-way ANOVA followed by post-hoc Sidak's multiple comparisons test when ANOVA showed significant differences (E); two-tailed unpaired *t* test (F). **$P < 0.01$, ***$P < 0.001$, ****$P < 0.0001$. The data presented are derived from: (E) *Itm2bf/f:Cx3cr1CreER/wt*, females $n = 5$, males $n = 7$; *Itm2bw/w:Cx3cr1CreER/wt*, females $n = 3$, males $n = 4$; (F) *Itm2bf/f:Cx3cr1CreER/wt*, females $n = 15$, males $n = 12$; *Itm2bf/f:Cx3cr1wt/wt*, females $n = 10$, males $n = 11$; the letter "n" indicates biological replicates. All data are expressed as means $+/-$ SEM. Source data are available online for this figure.

control WT animals (Fig. 9A). In addition, quantification of Trem2-CTF levels in the P100 brain fractions demonstrated significantly higher levels in both FDD-KI brains compared to control WT animals (Fig. 9B,C). Overall, these findings indicate that the Danish mutation leads to increased levels of sTrem2 and Trem2-CTF.

### *Itm2b-KO* primary microglia show enhanced Trem2 processing but not overt increase in α-secretase activity

To further investigate whether BRI2's influences Trem2 processing is cell-autonomous, we conducted experiments using primary mouse microglia. These cultures exhibit a high degree of purity and are largely devoid of significant levels of other neuronal cell types. Consequently, this setup eliminates potential cell-non-autonomous effects, which can impact Trem2 processing in the complex environment of the CNS. In our initial assessment, we measured sTrem2 levels in the conditioned media of primary

microglia from *Itm2b-KO* and WT mice after a 24-hour culture period (Fig. 10A). To quantify sTrem2 in the conditioned culture media, we employed two methods: ELISA and WB after deglycosylation of the cell culture conditioned media. Both techniques revealed a substantial increase in sTrem2 levels in the culture media collected from *Itm2b-KO* primary microglia. However, the ELISA, which boasts high specificity due to the use of two distinct anti-sTrem2 antibodies for detection, proved to be more sensitive than WB. Specifically, we could not detect sTrem2 by WB when concentrations fell below ~500 pg/ml (as determined by ELISA). Consequently, we opted to utilize ELISA for subsequent experiments for its enhanced sensitivity and high specificity.

Next, primary microglia were incubated in serum-free media for 5 h to minimize the influence of serum components on microglial activation. Following this, the cell culture media was replaced with fresh serum-free media, either with or without the addition of *E. coli*. Subsequent analysis using sTrem2 ELISA on the cell culture

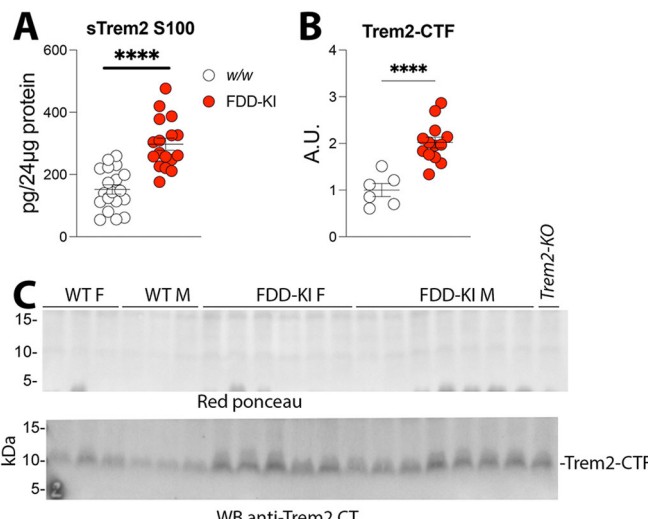

**Figure 9. Elevated CNS levels of sTrem2 and Trem2-CTF in FDD-KI mice.**

(A) Quantification of sTrem2 in the S100 brain fractions of ~245 days old control and FDD-KI mice. Data were analyzed by unpaired $T$-test. All data are shown as means $+/-$ SEM: ****$P < 0.0001$. (B) Quantification of Trem2-CTF in the P100 fraction by Western blot analysis and with Image Lab software. (C) Red Ponceau staining (upper panel) was used to normalize the Trem2-CTF signal (lover panel) obtained by Western blot. Data information: Statistical comparisons among the groups were conducted using unpaired $T$-test. ****$P < 0.0001$. The data presented are derived from: (A) $w/w$ mice (females, $n = 7$; males, $n = 12$) and FDD-KI mice (females, $n = 8$; males, $n = 9$) mice; (B) $w/w$ mice (females, $n = 3$; males, $n = 3$) and FDD-KI mice (females, $n = 6$; males, $n = 7$) mice; the letter "n" indicates biological replicates. All data are expressed as means $+/-$ SEM. Source data are available online for this figure.

media, after 5 h of serum-free culture, revealed a significant increase in sTrem2 levels in the conditioned media collected from *Itm2b-KO* primary microglia cultures compared to WT primary microglia cultures (Fig. 10B).

Cell culture media were collected from cells treated with *E. coli* or vehicle after 1 and 2 h of treatment, while cells were lysed for WB analysis at the 2-h time-point. We observed that *E. coli* significantly elevated sTrem2 levels compared to untreated cells within the same genotype (Fig. 10C). Notably, this increase was significantly higher in *Itm2b-KO* primary microglia compared to WT primary microglia cultures. Importantly, even without *E. coli* treatment, the conditioned media from *Itm2b-KO* primary microglia cultures contained significantly higher sTrem2 levels compared to WT cells cultures, with this difference being significant at the 2-h time-point (Fig. 10C).

To further assess the Trem2 processing status, lysates were analyzed for levels of Trem2 full-length (f.l.) and Trem2-CTF. Prior to Western blot analysis, samples were deglycosylated to aid visualization and quantification of full length Trem2. As illustrated in Fig. 10D, *E. coli* treatment significantly increased Trem2-CTF levels while reducing Trem2 f.l. levels, resulting in a significant elevation of the Trem2-CTF/Trem2 f.l. ratio. This increase was significantly higher in *Itm2b-KO* primary microglia cultures compared to WT cells cultures. Even without *E. coli* stimulation, the Trem2-CTF/Trem2 f.l. ratio was higher in *Itm2b-KO* primary microglia cultures compared to WT cells cultures (Fig. 10E). Analysis of *Itm2b* and *Trem2* mRNA expression confirms lack of

*Itm2b* expression in *Itm2b-KO* primary microglia and revealed that *Trem2* mRNA expression was not elevated in *Itm2b-KO* primary microglia (if anything, it exhibited a slight decrease, although not statistically significant) (Fig. 10F). Thus, the rise in sTrem2 levels observed in *Itm2b-KO* primary microglia does not appear to be driven by an increase in *Trem2* gene expression.

In a second series of primary microglia cultures, we observed similar effects, albeit with some variations that are expected when comparing primary microglia cultures grown at different times. In line with the previous experiment, *E. coli* significantly increased sTrem2 levels compared to untreated cells within the same genotype, and this increase was significantly higher in *Itm2b-KO* primary microglia cultures compared to WT cells (Fig. 10G). Furthermore, the conditioned media from unstimulated *Itm2b-KO* primary microglia cultures contained significantly higher sTrem2 levels compared to WT cells at both 1-h and 2-h time points. *E. coli* treatment also significantly elevated Trem2-CTF levels while reducing Trem2 f.l. levels, leading to a substantial rise in the Trem2-CTF/Trem2 f.l. ratio (Fig. 10H). This ratio was significantly higher in *Itm2b-KO* primary microglia cultures compared to WT cells. However, in contrast to the previous experiment, in these cultures, the Trem2-CTF/Trem2 f.l. ratio was similar in *Itm2b-KO* primary microglia cultures compared to WT cells without *E. coli* stimulation. Analysis of *Itm2b* and *Trem2* mRNA expression confirms the absence of *Itm2b* expression in *Itm2b-KO* primary microglia and shows a significant reduction in *Trem2* mRNA levels in *Itm2b-KO* primary microglia (Fig. 10I). In summary, these findings consistently support the hypothesis that Bri2 can modulate Trem2 processing in a cell-autonomous manner, with Bri2 expression acting to limit Trem2 processing. The observation that *Trem2* gene expression is downregulated in *Itm2b-KO* primary microglia, consistent with the findings in CNS-derived microglia with a cell-specific *Itm2b*-deletion (Fig. 8E), implies a potential positive regulatory role of Bri2 on *Trem2* gene expression.

In a preliminary experiment, *Itm2b-KO* primary microglia treated with a 2μM concentration of recombinant BRI2-ECD for 24 h. We tested BRI2-ECD over BRI2-BRICHOS in this experiment because it contains both TREM2-binding domains. BRI2-ECD treated primary microglia exhibited a reduced Trem2-CTF/Trem2 f.l. ratio (Fig. EV4A), a decrease in sTrem2 levels in the cell culture media (Fig. EV4B), and a reduction in Syk phosphorylation (Fig. EV4C). These findings suggest that BRI2-ECD might compensate for certain functions of Bri2 on Trem2 processing and activity.

Bri2 deletion could influence Trem2 processing by modulating the expression and overall activity of Adam10 and/or Adam17, the two α-secretases responsible for processing Trem2. To explore these possibilities, we examined the mRNA expression of *Adam10* and *Adam17* in WT and *Itm2b-KO* microglia. We observed that the expression of both genes is not elevated in *Itm2b-KO* primary microglia (Fig. 11A,B). Instead, there appears to be a down-regulation of both *Adam10* and *Adam17* in *Itm2b-KO* primary microglia compared to their WT counterparts: *Adam10* down-regulation reached statistical significance in one experiment (Fig. 11A), while *Adam17* downregulation was significant in the other experiment (Fig. 11B). In addition, we assessed Adam17 protein levels in primary microglia following *E. coli* stimulation and found comparable expression levels between *Itm2b-KO* and WT primary microglia in both experiments (Fig. 11C,D).

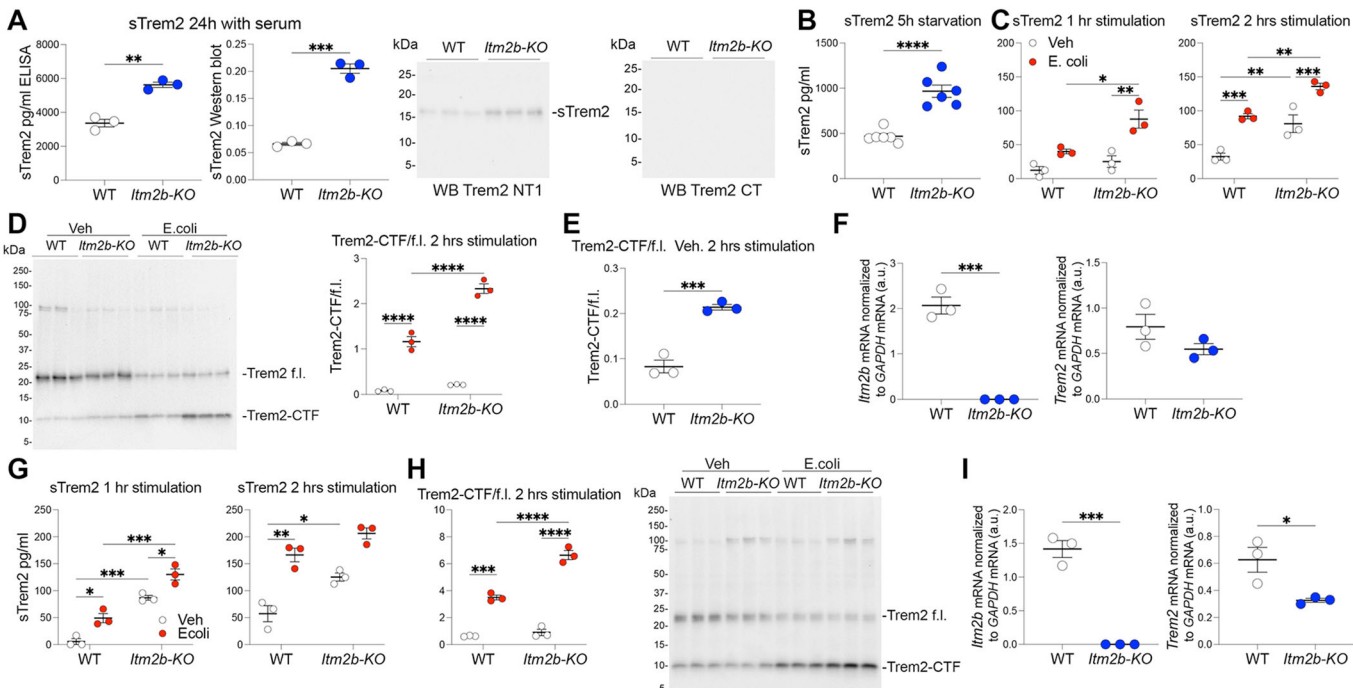

**Figure 10. Enhanced Trem2 processing in *Itm2b-KO* primary microglia.**

(A) Quantification of sTrem2 in the conditioned media of WT and *Itm2b-KO* primary microglia using ELISA (left panel) and Western blot of deglycosylated conditioned media with Trem2 NT antibody (quantification of Western blot is shown in the second panel). Trem2 CT antibody does not show any signal. (B) Quantification of sTrem2 in the conditioned media of WT and *Itm2b-KO* primary microglia after 5 h of serum starvation by ELISA. (C) Quantification of sTrem2 in the conditioned media of WT and *Itm2b-KO* primary microglia after 1 and 2 h with either vehicle (Veh) or *E. coli* by ELISA. (D) Western blot of deglycosylated cell lysates from WT and *Itm2b-KO* primary microglia, 2 h after *E. coli* stimulation, with Trem2 CT antibody to visualize Trem2 f.l. and Trem2-CTF, along with quantification of the Trem2-CTF/Trem2 f.l. ratio. (E) Quantification of the Trem2-CTF/Trem2 f.l. ratio for only the two untreated (Veh) groups. (F) Analysis of *Itm2b* and *Trem2* mRNA expression in WT and *Itm2b-KO* primary microglia using quantitative RT-PCR. (G) A second set of biological replicates was analyzed following the same procedure as in panel (C). (H) A second set of biological replicates was analyzed following the same procedure as in panel (D). (I) A second set of biological replicates was analyzed following the same procedure as in panel (F). Data information: Statistical comparisons among the groups were conducted using either a two-tailed unpaired *t*-test (A, B, E, F, I) or a two-way ANOVA followed by post-hoc Sidak's multiple comparisons test when ANOVA indicated significant differences (C, D, G, H). *$P < 0.05$, **$P < 0.01$, ***$P < 0.001$, ****$P < 0.0001$. The data presented are derived from WT primary microglia cultures ($n = 3$ for (A, C, D, E–I), $n = 6$ for (B)) and *Itm2b-KO* primary microglia ($n = 3$ for Exp. and $n = 3$ for (A,C,D,E-I), $n = 6$ for (B)); the letter "n" indicates biological replicates, except for (B), which includes 3 biological replicates with 2 technical replicates each. Each biological replicate was composed of primary microglia generated from 2 P2 pups. All data are expressed as means +/− SEM. Source data are available online for this figure.

Mature TNFα is a well-established substrate of Adam10 and Adam17, leading to the release of active soluble TNFα (sTNFα). The activation of microglia by *E. coli* rapidly induces mature TNFα shedding and subsequent sTNFα production, a pattern reminiscent of what we have observed for sTrem2 production (Fig. 10). Considering this, we examined sTNFα levels in primary microglia after 1-h *E. coli* stimulation, when sTrem2 levels significantly increased in stimulated *Itm2b-KO* compared to stimulated WT primary microglia (Fig. 10C,G). As anticipated, *E. coli* stimulation indeed led to a noteworthy rise in sTNFα release (Fig. 11E,F), but this increase displayed no significant difference between *Itm2b-KO* and WT primary microglia. While we cannot rule out the possibility that *Itm2b-KO* deletion might reduce mature TNFα availability, overall, the data shown in Fig. 11 do not substantiate the hypothesis that Bri2 deletion increases expression and overall activity of Adam10/Adam17.

Altogether, these findings align with the hypothesis that Bri2 interacts with and inhibits Trem2 processing. However, they also suggest potential functional interactions at the mRNA level, as evidenced by the decrease in *Trem2* mRNA in *Itm2b-KO* primary

microglia. Additionally, the data indicate a probable regulation of Trem2 f.l. stability via pathways unrelated to α-secretase processing, given the relatively stable Trem2 f.l. levels in *Itm2b-KO* primary microglia despite significant increases in Trem2-CTF and sTrem2.

### *E. coli*-induced p38-MAPK activation is attenuated in *Itm2b-KO* primary microglia

The TREM2 activation pathway hinges on the interplay between TREM2 and DAP12, culminating in the phosphorylation of tyrosine residues within the DAP12-ITAM domain, a process mediated by Src family kinases. This phosphorylation event triggers downstream signaling cascades, including the activation of Syk. Furthermore, PLCγ1/2 serves as a pivotal regulator of intracellular calcium signaling and cytokine production downstream of Syk. Syk-mediated phosphorylation of PLCγ1 at Tyr783 unleashes its enzymatic activity (Colonna, 2003; Peng et al, 2010). To determine whether Bri2 deficiency modulates the Trem2 pathway, we examined Syk phosphorylation at Tyr525/526 and PLCγ1

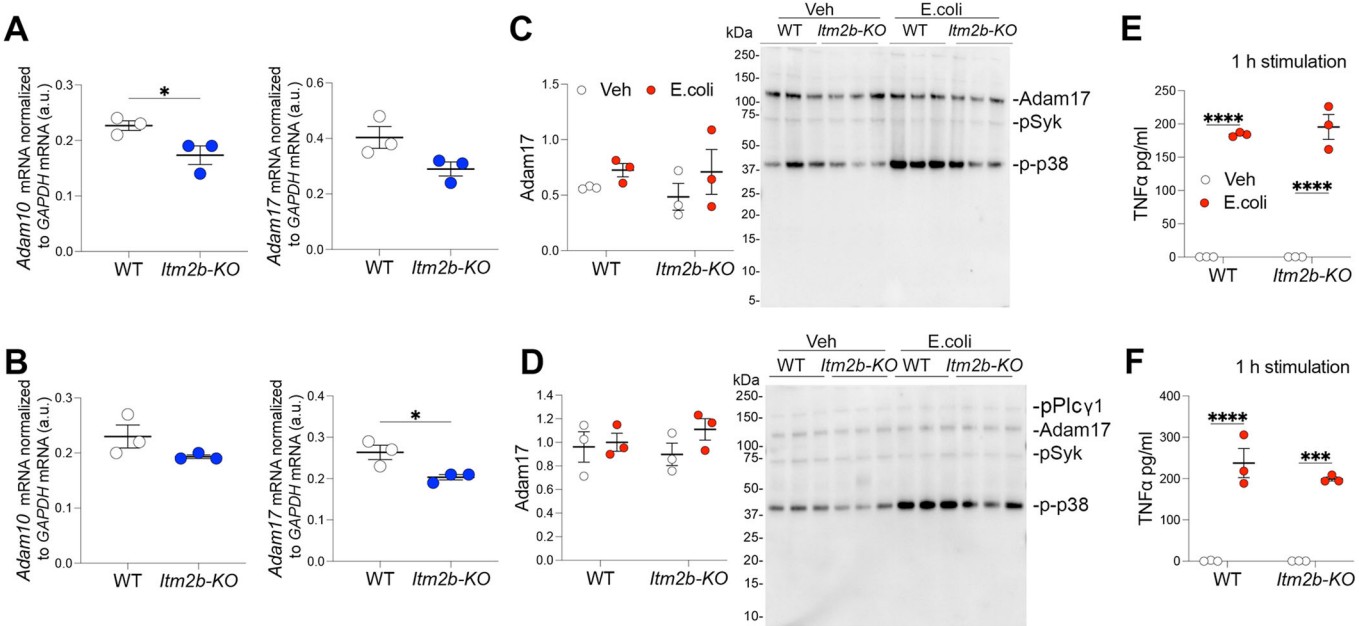

**Figure 11. Comparable α-secretase activity in WT and *Itm2b-KO* primary microglia.**

(A) Quantification of *Adam10* and *Adam17* mRNA expression in WT and *Itm2b-KO* primary microglia using quantitative RT-PCR. (B) The experiment is a replicate of the one described in (A), using separate biological replicates. (C) Quantification of Adam17 in cell lysates from WT and *Itm2b-KO* primary microglia taken 2 h post *E. coli* stimulation (including controls treated solely with vehicle, Veh). The Western blot corresponding to the quantification is shown to the right and was performed using the membrane probed with α-pSyk, α-pPlcγ1, and α-p-p38 antibodies in Fig. 12A. (D) The experiment is a replicate of the one described in (C), using separate biological replicates. The Western blot corresponding to the quantification is shown to the right and was performed using the membrane probed with α-pSyk, α-pPlcγ1, and α-p-p38 antibodies in Fig. 12C. (E) ELISA-based quantification of TNFα secretion in the culture supernatants of WT and *Itm2b-KO* primary microglia treated with *E. coli* for 1 h, including controls treated solely with vehicle (Veh). (F) The experiment is a replicate of the one described in (E), using separate biological replicates. Data information: This figure encompasses the comprehensive dataset employed for these specific experiments. We have included the images of the complete membranes used for Western blot analyses, without any cropping of information above or below the targeted signals. Statistical comparisons among the groups were conducted using either a two-tailed unpaired *t*-test (A, B) or a two-way ANOVA followed by post-hoc Sidak's multiple comparisons test when ANOVA indicated significant differences (C–F). *$P < 0.05$, ***$P < 0.001$, ****$P < 0.0001$. The data presented are derived from WT primary microglia cultures ($n = 3$) and *Itm2b-KO* primary microglia ($n = 3$); the letter "n" indicates biological replicates. Each biological replicate was composed of primary microglia generated from 2 P2 pups. All data are expressed as means $+/-$ SEM. Source data are available online for this figure.

phosphorylation at Tyr783, but we could not detect any noticeable increase in the phosphorylation of Syk and PLCγ1 in both WT and *Itm2b-KO* primary microglia following *E. Coli* stimulation (Fig. 12A–D). These signaling assays were conducted at a single time point, specifically 2 h post *E. coli* stimulation. Hence, it is plausible that the phosphorylation levels of Syk and PLCγ1 had already reached their peak and subsequently diminished, accounting for the absence of observable changes. Additional experiments conducted at distinct time points may be indispensable to resolve this matter.

Notably, the MAPK p38 pathway is also activated by *E. coli*. A recent study has demonstrated that a reduction in Trem2 expression intensifies, while Trem2 overexpression suppresses, the p38 MAPK signaling pathway in microglia (Zhang et al, 2020), directly linking Trem2 to p38 activation in microglia. Thus, we tested whether Bri2 might exert an influence on p38 phosphorylation. Intriguingly, whereas p38 phosphorylation exhibited a significant increase in WT primary microglia 2 h after *E. coli* treatment, *Itm2b-KO* primary microglia did not display significant increase in p38 phosphorylation induced by *E. coli* (Fig. 12A–D). These findings suggest that Bri2 has an impact on p38-MAPK activation in microglia. However, it is important to note that these results do not definitively establish whether this alteration is

Trem2-dependent. Further investigations are warranted to elucidate this aspect.

## Diminished phagocytic activity and altered cytokine secretion in *Itm2b-KO* primary microglia

The TREM2 p.H157Y mutation, associated with late-onset AD, occurs at the P1 position of the TREM2 α-secretase cleavage site and enhances the shedding of mutant TREM2 (Thornton et al, 2017). Cells expressing TREM2 p.H157Y display a significantly reduced ability to phagocytose *E. coli* conjugated to pHrodo when compared to cells expressing WT TREM2 (Schlepckow et al, 2017). Consequently, we investigated whether Bri2 deletion affects the phagocytic activity of *E. coli* conjugated to pHrodo. Our experiments consistently revealed a substantial impairment in *E. coli* phagocytosis in *Itm2b-KO* primary microglia (Fig. 13A–D). While it is conceivable that Bri2 might affect phagocytosis through Trem2-independent pathways, the resemblance between the phenotype resulting from Bri2 deletion and that of a TREM2 pathogenic variant linked to heightened TREM2 shedding suggests that Bri2 deletion could potentially impair phagocytosis by promoting Trem2 shedding. Although the exact mechanisms through which Bri2 is involved in microglial phagocytosis remain

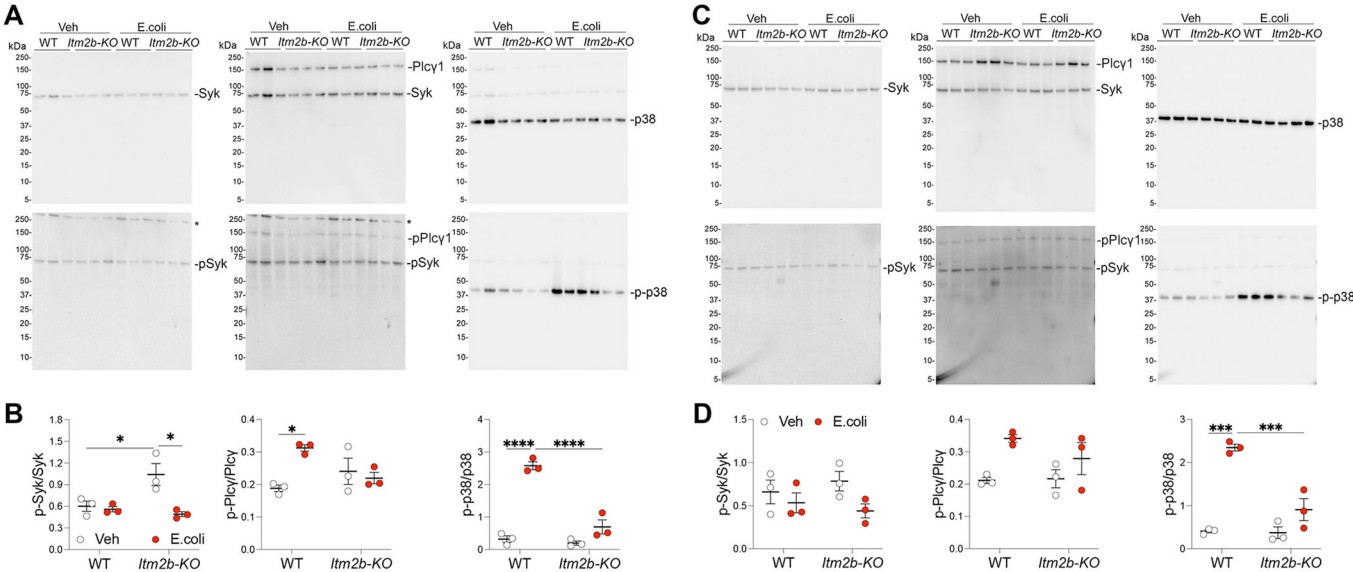

**Figure 12. Reduced p38 MAPK activation in *Itm2b-KO* primary microglia.**

(A) Western blot of cell lysates from both WT and *Itm2b-KO* primary microglia, taken 2 h post *E. coli* stimulation (along with controls treated solely with vehicle, Veh), for the detection of Syk, pSyk, Plcγ1, pPlcγ1, p38, and p-p38. Two membranes were employed for analysis. The first membrane was initially probed with an α-Syk antibody, followed by an α-Plcγ1 antibody, and subsequently with an α-p38 antibody. On the other hand, the second membrane was initially probed with an α-pSyk antibody, followed by an α-pPlcγ1 antibody, and concluded with an α-p-p38 antibody. *Indicates a non-specific band. (B) Quantification of Western blots from (A), depicting the ratios of pSyk/Syk, pPlcγ1/Plcγ1, and p-p38/p38, which reflect the activation levels of Syk, Plcγ1, and p38, respectively. (C) The experiment is a replicate of the one described in (A), using separate biological replicates. (D) Quantification of Western blots shown in (C). Data information: This figure encompasses the comprehensive dataset employed for these specific experiments. We have included the images of the complete membranes used for Western blot analyses, without any cropping of information above or below the targeted signals. Statistical comparisons among the groups were conducted using a two-way ANOVA followed by post-hoc Sidak's multiple comparisons test when ANOVA indicated significant differences. *$P < 0.05$, ***$P < 0.001$, ****$P < 0.0001$. The data presented are derived from WT primary microglia cultures ($n = 3$) and *Itm2b-KO* primary microglia ($n = 3$); the letter "n" indicates biological replicates. Each biological replicate was composed of primary microglia generated from 2 P2 pups. All data are expressed as means $+/-$ SEM. Source data are available online for this figure.

unresolved by these experiments, the impact of Bri2 on microglial phagocytic activity underscores the critical role played by BRI2 in microglial functions.

Next, we assessed whether Bri2 deletion had any impact on cytokine and chemokines secretion in response to *E. Coli*. We analyzed the levels of IL-10, IL-16, IL-1β, IL-6, Cxcl-1, MCP-1, MIP-1α, MIP-2, MIP-3, and TNFα, in the conditioned media from the 2-h cell cultures depicted in Fig. 14A. Although there was some variation in the strength of cytokine secretion induced by *E. coli* between the two experiments, primarily noticeable in the cases of IL-1β and MCP-1, and how the genotype influenced *E. coli*-induced secretion, particularly in the cases of MCP-1 and TNFα, there were consistent trends. IL-6 and Cxcl-1 induction were significantly higher in *Itm2b-KO* samples compared to WT samples in both experiments (Fig. 14A). In contrast, MIP-1α induction was significantly reduced in *Itm2b-KO* samples compared to WT samples in both experiments (Fig. 14A). In addition, IL-10 induction was diminished in *Itm2b-KO* samples compared to WT samples in both experiments, reaching statistical significance only in Exp. 1. It is worth mentioning that primary rat microglia, when subjected to oxygen glucose deprivation as an in vitro model of neonatal hypoxic-ischemic encephalopathy, exhibit heightened production of MIP-1α (also known as Ccl3) (Aoki et al, 2023). Notably, this increase in MIP-1α production can be inhibited by the p38-MAPK inhibitor SB203580HIE (Aoki et al, 2023), a pathway whose activation is diminished in *Itm2b-KO* primary microglia (Fig. 12).

Finally, we measured the levels of IL-6, Cxcl-1, MIP-1α, and TNFα in the conditioned media of *Itm2b-KO* and WT primary microglia after a 24-h culture period and following an additional 5-h incubation in serum-free media. As illustrated in Fig. 14B, while IL-6 and TNFα levels remained similar between both genotypes, Cxcl-1 secretion was significantly higher in *Itm2b-KO* samples compared to WT samples, whereas MIP-1α levels showed a converse pattern, being significantly lower in *Itm2b-KO* samples. Importantly, the patterns observed for Cxcl-1 and MIP-1α secretion under non-stimulating conditions mirror the trends observed after *E. coli* stimulation. This consistency further strengthens the credibility of these findings.

Although these experiments do not clarify whether Bri2 affects cytokine and chemokines secretion through Trem2-independent and/or Trem2-dependent pathways, nor do they address the in vivo relevance of these alterations in brain functions and neurodegeneration, the substantial impact of Bri2 deletion on both *E. coli*-induced and "basal" cytokine and chemokines secretion underscores the pivotal role of BRI2 in microglial biology.

## Discussion

The high expression of *ITM2B*, an AD-related familial dementia gene (Liu et al, 2021; Rhyu et al, 2023; Vidal et al, 1999, 2000), in both mouse and human microglia lead to the investigation of

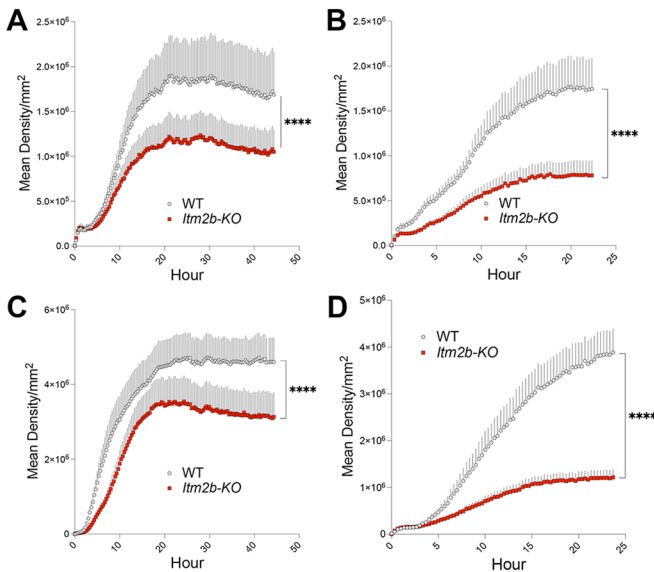

**Figure 13.  Reduced *E. coli* phagocytosis in *Itm2b-KO* primary microglia.**

(A–D) WT and *Itm2b-KO* primary microglia were incubated with *pHrodo Red E. coli* BioParticles and the kinetics of phagocytosis was measured in the IncuCyte imaging platform at 20 min intervals. Four independent experiments are shown. Data information: Statistical comparisons between genotypes were conducted using two-tailed unpaired *t*-test. ****$P < 0.0001$. The data presented are derived from WT primary microglia cultures ($n = 3$ for (**A, B**), $n = 4$ for (**C, D**)) and *Itm2b-KO* primary microglia ($n = 3$ for (**A, B**), $n = 4$ for (**C,D**)); the letter "n" indicates biological replicates. Four images (equivalent to 4 technical replicates) were captured for each sample at every time point. Each biological replicate was composed of primary microglia generated from 2 P2 pups. All data are expressed as means $+/-$ SEM. Source data are available online for this figure.

potential functional interactions between BRI2 and TREM2 in microglia, given the emerging importance of microglia and the microglia-specific gene *TREM2* in AD pathogenesis (Guerreiro et al, 2013; Schmid et al, 2002). To investigate these potential interactions, we employed a multi-faceted approach, including scRNAseq analysis, biochemical and molecular assays in transfected cells and mouse models. Our findings strongly support the hypothesis that BRI2 interacts with TREM2 and modulates the processing of TREM2 in microglia in a cell-autonomous fashion.

Using scRNAseq analysis, we identified a microglial subpopulation (cluster 3) that is almost exclusively represented in *Itm2b-KO* mice. The increase in cluster 3 in *Itm2b-KO* mice, but not *Itm2b/Trem2-dKO* mice, suggests that *Itm2b* may act epistatically to *Trem2* to regulate cluster 3 expression and that Bri2 inhibits Trem2's function: this inhibition is relieved in the absence of Bri2, leading to the expansion of clusters 3. Thus, we named this cluster the *Itm2b-Trem2* dependent cluster 3 (I/T-D3).

By which mechanism can BRI2 inhibit TREM2 function? BRI2 interacts with a region of APP containing the secretases cleavage sites and inhibits APP processing (Matsuda et al, 2008; Tamayev et al, 2012b). Deletion analysis suggested BRI2 and TREM2 interact in their extracellular domains. This was confirmed by experiments using recombinant proteins in a cell-free system, indicating a direct interaction between human TREM2 and BRI2 extracellular domains.

As Trem2 is also processed by the α-secretase (Wunderlich et al, 2013), we hypothesized that BRI2 may inhibit Trem2 processing in

a manner similar to APP. This hypothesis was tested in transiently transfected cells, and the results showed that BRI2 binds Trem2 and inhibits its processing by α-secretase, as shown by an increase of the α-secretase substrates Trem2 and a simultaneous decrease in the α-secretase products Trem2-CTF and sTrem2, and that BRI2's inhibition of α-secretase processing of Trem2 is correlated with its binding to Trem2.

Next, we tested the physiological relevance of these findings in vivo, and found that mice lacking Bri2 expression had increased levels of sTrem2 and Trem2-CTF in the brain. This suggests that Bri2 plays a role in regulating Trem2 processing in vivo.

The transient transfection experiments indicate that this effect of BRI2 is cell-autonomous and depends on the BRI2-Trem2 interaction. To test this hypothesis *in vivo*, we measured sTrem2 levels in mice with reduced Bri2 expression solely in microglia and found that these mice had increased brain levels of sTrem2 compared to control mice. To further explore the cell-autonomous influence of Bri2 on Trem2 processing, we conducted experiments using primary mouse microglia. The results revealed a significantly increased production of sTrem2 by *Itm2b-KO* microglia compared to WT controls. Furthermore, *E. coli* stimulation significantly raised Trem2 processing in both WT and *Itm2b-KO* microglia, with *Itm2b-KO* cells showing a significantly greater increase. In summary, these findings consistently support the hypothesis that Bri2 modulates Trem2 processing, limiting it, and provide strong support for the hypothesis that the expansion of the I/T-D microglia cluster in *Itm2b-KO* mice is a direct result of increased Trem2 processing. While further data are required to fully establish this claim, it is reasonable to propose that the emergence of this cluster in *Itm2b-KO* mice, where Trem2 processing is upregulated, is prevented in *Itm2b/Trem2-dKO* mice due to the simultaneous deletion of Trem2, which impedes increased Trem2 processing.

The study conducted in heterologous HEK293 cells effectively demonstrates that BRI2 diminishes α-secretase processing of TREM2 by providing clear evidence of increased levels of the TREM2 substrate, along with a simultaneous reduction in the cleavage products sTREM2 and TREM2-CTF. The correlation between the binding of BRI2 to TREM2 and the inhibition of TREM2 processing solidifies the notion that this observed effect is primarily a consequence of their direct interaction. One advantage of employing heterologous cells is the ability to isolate a specific biological effect, in this case, the impact of BRI2 binding to TREM2 on α-processing of TREM2. This isolation is essential because it enables to distinguish this effect from other potential influences stemming from various cellular components or indirect mechanisms that might be at play in cells where these pathways are naturally active. Indeed, we provide several lines of evidence suggesting other functional interactions between Bri2 and Trem2 in microglia. Firstly, we observed a decrease in *Trem2* mRNA expression in *Itm2b-KO* primary microglia and CNS-derived microglia, indicating a role of Bri2 in regulating *Trem2* gene expression. Furthermore, despite this reduction in *Trem2* mRNA and an increase in Trem2 processing, the overall levels of Trem2 full-length (f.l.) protein do not appear to be diminished in these models. This seeming contradiction could potentially be explained by compensatory mechanisms. For example, Bri2 might facilitate the clearance of Trem2 through alternative pathways, such as lysosomes and autophagosomes, or it could influence the synthesis and maturation of Trem2 protein. While these additional

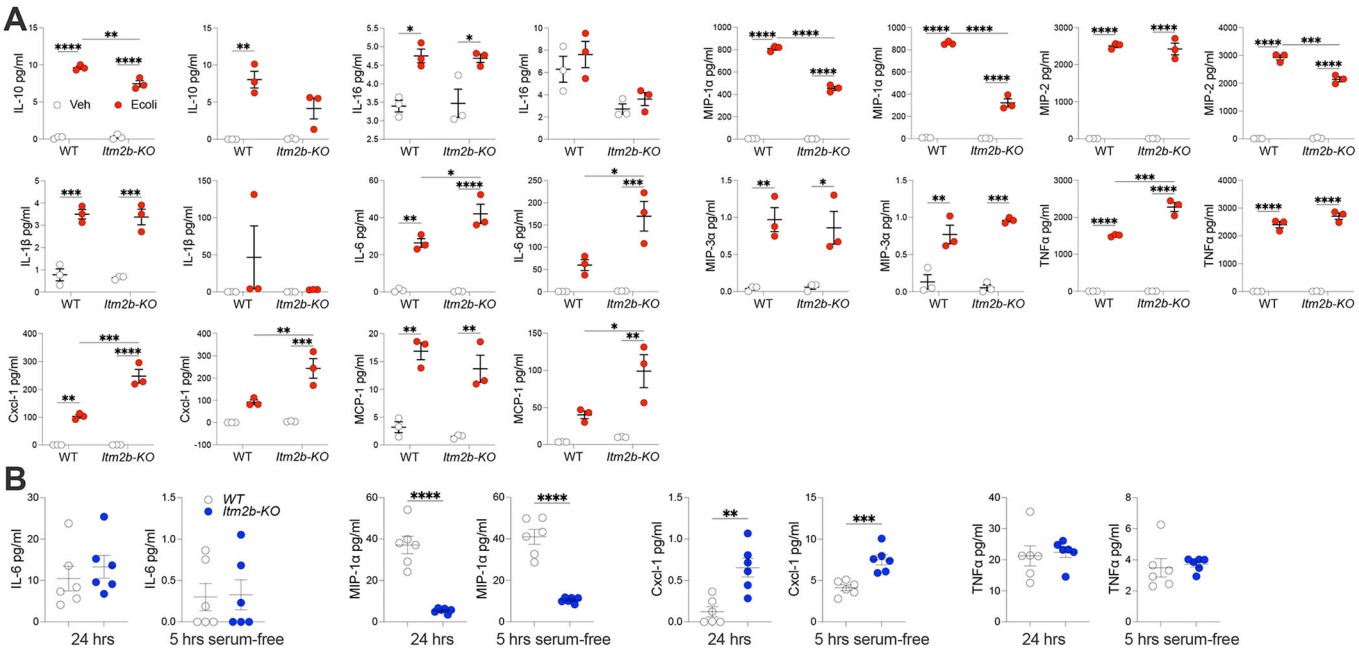

**Figure 14. Altered cytokine secretion profile in *Itm2b-KO* primary microglia.**

(A) Quantification of IL-10, IL-16, IL-1β, IL-6, Cxcl-1, MCP-1, MIP-1α, MIP-2, MIP-3, and TNFα, in the conditioned media of WT and *Itm2b-KO* primary microglia treated with *E. coli* for 2 h, along with controls treated with vehicle, Veh. (B) Quantification of IL-6, Cxcl-1, MIP-1α, and TNFα secreted by WT and *Itm2b-KO* primary microglia grown for 24 h in complete media and after 5 h of serum starvation. Data information: Statistical comparisons among the groups were conducted using either two-way ANOVA followed by post-hoc Sidak's multiple comparisons test when ANOVA indicated significant differences (A), or two-tailed unpaired t-test (B). *P < 0.05, **P < 0.01, ***P < 0.001, ****P < 0.0001. The data presented are derived from: (A) WT primary microglia cultures (n = 3 for Exp. 1 and n = 3 for Exp. 2) and *Itm2b-KO* primary microglia (n = 3 for Exp. 1 and n = 3 for Exp. 2); (B) WT primary microglia cultures (n = 6) and *Itm2b-KO* primary microglia (n = 6); the letter "n" indicates biological replicates, except for (B), which includes 3 biological replicates with 2 technical replicates each. Each biological replicate was composed of primary microglia generated from 2 P2 pups. All data are expressed as means +/− SEM. Source data are available online for this figure.

mechanisms are not explored in this study, they underscore the intricate interconnection between Trem2 and Bri2 in microglial biology.

The elevated levels of Trem2-CTF and sTrem2 in the CNS of FDD-KI mice provide evidence that pathogenic BRI2 mutations can impact the levels of TREM2 metabolites in the CNS. These changes may be attributed to neuroinflammatory effects resulting from this mutation, as seen in FDD patients. Neuroinflammation might indirectly affect Trem2-CTF and sTrem2 levels through various mechanisms, including alterations in microglia numbers and activation states, changes in the activity of secretases within microglia, fluctuations in Trem2 protein expression, and variations in the turnover rates of Trem2, Trem2-CTF, and sTrem2. Furthermore, this mutation might disrupt the functional interaction between BRI2 and TREM2, potentially due to the instability of the mutant BRI2 protein. Importantly, it should be noted that mature BRI2, which represents the functional form of BRI2, exhibits reduced levels in the CNS of both FDD-KI rodents (Tamayev et al, 2010b; Yin et al, 2021a, b) and FDD patients (Matsuda et al, 2011b; Tamayev et al, 2010a, b). These hypotheses are not mutually exclusive, and it is plausible that alterations in the BRI2-TREM2 functional interaction contribute to the neuroinflammation caused by the *ITM2B* Danish pathogenic mutation. Further studies are required to elucidate the relative impact of these mechanisms on the observed neuroinflammation in FDD patients.

The cleavage of TREM2 by α-secretase has divergent implications, involving the loss of function of the substrate TREM2 and a

potential gain of function through the creation of two novel TREM2 metabolites, TREM2-CTF and sTREM2. These metabolites may harbor distinct biological functions compared to the precursor protein. The evidence that sTREM2 levels are increased in the cerebrospinal fluid and CNS of AD patients suggest that the processing of TREM2 by α-secretase is linked to AD (Heslegrave et al, 2016; Lichtenthaler et al, 2022; Piccio et al, 2016; Suárez-Calvet et al, 2019). However, it is uncertain whether the elevation in sTREM2 levels observed in AD cases is a pathogenic factor contributing to the disease or a compensatory response aimed at mitigating an underlying pathogenic function. The evidence that the LOAD-associated TREM2 H157Y variant, located precisely at TREM2's ADAM10/17 cleavage site, enhances the shedding of sTREM2 implies that increased TREM2 α-processing could be a pathogenic mechanism (Suárez-Calvet et al, 2019). This might be attributed to the fact that this variant reduces TREM2 signaling and functionality by diminishing the presence of signaling-competent cell surface TREM2 (Schlepckow et al, 2017). In addition, it raises the possibility of pathogenic activities associated with increased functions of the metabolites sTREM2 and/or TREM2-CTF. On the contrary, recent findings suggests that microglia may increase the secretion of sTREM2 as a protective mechanism for neurons (Zhang et al, 2023). In this study, the authors show that sTREM2 plays a role in reducing tau phosphorylation, which is regarded as a pathogenic factor in AD. It achieves this by binding to transgelin-2 on the surface of

neurons, effectively deactivating the RhoA-ROCK-GSK3β kinase pathway, a critical tau kinase. Indeed, in the context of the widely accepted pathogenic AD model, where tau phosphorylation is considered a pathological factor, these findings initially appear contradictory to a pathogenic function of TREM2 shedding. However, while the idea that reducing tau phosphorylation is beneficial is widely accepted, it is not definitively proven. Furthermore, deactivating the RhoA-ROCK-GSK3β kinase pathway in neurons might not necessarily be advantageous and could potentially have detrimental effects on neuronal function unrelated to tau phosphorylation.

The pathogenic TREM2 p.H157Y mutation enhances shedding of mutant TREM2, and intriguingly, it causes a significant reduction in phagocytic capacity (Schlepckow et al, 2017). In parallel, our experiments with *Itm2b-KO* primary microglia demonstrate a similar reduction in phagocytic activity. While it is conceivable that Bri2 might influence phagocytosis through pathways unrelated to Trem2, the remarkable similarity in the phenotypes resulting from Bri2 deletion and the presence of a TREM2 variant associated with increased TREM2 shedding suggests that the phagocytosis deficit caused by Bri2 deletion may indeed be related to the concurrent increase in Trem2 processing.

We also provide evidence that the deletion of Bri2 can impact p38-MAPK activation in response to *E. coli* and alter the patterns of both "basal" and stimulated chemokine/cytokine secretion by microglia. While this study has not delved into the specific Trem2-dependent and/or Trem2-independent pathways responsible for these effects, the evidence strongly suggests that Bri2 plays a pivotal role in the functions of microglia.

In summary, this study highlights the significant role played by BRI2 in shaping microglial functions and emphasizes the need for further research to fully understand the mechanisms through which BRI2 influences microglial behavior and its potential implications for neurodegenerative diseases. BRI2's dual role in regulating both APP and TREM2 processing, coupled with its cell-autonomous impact on neurons and microglia, underscores its multifaceted involvement in the pathogenesis of these diseases. Given the absence of effective treatments for AD and related dementias, the pursuit of new therapeutic targets becomes paramount. The potential of BRI2 as such a target demands extensive exploration, and the development of drugs that modulate BRI2 function holds promise as a novel approach to addressing these devastating diseases.

# Methods

## Reagents and tools

See Table 3.

**Table 3. Reagents and tools.**

| Reagent/Resource | Reference or Source | Identifier or Catalog Number |
|---|---|---|
| **Experimental models** | | |
| Neuro-2A (N2A) cells | ATCC | CCL-131, RRID:CVCL_0470 |
| HEK293Tcells | ATCC | CRL-3216 |
| *Wild type (wt) mouse* | Jackson lab | C57BL/6 J (#000664), RRID:IMSR_JAX:000664 |
| *Itm2b-KO mouse* | D'Adamio Lab | https://doi.org/10.1523/JNEUROSCI.2094-08.2008 |
| *Trem2-KO mouse* | Jackson lab | C57BL/6J-Trem2em2Adiuj/J(#027197), RRID:IMSR_JAX:027197 |
| *Itm2b/Trem2-dKO mouse* | This paper | cross breeded from $Itm2b^{KO/KO}$ strain and $Trem2^{KO/KO}$ |
| *Itm2b$^{fl/fl}$ mouse* | Jackson lab | B6.129(Cg)-Itm2btm1Ldad/J(#018132), RRID:IMSR_JAX:018132 |
| *Cx3cr1$^{CreER/wt}$ mouse* | Jackson lab | B6.129P2(C)-Cx3cr1tm2.1(cre/ERT2)Jung/J (#020940), RRID:IMSR_JAX:020940 |
| *Itm2b$^{fl/fl}$Cx3cr1$^{CreER/wt}$ mouse* | This paper | cross breeded from Itm2b$^{fl/fl}$ strain and Cx3cr1$^{CreER}$ |
| FDD-KI mouse | D'Adamio Lab | https://doi.org/10.1371/journal.pone.0007900 |
| **Recombinant DNA** | | |
| rat Trem2-mia | D'Adamio Lab | https://doi.org/10.1038/s41598-020-60800-1 |
| F-BRI2 (F-BRI2$_{1-266}$) | D'Adamio Lab | https://doi.org/10.1002/emmm.201100195 |
| F-BRI2$_{1-243}$ | D'Adamio Lab | https://doi.org/10.1002/emmm.201100195 |
| F-BRI2$_{1-131}$ | D'Adamio Lab | https://doi.org/10.1002/emmm.201100195 |
| F-BRI2$_{1-117}$ | D'Adamio Lab | https://doi.org/10.1002/emmm.201100195 |
| F-BRI2$_{1-105}$ | D'Adamio Lab | https://doi.org/10.1002/emmm.201100195 |
| F-BRI2$_{1-93}$ | D'Adamio Lab | https://doi.org/10.1002/emmm.201100195 |
| F-BRI2$_{1-80}$ | D'Adamio Lab | https://doi.org/10.1371/journal.pone.0007900 |
| *3xflag-BRI2 + TREM2-delta ICD-myc* | This paper | https://en.vectorbuilder.com/vector/VB230921-1251ezd.html |
| *3xflag-BRI2 + TREM2-delta aCleavage-myc* | This paper | https://en.vectorbuilder.com/vector/VB230921-1146rwa.html |
| *3xflag-BRI2 + TREM2-myc* | This paper | https://en.vectorbuilder.com/vector/VB230721-1358umd.html |
| *3xflag-BRI2 + TREM2-CTF-myc* | This paper | https://en.vectorbuilder.com/vector/VB230921-1143ste.html |

**Table 3.** (continued)

| Reagent/Resource | Reference or Source | Identifier or Catalog Number |
| --- | --- | --- |
| *3xflag-BRI2 + TREM2-delta IG-myc* | This paper | https://en.vectorbuilder.com/vector/VB230921-1154vfk.html |
| *BRI2-1-131-myc + TREM2-3xflag* | This paper | https://en.vectorbuilder.com/vector/VB230921-1230kta.html |
| *BRI2-1-80-myc + TREM2-3xflag* | This paper | https://en.vectorbuilder.com/vector/VB230921-1234cfu.html |
| *BRI2-myc + TREM2-3xflag* | This paper | https://en.vectorbuilder.com/vector/VB230921-1229vse.html |
| *BRI2-delta80-131-myc + TREM2-3xflag* | This paper | https://en.vectorbuilder.com/vector/VB230921-1379umg.html |
| **Antibodies** | | |
| Trem2-CT | Cell Signaling Technology | 76765, RRID:AB_2799888 |
| Trem2-NT1 | Cell Signaling Technology | 61788, RRID:AB_2799615 |
| Trem2-NT2 | R&D system | AF1729, RRID:AB_354956 |
| Flag-M2 | Sigma | F3165, RRID:AB_259529 |
| anti-mouse HRP ab | Southern Biotech | 1031-05, RRID:AB_2794307 |
| anti-rabbit HRP ab-1 | Southern Biotech | OB405005, RRID:AB_2795955 |
| anti-rabbit HRP ab-2 | Cell Signaling Technology | 7074, RRID:AB_2099233 |
| biotinylated Trem2 Ab (ELISA) | R&D system | BAF1729, RRID:AB_356109 |
| Trem2 Ab (ELISA) | R&D system | MAB17291, RRID:AB_2208679 |
| sulfo-tag anti rat Ab (ELISA) | Meso Scale Diagnostics | R32AH |
| anti-Bri2 monoclonal ab | Cell Signaling Technology | provided by Richard W. Cho, https://doi.org/10.1016/j.jbc.2021.101089 |
| GAPDH | Sigma | g9545, RRID:AB_796208 |
| anti-sheep HRP | Novus | NBP1-73267, RRID:AB_11021350 |
| APC-CD11b ab | Miltenyi | 130-113-793, RRID:AB_2726317 |
| sulfo-tag anti rabbit Ab (ELISA) | Meso Scale Diagnostics | R32AB, RRID:AB_2892814 |
| *human Trem2-CT (D8I4C)* | Cell Signaling Technology | 91068, RRID:AB_2721119 |
| *human Trem2-NT (E6T1P)* | Cell Signaling Technology | 61788, RRID:AB_2799615 |
| *Myc-Tag (9B11) Mouse mAb (HRP Conjugate)* | Cell Signaling Technology | 2040, RRID:AB_2148465 |
| *CD45-APC/Vio770* | Miltenyi | 130-118-687 |
| *p38 MAPK (D13E1)* | Cell Signaling Technology | 8690, RRID:AB_10999090 |
| *Phospho-p38 MAPK (Thr180/Tyr182) (D3F9)* | Cell Signaling Technology | 4511, RRID:AB_2139682 |
| *Phospho-Zap-70 (Tyr319)/Syk (Tyr352) (65E4)* | Cell Signaling Technology | 2717, RRID:AB_2218658 |
| *Syk (D3Z1E) XP®* | Cell Signaling Technology | 13198, RRID:AB_2687924 |
| *phospho-PLCg1(Tyr783)* | Cell Signaling Technology | 14008, RRID:AB_2728690 |
| *PLCg1* | Cell Signaling Technology | 5690, RRID:AB_10691383 |
| *TACE (E8R8M) Rabbit mAb* | Cell Signaling Technology | 61048 |
| *Mouse Anti-rabbit IgG (Conformation Specific) (L27A9) mAb (HRP Conjugate)* | Cell Signaling Technology | 5127, RRID:AB_10892860 |
| **Oligonucleotides and other sequence-based reagents** | | |
| TaqMan™ Fast Advanced Master Mix | Thermo fisher | 4444556 |
| TaqMan™ gene expression probe(Itm2b) | Thermo fisher | Mm01310552_mH |
| TaqMan™ gene expression probe(Gapdh) | Thermo fisher | Mm99999915_g1 |
| TaqMan™ gene expression probe(Trem2) | Thermo fisher | Mm04209424_g1 |

**Table 3.** (continued)

| Reagent/Resource | Reference or Source | Identifier or Catalog Number |
|---|---|---|
| Forward primer for *Itm2b*^f | Mouse genotype Inc | CAGAGCTCCAGACACTGTTAG |
| Reverse primer for *Itm2b*^f | Mouse genotype Inc | GTCCAACCGGAACCACGTCACC |
| Forward primer for *Itm2b*^KO | Mouse genotype Inc | CAGAGCTCCAGACACTGTTAG |
| Reverse primer for *Itm2b*^KO | Mouse genotype Inc | AATTGTCTGCAGAATTGGCAAGAC |
| *TaqMan™ gene expression probe(ADAM10)* | Thermo fisher | Mm00545742_m1 |
| *TaqMan™ gene expression probe(ADAM17)* | Thermo fisher | Mm00456428_m1 |
| **Chemicals, enzymes and other reagents** | | |
| Fugene | Promega | E2311 |
| HALT protease and phosphatase inhibitor | Thermo fisher | 78444 |
| M2 cross-linked to agarose beads | Sigma | A2220 |
| 3X FLAG peptide | Sigma | F4799 |
| Protein A/G beads | Thermo fisher | 20421 |
| blocking solution | Thermo fisher | 37573 |
| Clarity Western ECL reagent | Bio-rad | 1705061 |
| Deglycosylation mix 2 | New England Biolabs | P6044s |
| Recombinant Mouse TREM2 Fc Chimera Protein (Elisa standard) | R&D system | 1729-T2-050 |
| MSD GOLD 96-well Small Spot Streptavidin SECTOR Plate | Meso Scale Diagnostics | L45SA |
| Fetal bovine serum | Gibco | A3840102 |
| Antibiotic- antimycotic | Gibco | 15240112 |
| EGTA | MP | 195173 |
| EDTA | Sigma | N6507 |
| β-mercaptoethanol | Invitrogen | NP0007 |
| 4–12% Bis–Tris polyacrylamide gel | Bio-rad | 3450125 |
| Propidum iodide | Invitrogen | p3566 |
| MSD read buffer | Meso Scale Diagnostics | R92TC |
| Proteinase K | Sigma | 70663 |
| *pHrodo RED E.coli* | Invitrogen | P35361 |
| *Beta-Amyloid (1–42) Aggregation Kit* | rPeptide | A-1170-02 |
| *pHrodoTM Red, Succinimidyl Ester* | ThermoFisher Scientific | P36600 |
| *ACK Lysing Buffer* | Gibco | A1049201 |
| *MEM* | Corning | 10-010-CV |
| *DMEM/F12* | Corning | 10-090-CV |
| *IMDM* | Corning | 10-016-CV |
| *r-MCSF* | Biolengend | 576404 |
| *soluble BRI2-BRICOS* | D'Adamio Lab | Genescript U706S742G0-4 |
| *soluble BRI2-ECD* | D'Adamio Lab | Genescript U706S742G0-9 |
| *soluble TREM2* | D'Adamio Lab | Genescript U706S742G0-14 |
| *soluble TREM2-ECD* | D'Adamio Lab | Genescript U706S742G0-19 |
| Adult Brain Dissociation Kit | Miltenyi | 130-107-677 |
| MACS Octo Dissociator | Miltenyi | 130-095-937 |
| Annexin V magnetic microbeads | Miltenyi | 130-090-201 |
| CD11b magnetic microbeads | Miltenyi | 130-049-601 |
| RNeasy RNA Isolation kit | Qiagen | 74104 |
| High-capacity cDNA RT Kit | Thermo fisher | 4368814 |
| Chromium Next GEM Single Cell 3' HT Reagent Kits v3.1 (Dual Index) | 10X Genomics | CG000416 |

**Table 3.** (continued)

| Reagent/Resource | Reference or Source | Identifier or Catalog Number |
|---|---|---|
| V-PLEX Plus Aβ Peptide Panel 1 6E10 | Meso Scale Diagnostics | k15200g |
| *U-PLEX Custom Biomarker Group 1 (ms) Assays, SECTOR* | Meso Scale Diagnostics | K15069M-2 |
| **Software** | | |
| LinRegPCR software | hartfaalcentrum.nl | version 2020.2 |
| Cell Ranger | 10X Genomics | RRID:SCR_017344, performed by Azenta life science |
| Seurat (v4.0) | satijalab | RRID:SCR_016341, https://doi.org/10.1016/j.cell.2021.04.048 |
| R (v4.1.0) | | RRID:SCR_001905 |
| Rstudio | Posit software | RRID:SCR_000432 |
| Image Lab software | Bio-rad | RRID:SCR_014210 |
| Graphpad Prism | Graphpad Inc | RRID:SCR_002798 |
| FACS DIVA 8.0.2 | BD | |
| Flowjo | Flowjo LLC | RRID:SCR_008520 |
| **Other** | | |
| Azenta life science service | Azenta life science | cDNA libiary sequencing service, 10x Genomics v3 procedure |
| BISC Inc RNAseq consulting service | BISC Inc. | consulting service on RNAseq analysis |
| ChemiDoc MP Imaging System | Bio-rad | |
| Chromium X | 10X Genomics | |
| Applied QuantStudio™ 6 Flex Real-Time PCR System | Thermo fisher | 4485691 |
| BD LSRII | BD | 405 nm, 488 nm and 633 nm laser |
| Aria Fusion Sorter | BD | 85-micron nozzle |
| meso QuickPlex SQ120 | Meso Scale Diagnostics | SQ120 |
| *Incucyte live imaging* | Sartorius | SX5 |

## Mice and ethics statement

All experiments were done according to policies on the care and use of laboratory animals of the Ethical Guidelines for Treatment of Laboratory Animals of the NIH. Relevant protocols were approved by the Rutgers Institutional Animal Care and Use Committee (IACUC) (Protocol #201702513). All efforts were made to minimize animal suffering and reduce the number of mice used. *Cx3cr1*^CreER/wt^ and *Trem2-KO* mice were purchased from The Jackson laboratory (Stock No. 020940 and 027197, respectively). Wild type, *Itm2b-KO* and *Itm2b-floxed* animals were generated by our laboratory.

## Adult microglia isolation

Mouse brains were extracted from 15-month-old mice (3 females and 3 males) after intracardiac PBS perfusion to remove blood from brain blood vessels. Brains were enzymatically and mechanically dissociated into a cell suspension using the Adult Brain Dissociation Kit (Miltenyi 130-107-677) and gentle MACS Octo Dissociator (Miltenyi 130-095-937). Apoptotic cells were removed using Annexin V magnetic microbeads (Miltenyi 130-090-201) and microglia were isolated using CD11b magnetic microbeads (Miltenyi 130-049-601) according to the manufacturer's instructions.

## Quantitative RT-PCR

Bound cells (microglia) and unbound cells (flow through, non-microglia fraction), or *WT* and *Itm2b-KO* primary microglia were analyzed by quantitative RT-PCR. Total RNA was extracted from isolated cells (microglia or non-microglia cells) with RNeasy RNA Isolation kit (Qiagen 74104) and used to generate cDNA with a High-Capacity cDNA Reverse Transcription Kit (Thermo 4368814). 10 ng of cDNA, TaqMan™ Fast Advanced Master Mix (Thermo 4444556), and the appropriate TaqMan probes were used in the real time polymerase chain reaction. Samples were analyzed on an Applied QuantStudio™ 6 Flex Real-Time PCR System, and relative RNA amounts were quantified using LinRegPCR software (hartfaalcentrum.nl). The probes Mm01310552_mH (exon junction 1–2) and Mm04209424_g1 (exon junction 3–4), Mm00545742_m1 (exon junction 14–15) and Mm00456428_m1 (exon junction 3-4), were used to detect mouse *Itm2b, Trem2, Adam10* and *Adam17*. Expression of *Itm2b* and *Trem2* were normalized to *Gapdh* levels, as detected with Mm99999915_g1 (exon junction 2–3) probe. To normalize the data, we calculated the ratio between the quantitative values of the target mRNA and those of *Gapdh* mRNA.

## ScRNAseq data generation and analysis

Microglia for scRNAseq analysis were prepared from 9–12-month-old mice in two independent experiments. Experiment-1 (Data 1)

included 1 male and 1 female WT (controls), 1 male and 1 female *Itm2b-KO*, 1 male and 1 female *Trem2-KO*, 1 male and 1 female *Itm2b/Trem2-dKO*. Experiment-2 (Data 2) included 1 female WT, 1 male and 1 female *Itm2b-KO*. Microglia were isolated as described above.

Purified microglia were used to generate Gel-bead-in-emulsion (GEMs) containing single cells using Chromium X; cDNA libraries were generated following Chromium Next GEM Single Cell 3' HT Reagent Kits v3.1 (Dual Index) instruction (10x Genomics). Sequencing was performed by Azenta life science following standard 10x Genomics v3 procedure. Raw data were pre-filtered through Cell Ranger Software. The pre-filter scRNAseq data were further analyzed using the most updated Seurat (v4.0) package in R (v4.1.0) (Hao et al, 2021) with assistance from BISC Inc. Sequencing data for experiments-1 and -2 were assembled into two Seurat objects (called Data-1 and Data-2, respectively), which were constructed using the CreateSeuratObject function. Clusters requirements were a minimum of 3 cells and 1000 features. For quality control, cells with more than 5% mitochondrial content and/or more than 45% ribosomal content were removed. Cells with low UMI and gene number per cell were filtered out. Cutoffs for UMI and gene number were empirically determined based on histograms showing cell density as a function of UMI per gene counts. Cutoffs of UMI greater than 300 and less than 50,000, and genes greater than 1000 and less than 6000 were applied to eliminate any potential subclusters formed solely due to the cells having insufficient features present to be accurately categorized while also eliminating potential doublets. We determined the dimensionality of each object to retain for downstream analysis in a heuristic manner, using an Elbow plot to establish the number of principal components (PCs) necessary to capture 95% of the variance in gene expression (Fig. EV5). We next constructed a K-nearest neighbors (KNN) graph based on the Euclidean distance in PCA space, refining the edge weights between any two cells based on the shared overlap in their local neighborhoods (a measure known as the Jaccard similarity) with the FindNeighbors() function (Hao et al, 2021). To cluster the cells, we then applied a modularity optimization technique (using the Louvain algorithm) implemented in the FindClusters() function (using a resolution of 0.4). Datasets were visualized in two dimensions using a uniform manifold approximation and projection (UMAP) dimensional reduction (https://doi.org/10.21105/joss.00861).

To specifically select microglia for further analysis, we performed cell type annotation using a single cell dataset published by Van Hove (Van Hove et al, 2019) as a reference. First, the Van Hove reference dataset was re-normalized using the SCTransform (v2) to be compatible with Data1 and Data2, before finding mutual nearest neighbor gene "anchors" between the reference and the Data1 and Data2 objects respectively using the FindTransferAnchors function. Next, we used this set of anchors to predict the identities of the Data1/Data2 cells. Cells predicted to be of the type "microglia" with > 95% confidence were retained for further analysis.

Next, we evaluated the effects of *Trem2* and/or *Itm2b* deletion on microglia's gene expression. However, as the samples were sequenced in two different experiments, the scRNA dataset integration functionality of the Seurat package was used to perform the joint analysis as previously described. This resulted in an object containing information on 297,215 cells. The top 3000 most highly

variable genes of each dataset were detected by the SelectInte-grationFeatures function to use as feature anchors in the PrepSCTIntegration function. The FindIntegrationAnchors function was then run on the SCT(v2) select sample datasets indicated above from Data1 and Data2 were integrated using the first 20 principal components into Object1.

Principal component analysis (PCA) was reperformed with PCA function before carrying out further sequencing batch correction normalization with Harmony(Korsunsky et al, 2019). Clustering of the integrated data was then executed using the top 30 harmonized PCA components to find neighbors and an SNN clustering resolution of 0.4. This resolution appeared to produce the most informative microglial clustering, but the microglia subtype investigation was generally robust to various choices of this hyperparameter.

## Kegg pathway analysis

Cluster 3 pathway analysis was performed using DEenrichRPlot function referenced with KEGG_2019_Mouse database.

## Cell culture, plasmids, transfection, immunoprecipitation, brain samples preparation, and Western blots analysis

Neuro-2A (N2A) cells (ATCC CCL-131) and HEK293Tcells (ATCC CRL-3216) were maintained in Eagle's Minimum Essential Medium (EMEM) (Gibco 11095-098) supplemented with 10% fetal bovine serum (Gibco A3840102) and Antibiotic-Antimycotic (Gibco 15240112). Plasmids used where described previously in the included references. Rat Trem2 (the Trem2-Miα isoform, UniProtKB - A0A6G8MV71); F-BRI2 (A.K.A. F-BRI2$_{1-266}$), F-BRI2$_{1-243}$, F-BRI2$_{1-131}$, F-BRI2$_{1-117}$, F-BRI2$_{1-105}$, F-BRI2$_{1-93}$ and F-BRI2$_{1-80}$ (Matsuda et al, 2005). Both HEK293T cells and N2A cells were transfected with indicated plasmids *via* Fugene (Promega, E2311) as previously described (Noviello et al, 2003; Scheinfeld et al, 2002b). Bicistronic plasmids were designed by D'Adamio's lab, and the construction of these plasmids was outsourced to VectorBuilder. The two coding regions were separated by an Internal Ribosomal Entry Site sequence. Detailed information about these constructs can be accessed as indicated in the Reagents and Tools Table.

Transfected cells were lysed in immunoprecipitation buffer (50 mm Tris, 150 mm NaCl, 1 mm EGTA (MP 195173), and 1 mm EDTA (pH 8.0) (Sigma E4378) with 0.5% NP40 (Sigma N6507) and HALT protease/phosphatase inhibitor (Thermo fisher 78444) solubilized for 30 min at 4 °C with rotating and spun at $20,000 \times g$ for 10 min. Solubilized cell lysate was used as input for immunoprecipitation. FLAG-BRI2 proteins were immunoprecipi-tated with anti-FLAG mouse monoclonal antibody M2 cross-linked to agarose beads (Sigma A2220). Immunoprecipitated samples were washed 5 times with immunoprecipitation buffer and proteins bound to M2-Agarose beads were eluted incubating the M2-Agarose beads with 3xFLAG peptide (Sigma F4799) at a 1 mg/ml concentration (370μM), for 30 min at room temperature. Trem2 was immunoprecipitated with either a Rabbit monoclonal antibody raised against the carboxy-terminus of mouse Trem2 (Cell Signaling Technology, 76765,) referred to as αTrem2-CT, a Rabbit monoclonal antibody raised against the amino-terminus of mouse

Trem2 (CST 61788) αTrem2-NT2, or the sheep polyclonal IgG raised against the amino-terminus of mouse Trem2 anti (R&D AF1729) αTrem2-NT1. Immunocomplexes were isolated with protein A/G beads (Thermo, 20421) and eluted 1× LDS sample buffer with 10% β-mercaptoethanol (Invitrogen; NP0007) at 55 °C. Input (total lysates, T.L.) and immunoprecipitation (I.P.) eluates were analyzed by Western blot.

Total brain lysates for the mice and rats were prepared as follows. Briefly, the mice were anesthetized with isoflurane and rapidly perfused via intracardiac catheterization with ice-cold PBS. Brains were extracted and homogenized with a glass-Teflon homogenizer in S1 buffer (250 mM sucrose, 20 mM Tris-base pH 7.4, 1 mM EDTA, 1 mM EGTA) supplemented with protease and phosphatase inhibitors. All steps were carried out on ice. The homogenates were further centrifuged at 4 °C at $100,000 \times g$ for 1 h to collect supernatants (the S100 fraction). The pellets were resuspended and solubilized with 1% NP-40 for 30-min and then centrifuged at $20,000 \times g$ for 10 min to collect solubilized supernatants (P100).

For Western blot analyses, total lysates proteins were diluted with PBS and LDS sample buffer—10% β-mercaptoethanol (Invitrogen; NP0007), separated on a 4–12% Bis–Tris polyacrylamide gel (Bio-Rad; 3450125), and transferred onto nitrocellulose at 25 V for 7 min using the Trans-blot Turbo system (Bio-Rad). Blotting efficiency was visualized by red Ponceau staining on membranes. Membranes were blocked in 5% milk (Bio-Rad; 1706404) for 45 min and washed in PBS/0.05% Tween-20. Primary antibodies were applied in blocking solution (Thermo; 37573). The following antibodies were used: anti-Flag M2 (Sigma F3165), anti-Trem2-CT, anti-Trem2-NT1, anti-Bri2 monoclonal antibody (provided by Richard W. Cho Cell Signaling Technology), anti-GAPDH (Sigma g9545). Secondary antibodies [either anti-mouse (Southern Biotech; 1031-05), anti-sheep (Novus NBP1-73267) or a 1:1 mix of anti-rabbit (Southern Biotech; OB405005) and anti-rabbit (Cell Signaling; 7074)] were diluted 1:1000 in 5% milk and used against either mouse or rabbit primary antibodies for 1 h at room temperature, with shaking. Membranes were washed with PBS/Tween-20 to 0.05% (three times, 10 min each time), developed with Clarity Western ECL reagent (Bio-rad 1705061) and visualized on a ChemiDoc MP Imaging System (Bio-Rad). Signal intensity was quantified with Image Lab software (Bio-Rad). Data were analyzed using Prism software (GraphPad Software, Inc) and represented as mean ± SEM.

### Recombinant BRI2-BRICHOS, BRI2-ECD, sTREM2, and TREM2-ECD

These proteins were generated as follows: Plasmids for these constructs were designed by D'Adamio's lab. The construction of these plasmids and the production of recombinant proteins were outsourced to GenScript. The plasmids were expressed in CHO-S cells and recombinant proteins were purified from the cell culture supernatants using Nickel-columns that interacted with the 7xHis tag. The specific proteins expressed included:

---

**BRI2-BRICHOS**

MGWSCIILFLVATATGVHS HHHHHHH DYKDHDGDYKDHDIDYKDDDDK ADSDPANIVHDFNKKLT AYLDLNLDK**C**YVIPLNTSIVMPPRNLLELLINIKAGTYLPQSYLIHEHMVITDRIENIDHLGFFIYRL**C**H DKETYKLQRRETIKGIQKREASNCFAIRHFENKFAVETLICS

**BRI2-ECD**

MGWSCIILFLVATATGVHS HHHHHHH DYKDHDGDYKDHDIDYKDDDDK LYKYFALQPDDVYYCGI KYIKDDVILNEPSADAPAALYQTIEENIKIFEEEEVEFISVPVPEFADSDPANIVHDFNKKLTAYLDLN LDK**C**YVIPLNTSIVMPPRNLLELLINIKAGTYLPQSYLIHEHMVITDRIENIDHLGFFIYRL**C**HDKETYK LQRRETIKGIQKREASNCFAIRHFENKFAVETLICs

The 7 His in yellow are the His tag for purification, the sequence in red is the 3XFLAG epitope for IPs. The sequence in green represents the signal peptide inserted at the N-Terminus to allow for secretion.

---

**sTREM2**

MEPLRLLILLFVTELSGA HNTTVFQGVAGQSLQVSCPYDSMKHWGRRKAWCRQLGEKGPCQRV VSTHNLWLLSFLRRWNGSTAITDDTLGGTLTITLRNLQPHDAGLYQCQSLHGSEADTLRKVLVEVL ADPLDHRDAGDLWFPGESESFEDAHVE HHHHHHH

**TREM2-ECD**

MEPLRLLILLFVTELSGA HNTTVFQGVAGQSLQVSCPYDSMKHWGRRKAWCRQLGEKGPCQRV VSTHNLWLLSFLRRWNGSTAITDDTLGGTLTITLRNLQPHDAGLYQCQSLHGSEADTLRKVLVEVL ADPLDHRDAGDLWFPGESESFEDAHVE HSISRSLLEGEIPFPPTS HHHHHHH

The 7 His in yellow at the C-terminus are the His tag for purification, the sequence in green is the endogenous signal peptide at the N-Terminus.

---

## Deglycosylation prior to Western blot analysis

Cell lysate or total microglia were solubilized with 1% NP-40 for 30 min rotating, spun at $20,000 \times g$ and the supernatant was used as input for deglycosylation reactions, according to the manufacturer's specifications (NEB P6044S). Conditioned media media were deglycosylated directly, with no prior solubilization step.

## Primary mouse microglia isolation

WT and *Itm2b-KO* primary microglia cultures were obtained from neonatal pups of both sexes (Daniele et al, 2014). Briefly, postnatal day 1–2 mice were euthanized, and their brains were extracted, with the meninges removed. Cortices were micro-dissected and minced in HBSS medium supplemented with 100 µg/mL penicillin/streptomycin (P/S) and 0.01 M HEPES. The minced tissue was then mixed with 3 ml of microglia complete media (MEM supplemented with 1 mM L-glutamine, 1 mM sodium pyruvate, 0.6% d-(+)-glucose, 100 µg/mL P/S, 4% FBS, 6% horse serum) and centrifuged at $1000 \times g$ for 5 min at room temperature. The resulting cell pellet was resuspended in complete medium and cultured in T-75 flasks in a humidified tissue culture incubator at 37 °C with 5% $CO_2$. After 24 h, the media were replaced with 13 ml of fresh microglia complete media. On days in vitro (DIV) 15–20, flasks were placed on a rotary shaker for 4 h to detach primary microglia. Microglia were collected by centrifugation at $1000 \times g$ for 5 min and resuspended in growth media (MEM supplemented with 1 mM L-glutamine, 1 mM sodium pyruvate, 0.6% d-(+)-glucose, 100 µg/mL P/S, 5% FBS). Live microglia were counted using the Denovix Celldrop automated cell counter with AO/PI dye solution. 150,000 live microglia were seeded per well in a 12-well plate for biochemical and molecular experiments, while 30,000 live microglia were plated per well in a 96-well plate specifically for phagocytosis experiments. Each biological replicate utilized a combination of primary microglia harvested from two mice of the same genotype. This approach was adopted to ensure a sufficient supply of primary microglia for each biological replicate required to conduct the experiments effectively.

## Primary bone marrow-derived macrophage isolation

WT and *Itm2b-KO* mice were euthanized, and their tibia and femurs were isolated and kept in PBS on ice. After two washes with 70% EtOH and an additional rinse with PBS, bone marrow was isolated from cut tibia and femurs by centrifugation at $15,000 \times g$ for 15 s. The bone marrow was then resuspended in 80 µl of macrophage complete media (IMDM supplemented with 10% heat inactivated FBS, 100 µg/mL P/S) and transferred to 5 ml of ACK Lysing Buffer for 5 min at room temperature. Cells were collected after centrifugation at $200 \times g$ for 5 min and resuspended in 5 ml of complete media. The suspension was then passed through a 100 µm nylon filter with an additional 10 ml of complete media rinse. Live macrophages were counted using the Denovix Celldrop automated cell counter with AO/PI dye solution. Approximately 6 million live cells were plated in each 10 cm dish in macrophage differentiation media (macrophage complete media with 20 ng/ml r-MCSF). On DIV 3, 70% of the media was replaced with fresh differentiation media. On DIV 7–14, primary macrophages were collected using a cell scraper and counted using the Denovix Celldrop automated cell counter with AO/PI dye solution.

## Co-immunoprecipitation of Trem2/Bri2 from primary macrophages

Primary macrophages were lysed in immunoprecipitation buffer containing 0.5% NP40 and HALT protease/phosphatase inhibitor. The lysates were solubilized for 30 min at 4 °C with continuous rotation and then centrifuged at $20,000 \times g$ for 10 min. The resulting solubilized cell lysate was used as input for immunoprecipitation. Immunoprecipitation was performed by incubating the lysate with αTrem2-CT antibody and protein A/G agarose beads. Following incubation, immunoprecipitated samples were washed five times with immunoprecipitation buffer and subsequently eluted using 1× LDS sample buffer containing 10% β-mercaptoethanol at 55 °C. The input (total lysates, T.L.) and immunoprecipitation (I.P.) eluates were then subjected to analysis by Western blot after deglycosylation.

## Primary microglia activation assays

150,000 live primary microglia were plated per well in a 12-well plate for subsequent experiments. Twenty-four hours after plating, primary microglia were starved with DMEM/F12 for 5 h and subsequently incubated with pHrodo Red-labeled *E. coli* (10 µg/ml) or vehicle only. Cell culture supernatants were harvested after the 24 h initial incubation, after the 5 h starvation, and 1 h and 2 h after stimulation for sTrem2 and cytokines/chemokines measurements. Cell lysates were prepared after 2 h stimulation and were used to determine levels of Trem2, Trem2-CTF, Adam17, of Syk, pSyk, Plcγ1, pPlcγ1, p38, and p-p38.

## Primary microglia phagocytosis assay

30,000 live primary microglia were plated per well in a 96-well plate for subsequent experiments. Twenty-four hours after plating, primary microglia were starved with DMEM/F12 for 5 h and subsequently incubated with pHrodo Red-labeled *E. coli* (10 µg/ml). Treated cells were incubated in the IncuCyte system allowing real-time measurement of the fluorescent signal.

## Flow cytometry

Cells were dissociated from adult mouse brains as described above and prepared in FACS buffer (PBS w/w 2% BSA + 1 mM EDTA). The cells were stained with APC-CD11b antibody (Miltenyi 130-113-793) and CD45 antibody (Miltenyi 130-118-687) for 30 min with three subsequent washes. Propidium iodide (1%, Invitrogen p3566) was added to eliminate dead cells from analysis. Cells were acquired using a LSRII (BD Bioscience) and DIVA 8.0.2 software, and data were analyzed using FlowJo software.

## Cell sorting

Cells were dissociated from adult mouse brains as described above. Propidium iodide (1%, Invitrogen p3566) was added to a single-cell suspension before sorting EYFP⁺ and EYFP⁻ cells using a FACS Aria Fusion Sorter (BD Bioscience). To verify sorting efficiency, sorted cells were acquired using a LSRII (BD Bioscience) and DIVA 8.0.2 software, and data were analyzed using FlowJo software.

## ELISAs

Aβ levels were assessed using the Meso Scale Discovery kit V-PLEX Plus Aβ Peptide Panel 1 (K15200G), following the manufacturer's recommendations with a modification for detecting rodent Aβ. For detection, we substituted sulfo-tag 6E10, a mouse monoclonal antibody targeting the human Aβ1–16 sequence, with sulfo tag M3.2, a mouse monoclonal antibody recognizing the rat/mouse Aβ1–16 sequence (Biolegend 11465). The plate readings were conducted using the MESO QuickPlex SQ 120.

The ELISAs for analysis of full length Trem2 and sTrem2 were modified from Kleinberger's protocol (Kleinberger et al, 2014). Briefly, a streptavidin-coated plate (Meso Scale Discovery, L15SA) was blocked with 3% BSA/PBST (0.05% Tween-20) overnight at 4 °C. The plates were incubated with 0.25 µg/ml biotinylated sheep anti-mouse TREM2 antibody (R&D Systems, BAF1729) for 1 h at RT with shaking. After washing four times with PBST, the samples and standards (Trem2, R&D Systems, 1729-T2-050) were incubated for 2 h at RT with shaking. The plates were washed three times with PBST and incubated with either 1 µg/ml of rabbit monoclonal anti-mouse TREM2 (Cell Signaling Technology, 76765) for full length Trem2 ELISA (ELISA 1), or with 1 µg/ml rat monoclonal anti-mouse/human TREM2 (R&D Systems, MAB17291) for sTrem2 ELISA (ELISA 2) for 1 h at RT. After four washes in PBST, 0.5 µg/ml SULFO-TAG labeled antibody (Meso Scale Discovery, anti-rabbit R32AB for ELISA 1; anti-rat R32AH for ELISA 2) was added and incubated for 1 h with shaking. The plates were washed three times in PBST, developed in Meso Scale Discovery read buffer (Meso Scale Discovery, R92TC), and read on a MESO QuickPlex SQ 120.

The ELISA for analysis of secreted cytokines and chemokines was performed with the U-PLEX Custom Biomarker Group 1 (ms) Assays, SECTOR, provided Meso Scale Discovery. IL-10, IL-16, IL-1β, IL-6, Cxcl-1, MCP-1, MIP-1α, MIP-2, MIP-3, and TNFα were selected in this paper.

## Genomic DNA isolation and PCR analysis

FACS sorted EYFP$^+$ cells were incubated in 300 µl of lysis buffer (100 mM Tris, 5 mM EDTA, 0.2% SDS, 200 mM NaCl, PH 8.0) with 60 µg/ml of protease K (Sigma 70663) at 55° C for 2 h. 100 µl of protein precipitation solution (7.5 M Ammonium Acetate) was added and vortex for 30 s before centrifugation at $15,000 \times g$ for 5 min. The supernatant was transferred into a new Eppendorf tube with 300 µl Isopropanol and mixed by inverting 30 times. The samples were subsequently centrifuged at $15,000 \times g$ for 5 min and washed with 70% EtOH. Dried tube with genomic DNA were resuspend in 100 µl of water for PCR analysis. Primer pairs were: Forward primer-CAGAGCTCCAGACACTGTTAG, Reverse primer-GTCCAACCGGAACCACGTCACC to amplify the $Itm2b^f$ allele (804 bp PCR product); Forward primer-CAGAGCTCCAGA-CACTGTTAG, Reverse primer-AATTGTCTGCAGAATTGG CAAGAC to amplify the $Itm2b^{KO}$ allele (155 bp PCR product). The PCR was performed as follows: denaturation at 94 °C for 3 min, followed by 34 cycles of denaturation at 94 °C for 30 s, annealing at 58 °C for 30 s, extension at 72 °C for 40 s, followed by a final "filling" extension at 72 °C for 2 min. PCRs were performed by Mouse genotype Inc.

## Statistical analysis

Data were analyzed using GraphPad Prism software and expressed as mean ± SEM. Statistical tests used to evaluate significance and statistical data are shown in figures. Significant differences were accepted at $P < 0.05$. Sample size estimation was not performed for this study. In the case of experiments involving primary microglia cultures, the sample size was determined based on the number of neonatal pups and the yield of primary microglia obtained from each litter. Experiments were conducted when a substantial number of primary microglia cultures from distinct pups were available, ensuring a minimum of three biological replicates for each genotype/condition. It's important to note that the researchers involved in the experiments were not blinded to the conditions.

# Data availability

The datasets and materials used and/or analyzed during the current study are available from the corresponding author on reasonable request. The scRNAseq data are deposited at Gene Expression Omnibus (GEO), https://www.ncbi.nlm.nih.gov/geo/query/acc.cgi? acc=GSE233601, to allow public access once the data are published. The following secure token has been created to allow review of record GSE233601 while it remains in private status: ilsnuawkpdmlfkr.

# Peer review information

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

## Acknowledgements

This work was supported by National Institute on Aging (To LD: RO1AG033007 and R01AG073182). We thank Ian Nackman for technical support.

## Author contributions

**Tao Yin**: Conceptualization; Data curation; Formal analysis; Validation; Investigation; Visualization; Methodology; Writing—review and editing. **Metin Yesiltepe**: Data curation; Formal analysis; Validation; Investigation; Visualization; Methodology; Writing—review and editing. **Luciano D'Adamio**: Conceptualization; Resources; Data curation; Formal analysis; Supervision; Funding acquisition; Validation; Investigation; Visualization; Methodology; Writing—original draft; Project administration; Writing—review and editing.

## Disclosure and competing interests statement

The authors declare no competing interests.

# Expanded View Figures

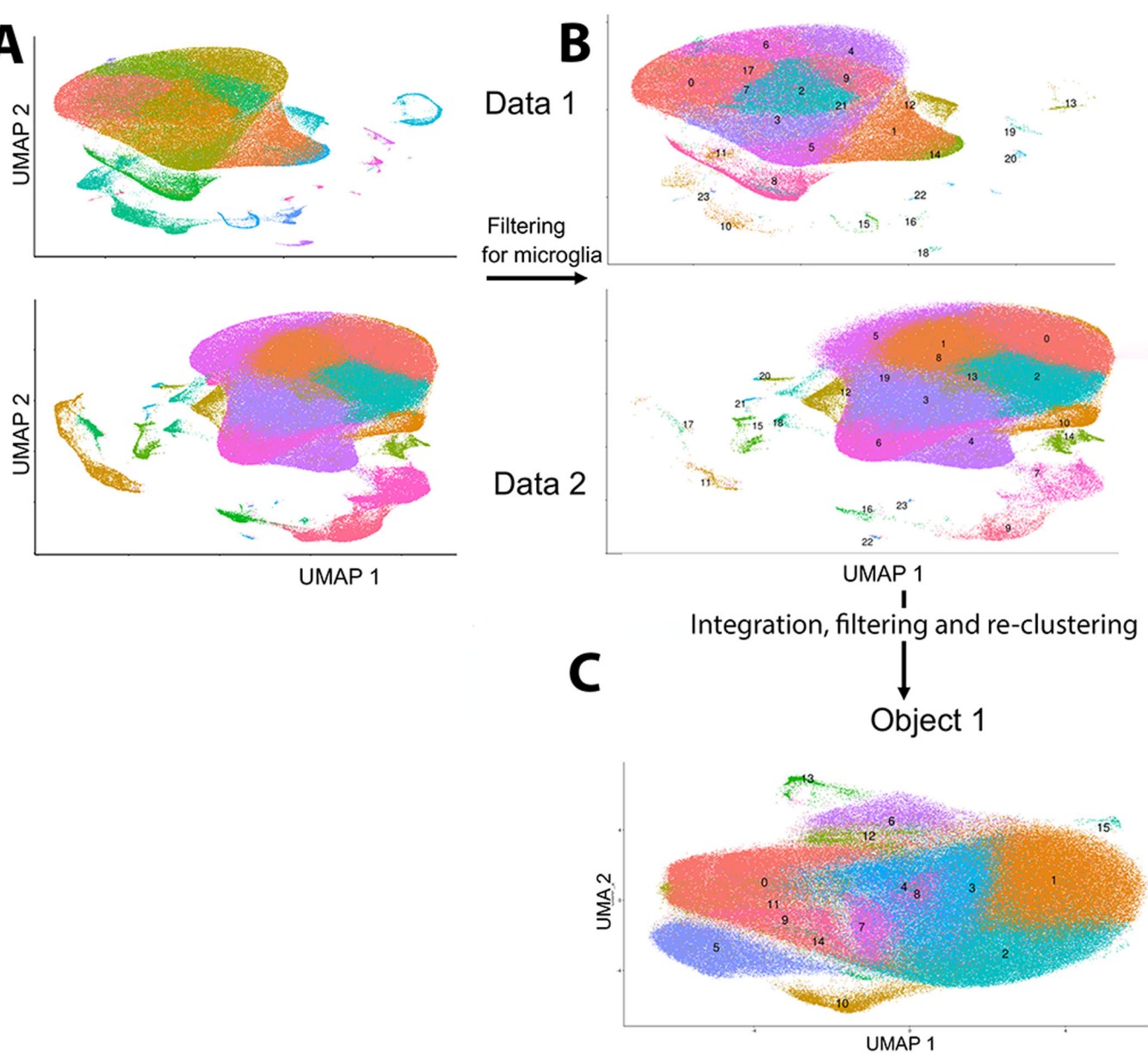

**Figure EV1.  UMAP of Data 1 and Data 2.**

(A) UMAPs of objects Data 1 and Data 2 before filtering. (B) UMAPs of objects Data 1 and Data 2 after filtering. (C) UMAP of Object 1 after integration, filtering, and re-clustering, shows 16 microglia clusters. Data information: The data presented in this analysis are the result of two experiments, namely Data 1 and Data 2. To combine specific sample datasets from both Data 1 and Data 2, we employed the integration feature within the Seurat package. By utilizing the first 20 principal components, we integrated these datasets into a single entity referred to as "Object1," which encapsulates information from a total of 297,215 cells. These cells derive from: *Trem2-KO*, 1 male and 1 female; *Itm2b-KO*, 2 males and 2 females; WT controls, 1 male and 2 females; *Itm2b/Trem2-dKO*, 1 male and 1 female. The scRNAseq data are deposited at https://www.ncbi.nlm.nih.gov/geo/info/seq.html, GSE233601 to allow public access once the data are published.

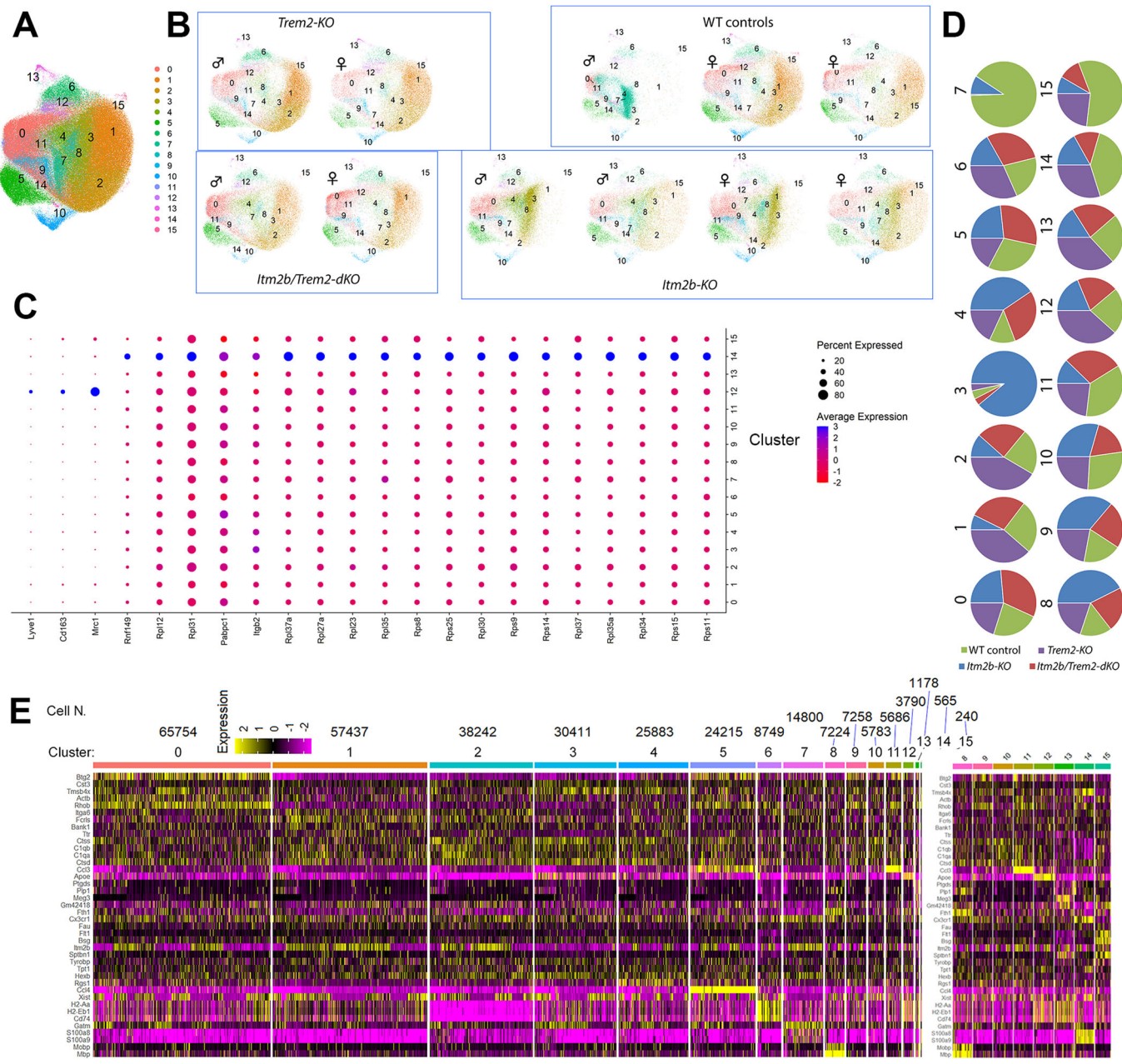

**Figure EV2. Object 1 supporting information.**

(A) UMAPs of re-clustered microglia in Object 1. (B) UMAPs split by individual samples. (C) Average scaled expression levels of selected signature genes per cluster and cluster's annotation based on expression of signature genes. (D) Proportional contribution of each genotype to each cluster. Cluster 3 was highly represented in *Itm2b-KO* mice, with 89% of microglia in this cluster originating from these mice. Conversely, Cluster 7 was preponderant in WT controls. However, ~93% of the cells assigned to cluster 7 derived from one WT control animal (the male WT control, as depicted in UMAP plot b). Therefore, the observed expansion of cluster 7 is attributed to animal-specific factors rather than genotype-specific factors. (E) Gene expression heatmap showing the top 5 enriched genes for each microglia cluster. The number of cells per cluster is denoted above the cluster. *Itm2b* is one of the top genes downregulated in cluster 3 because 89% of the cells in this cluster are from *Itm2b-KO* mice. Enlarged heatmap of Clusters 8 to 15 is also show (right) for better visibility. Data information: The data presented in this analysis are deposited at https://www.ncbi.nlm.nih.gov/geo/info/seq.html, GSE233601 to allow public access once the data are published.

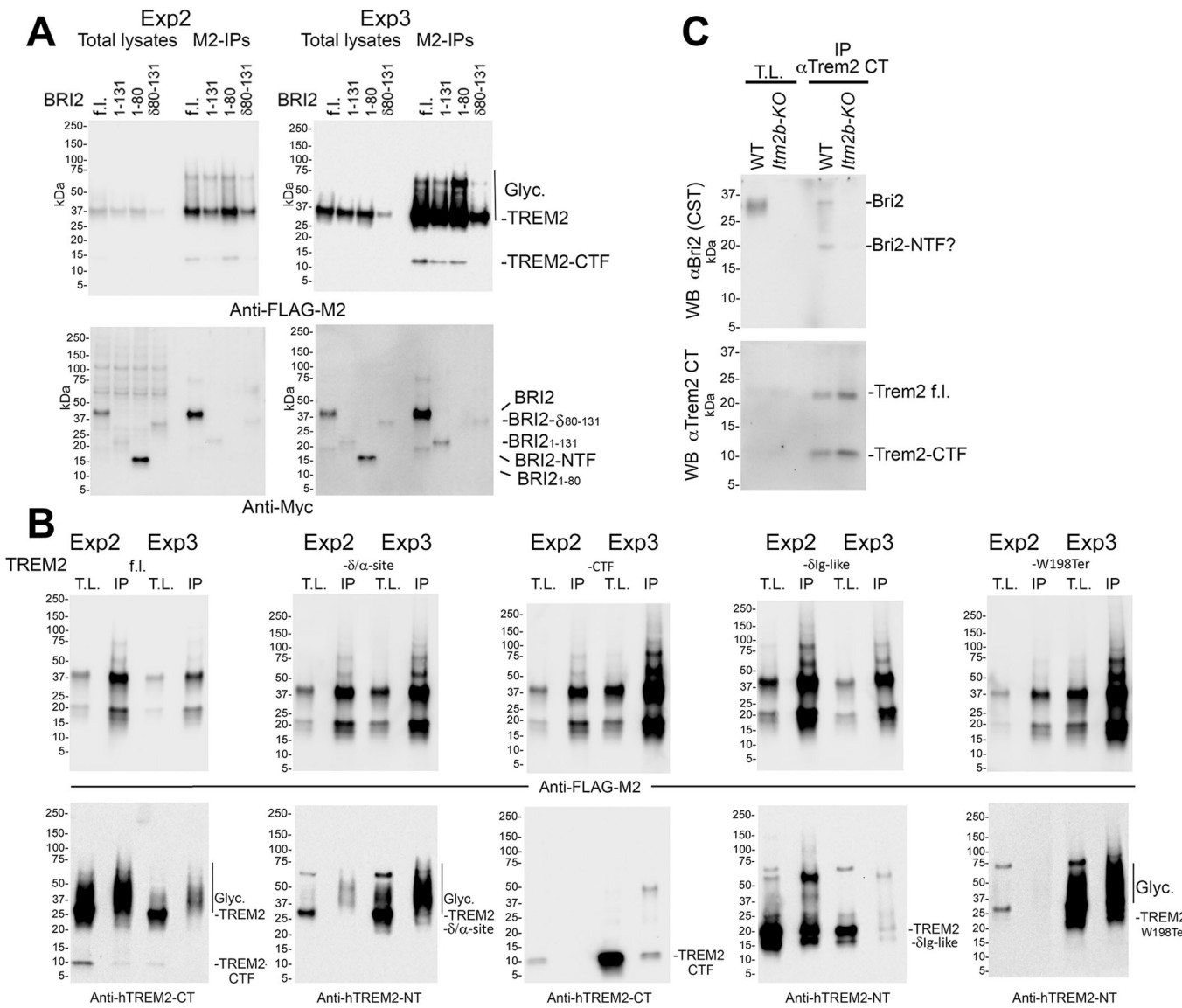

**Figure EV3. Interaction between human BRI2 and TREM2 in transfected cells and co-immunoprecipitation of endogenous Bri2 and Trem2 in mouse primary macrophages.**

(A) Western blot analysis with anti-FLAG, anti-Myc antibodies, and anti-human BRI2 antibodies of total lysates and immunoprecipitated samples (IP-M2) from transfected HEK293 cells. These experiments are biological replicates of the experiment shown in Fig. 4B. (B) Western blot analysis with anti-FLAG, anti-human TREM2-NT, and anti-human TREM2-CT antibodies of total lysates (T.L.) and immunoprecipitated samples (IP-M2) from transfected HEK293 cells. These experiments are biological replicates of the experiment shown in Fig. 4D. (C) Co-immunoprecipitation of endogenous Bri2 and Trem2 from mouse primary macrophages. Samples were deglycosylated before Western blot. Data Information: Panels (A) and (B) represent two independent experiments conducted similarly to those in Figs. 4B and 4D, respectively. Panel (C) shows the only co-immunoprecipitation of endogenous Bri2 and Trem2 performed to date. The complete membrane images used for Western blot analyses are included without any cropping of information above or below the targeted signals.

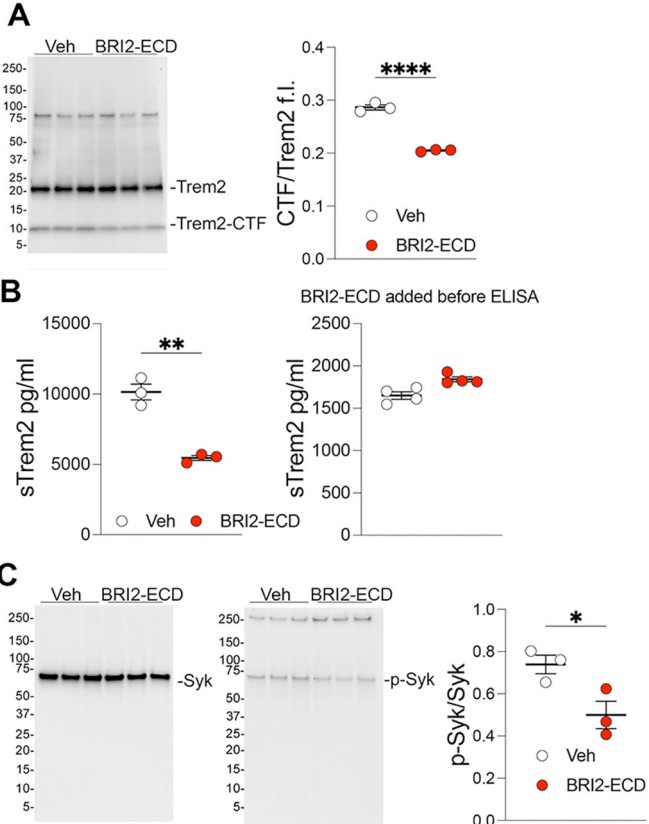

**Figure EV4. Evidence of the effect BRI2-ECD on Trem2 processing and signaling.**

(A) Western blot analysis with Trem2 CT antibody of deglycosylated cell lysates from *Itm2b-KO* microglia treated with either vehicle (PBS) or a 2 µM concentration of BRI2-ECD. Quantification of the Trem2-CTF/Trem2 f.l. ratios from the Western blot shown in the right panel. (B) sTrem2 ELISA on conditioned media from these cell cultures (left panel). The right panel shows an ELISA performed using media from cells treated with vehicle and incubated before and during the ELISA with either vehicle (PBS) or 2 µM of BRI2-ECD. The evidence that incubation with BRI2-ECD does not change the ELISA quantification indicates that BRI2-ECD does not interfere with the quantification of sTrem2 by ELISA. (C) Western blot analysis with anti-Syk and anti-pSyk antibodies of cell lysates from *Itm2b-KO* microglia treated with either vehicle (PBS) or a 2 µM concentration of BRI2-ECD. Quantification of the pSyk/Syk ratios from the Western blot is shown in the right panel. Data information: This figure encompasses the comprehensive dataset employed for these specific experiments. We have included the images of the complete membranes used for Western blot analyses, without any cropping of information above or below the targeted signals. Statistical comparisons among the groups were conducted using a two-tailed unpaired *t*-test. *$P < 0.05$, **$P < 0.01$, ****$P < 0.0001$. The data presented are derived from *Itm2b-KO* primary microglia ($n = 3$ for each condition); the letter "n" indicates biological replicates. Each biological replicate was composed of primary microglia generated from 2 P2 pups. All data are expressed as means $+/-$ SEM. Source data are available online for this figure.

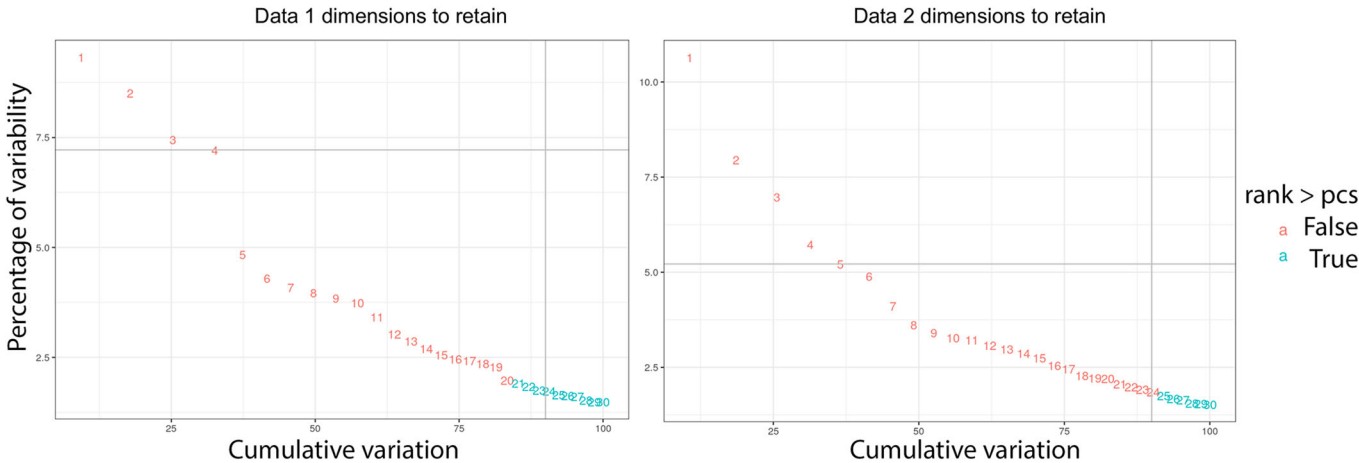

**Figure EV5.** Heuristic establishment of the number of dimensions/principal components (PCs) needed to capture 90% of the variance in gene expression from the datasets.

