## [Peer Review File · EMBO Reports]

Functional BRI2-TREM2 interactions in microglia: implications for Alzheimer's and related dementias

Tao Yin, Metin Yesiltepe, and Luciano D'Adamio

Corresponding author(s): Luciano D'Adamio (luciano.dadamio@rutgers.edu) , Tao Yin (ty183@gsbs.rutgers.edu)

Review Timeline:

Submission Date:	13th Jul 23
Editorial Decision:	5th Sep 23
Revision Received:	30th Nov 23
Editorial Decision:	16th Jan 24
Revision Received:	17th Jan 24
Accepted:	19th Jan 24

Editor: Martina Rembold / Esther Schnapp

Transaction Report:

Dear Dr. D'Adamio

Thank you for the submission of your research manuscript to our journal. As my colleague Esther Schnapp is currently traveling, I have temporarily taken over the handling of your manuscript.

We have now received the full set of referee reports that is copied below. As you will see, the referees acknowledge that the findings are potentially interesting, but they also raise a number of concerns and have suggestions on how to strengthen your study that seem all pertinent and need to be addressed.

Given these constructive comments, we would like to invite you to revise your manuscript with the understanding that the referee concerns (as detailed above and in their reports) must be fully addressed and their suggestions taken on board. Please address all referee concerns in a complete point-by-point response. Acceptance of the manuscript will depend on a positive outcome of a second round of review. It is EMBO Reports policy to allow a single round of revision only and acceptance or rejection of the manuscript will therefore depend on the completeness of your responses included in the next, final version of the manuscript.

We realize that it is difficult to revise to a specific deadline. In the interest of protecting the conceptual advance provided by the work, we recommend a revision within 3 months (December 5th). Please discuss the revision progress ahead of this time with the editor if you require more time to complete the revisions.

I am also happy to discuss the revision further via e-mail or a video call, if you wish.

*****IMPORTANT NOTE:

We perform an initial quality control of all revised manuscripts before re-review. Your manuscript will FAIL this control and the handling will be delayed IF the following APPLIES:

- 1) A data availability section providing access to data deposited in public databases is missing. If you have not deposited any data, please add a sentence to the data availability section that explains that.
- 2) Your manuscript contains statistics and error bars based on $n=2$. Please use scatter blots in these cases. No statistics should be calculated if $n=2$.

When submitting your revised manuscript, please carefully review the instructions that follow below. Failure to include requested items will delay the evaluation of your revision.*****

- 1) a .docx formatted version of the manuscript text (including legends for main figures, EV figures and tables). Please make sure that the changes are highlighted to be clearly visible.
- 2) individual production quality figure files as .eps, .tif, .jpg (one file per figure). Please download our Figure Preparation Guidelines (figure preparation pdf) from our Author Guidelines pages <https://www.embopress.org/page/journal/14693178/authorguide> for more info on how to prepare your figures.
- 3) a .docx formatted letter INCLUDING the reviewers' reports and your detailed point-by-point responses to their comments. As part of the EMBO Press transparent editorial process, the point-by-point response is part of the Review Process File (RPF), which will be published alongside your paper.
- 4) a complete author checklist, which you can download from our author guidelines (<<https://www.embopress.org/page/journal/14693178/authorguide>>). Please insert information in the checklist that is also reflected in the manuscript. The completed author checklist will also be part of the RPF.
- 5) Please note that all corresponding authors are required to supply an ORCID ID for their name upon submission of a revised manuscript (<<https://orcid.org/>>). Please find instructions on how to link your ORCID ID to your account in our manuscript tracking system in our Author guidelines (<<https://www.embopress.org/page/journal/14693178/authorguide#authorshipguidelines>>)
- 6) We replaced Supplementary Information with Expanded View (EV) Figures and Tables that are collapsible/expandable online. A maximum of 5 EV Figures can be typeset. EV Figures should be cited as "Figure EV1, Figure EV2" etc... in the text and their respective legends should be included in the main text after the legends of regular figures.

7) Data Availability section: Please provide a reviewer password if the datasets are not yet public.

Additional information on source data and instruction on how to label the files are available <<https://www.embopress.org/page/journal/14693178/authorguide#sourcedata>>.

10) Figure legends and data quantification:
The following points must be specified in each figure legend:

- the name of the statistical test used to generate error bars and P values,
 - the number (n) of independent experiments (please specify technical or biological replicates) underlying each data point,
 - the nature of the bars and error bars (s.d., s.e.m.)
- If the data are obtained from n {less than or equal to} 5, show the individual data points in addition to the SD or SEM.
- If the data are obtained from n {less than or equal to} 2, use scatter blots showing the individual data points.

See also the guidelines for figure legend preparation:
<https://www.embopress.org/page/journal/14693178/authorguide#figureformat>

11) Our journal encourages inclusion of *data citations in the reference list* to directly cite datasets that were re-used and obtained from public databases. Data citations in the article text are distinct from normal bibliographical citations and should directly link to the database records from which the data can be accessed. In the main text, data citations are formatted as follows: "Data ref: Smith et al, 2001" or "Data ref: NCBI Sequence Read Archive PRJNA342805, 2017". In the Reference list, data citations must be labeled with "[DATASET]". A data reference must provide the database name, accession number/identifiers and a resolvable link to the landing page from which the data can be accessed at the end of the reference. Further instructions are available at <<https://www.embopress.org/page/journal/14693178/authorguide#referencesformat>>.

12) As part of the EMBO publication's Transparent Editorial Process, EMBO Reports publishes online a Review Process File to accompany accepted manuscripts. This File will be published in conjunction with your paper and will include the referee reports, your point-by-point response and all pertinent correspondence relating to the manuscript.

Yours sincerely,

Referee #1:

In this study D'Adamio and colleagues show that BRI2 (encoded by the ITM2B gene), a protein mutated in cases of Familial non-Alzheimer Dementia, physically interacts with TREM2 and inhibits its cleavage by α -secretase. The authors follow up with in vivo experiments showing that *Itm2b* total KO mice or ablation of *Itm2b* specifically in microglia have increased soluble TREM2 levels in CNS.

The results are novel and interesting; the experiments are for the most part convincing and properly executed. However, there are some issues that require further discussion or improvement.

Major:

1) In Fig.2 the authors focus their attention on cluster 3, which appears de novo in *Itm2b*-KO. However, it seems that cluster 7 is partially reduced in *Itm2b*-KO and totally gone in *Trem2*-KO and *Itm2b* \times *Trem2*-KO. In addition, cluster 1, 2 and 3 seem also to diminish in *Itm2b*-KO. These differences, if real, should be reported and carefully discussed. The authors should re-analyze the data reducing the number of clusters (16 clusters for microglia only seems a little too excessive) and make sure that they have eliminated "wetting" artefacts in their scRNAseq dataset, a problem that might generate non-existing "cell" clusters. They should also show Volcano plots of genes differentially expressed by altered clusters.

2) In Fig.5E the authors show levels of Ab40 and Ab42. This is confusing. Ab40 and Ab42 generated in this context (if any) should be of mouse origin. However, in the Material and Methods the authors mention ELISA for human Ab peptides. Is the ELISA cross-reactive? If so, the authors should state this in the text.

3) In Fig.6A they should add CD45 to CD11b, to distinguish microglia from BAMs.

Minor:

1) page 5 of results line 16: "which suggest a reduction in *Trem2* processing" should be probably an increase in *Trem2* processing?

2) In references: Yin et al. JBC is quoted twice.

Referee #2:

Overall this is an interesting and timely manuscript, my concerns and suggestions for improvement are outlined below:

General: The manuscript under review lacks explicit elucidation regarding the normalization of mRNA levels with respect to GAPDH. This crucial methodological detail remains conspicuously absent from both the main body of the text and the accompanying figure legend. Furthermore, the specific analytical approach employed for Quantitative RT-PCR, whether it be the delta delta Ct method, Efficiency-Corrected Delta Ct (E Δ Ct), Linear Models, Generalized Linear Models (GLMs), GeNorm, or NormFinder, is regrettably omitted from the presented work. Clarification and inclusion of these pertinent aspects would undoubtedly enhance the scientific rigor and comprehensiveness of the study.

Detailed criticism:

Regarding Figure 1A (right panel), it is advisable to consider repositioning the labels currently situated at the uppermost part of the graph. A suggested improvement entails relocating these labels within an adjacent box, akin to the layout demonstrated in Figure 1A and 1B (left panel). This adjustment would likely contribute to the visual clarity and alignment consistency across the depicted panels.

Itm2b modulates microglial transcriptome in a *Trem2*-dependent manner:

2 independent experiments are done with un-equal groups:

Data 1:

1 male and 1 female WT (controls)

1 male and 1 female Itm2b-KO
1 male and 1 female Trem2- KO
1 male and 1 female Itm2b/Trem2-dKO
Data 2:
1 female WT
1 male and 1 female Itm2b-KO

It is worth noting that the utilization of two separate datasets, namely Dataset 1 and Dataset 2, introduces a potential source of bias stemming from variations in biological variability. Specifically, the presence of dissimilar group sizes across the datasets may lead to incongruities in variability. Should the smaller groups within one dataset exhibit heightened variability compared to the larger groups within the other dataset, the integrity of your conclusions may be compromised.

Figure 2b:

In the initial assessment, it is recommended that special emphasis be placed on cluster number 6 to enhance its visibility and facilitate improved referencing. A simple alteration, such as modifying the color of the cluster number, should effectively achieve this goal. Additionally, it is advised to replace the term "high expression," currently utilized to delineate differential expression, with the more precise descriptors of "downregulation" or "up-regulation." This adjustment will contribute to the precision and clarity of the terminology employed.

Furthermore, a key clarification is warranted regarding the specific genotype to which the depicted graph pertains. Precisely identifying whether the graph corresponds to the WT control, Itm2b-KO, or Trem2-KO genotype is imperative for contextual understanding and accurate interpretation.

In the text on page 6, the paragraph spanning from "We next" to "genotype-specific factors" articulates three concepts:

1. A significant majority of cells in cluster 3 exhibit Itm2b-KO (89%).
2. Itm2b registers as one of the most profoundly down-regulated genes within cluster 3.
3. Among all four Itm2b-KO mice, there exists an ample presence of microglia assigned to cluster 3.

The composition of this paragraph raises concerns regarding potential "circular reasoning" wherein the ideas presented appear to reinforce one another in a circular manner.

Concerning Figure 3A, an elucidation is warranted to account for the observed weak signal of TREM2. Given the current blotting data, this attenuated signal intensity might suggest a scenario wherein the extent of TREM2 engagement with BRI2 is relatively limited or minimal. Such an interpretation would contribute to a clearer understanding of the presented results and their potential implications.

Turning attention to Figure 3C, it is noteworthy that the significance or implications of the asterisk symbol (*) remain unexplained within both the main text and the figure's legend. Additionally, the meaning behind the designation "(f.l)" lacks clarification.

In relation to Figure 3D, a concise clarification regarding the identity of the bands falling within the 100-150 kDa range under the NT2 condition would greatly enhance the figure's clarity and contextual understanding.

Directing attention to Figure 4A, it is pertinent to note that the significance or interpretation associated with the asterisk symbol (*) remains undisclosed within both the main text and the figure's legend.

Regarding Figure 5C, it is evident that the signal intensity of the Trem2 blot is notably weak. This observation should be acknowledged and addressed, possibly by providing an explanation for the diminished signal strength.

With regards to Figure 5D, while a discernible distinction is observable within the female group, the same differentiation is less apparent within the male group. This observation merits acknowledgement and consideration, particularly in terms of potential underlying factors that may account for the disparity in results between the male and female

cohorts.

Concerning Figure 5E, the observation of substantially reduced CNS A β 40 and A β 42 levels in Trem2-KO mice indeed presents an intriguing contrast. This phenomenon appears counterintuitive given the established understanding that loss of TREM2 function is associated with compromised microglial uptake of A β and lipoproteins. The text acknowledges this potential discrepancy by noting, "This finding may appear contradictory to the notion that TREM2 mediates A β clearance." However, a more comprehensive analysis or discussion of this apparent contradiction, along with potential explanations or alternative mechanisms, would add depth to the interpretation of the results and enrich the scientific discourse.

The discussion presents compelling evidence for the interaction between BRI2 and TREM2, especially with respect to their involvement in microglial function. Consideration of potential downstream effects and regulatory mechanisms resulting from this interaction could enhance the depth of analysis. The discussion adequately delves into the expansion of this cluster in *Itm2b*-KO mice, but further interpretation of the functional significance of this expansion, particularly in the context of AD pathology, would be valuable. The potential roles of Trem2 processing and its modulation by BRI2 in shaping microglial phenotypes could be discussed more extensively. In consonance, while the paper discusses the implications of altered Trem2 processing in the context of FDD and AD, it would benefit from an exploration of potential downstream consequences and how these alterations may contribute to disease progression. Connecting the findings to Figure 7 A,B & C:

BRI2 regulation of TREM2 processing in microglia: implications for Alzheimer's and related dementias (Tao Yin* & Luciano D'Adamio*) 37

microglial dysfunction and neuroinflammation, which are central to AD pathogenesis, would strengthen the discussion's disease-related implications.

Referee #3:

Yin and D'Adamio analyzed the role of *Itm2b*/BRI2 in microglia, and the proteolytic processing of Trem2. RNA expression analysis showed elevated expression of *Itm2b* in microglia as compared to other brain cell types. scRNAseq analysis showed that deletion of the *Itm2b* gene in mice led to enrichment of a specific microglial cluster (cluster 3) that was not induced in Trem2 ko mice or *Itm2b*/Trem2 dko mice, suggesting that *Itm2b* dependent regulation of microglia might involve Trem2. Authors further show co-IP of BRI2 and Trem2 in transiently transfected cells, indicating interaction of both proteins. Co-IP studies with deletion constructs indicated a role of the Brichos domain and the APP interacting domain of BRI2 in the interaction with Trem2. Quantification of Trem2 processing products sTrem2 and CTFs in transfected cells and mouse brains indicated BRI2 dependent proteolytic processing of full-length Trem2 by alpha-secretase in microglia. Overall, the findings are potentially interesting, and support a functional role of *Itm2b*/BRI2 in the regulation of microglia. However, the potential roles of BRI2 in Trem2 processing and the underlying mechanisms are less clear. Additional experiments and controls could help to strengthen the conclusion of the authors that BRI2 regulates microglial function by altering the proteolytic processing of Trem2.

Specific comments:

1. Co-IP experiments were performed only with transient overexpression cell models (Fig. 3). Demonstration of co-IP from cells that endogenously express BRI2 and Trem2 (i.e. primary microglia) could support the conclusion on BRI2-Trem2 interaction. Authors should explain the bands migrating above 100 kDa in the NT2 IP samples (fig. 3D). In this panel (upper blot), an F-BRI2-NTF is indicated, but hardly visible on the blot, and apparently does not co-IP with Trem2. Authors should comment on the length of this fragment, and whether it would contain the putative APP interacting domain.
2. The data in Fig. 3F indicate that the C-terminal half of BRI2 that contains the Brichos domain is mainly responsible for the interaction with Trem2. Thus, it should be tested whether the soluble ectodomain of BRI2 that is generated by alpha-secretase also interacts with Trem2 and affects the proteolytic processing.
3. It is stated that the deletion constructs 1-131, 1-117, 1-105 bind with similar efficiency. However, the expression level of the different BRI2 constructs seem to vary (on the provided blot, Fig. 3F). Quantitative analysis could help to support the conclusion of the authors.
4. The Trem2 CTF co-precipitates with BRI2 (Fig.3C), raising the question which domains of Trem2 are involved in the interaction. Using deletion constructs of Trem2 could help to map interacting domains of Trem2 (as done for BRI2). Do disease-associated variants of Trem2 affect the interaction with BRI2? As co-IP approaches from cell lysates do not rule out the involvement of additional proteins/factors, a direct interaction of the ectodomains of BRI2 and Trem2 could be tested in cell free systems with recombinant proteins.
5. BRI2 also represents a protein substrate of alpha-secretase, and thus, it is possible that the overexpression of BRI2 (Fig. 4) could affect Trem2 processing by substrate competition. Overexpression of the Brichos domain could help to clarify whether the observed effects could result from direct interaction with Trem2 or substrate competition or both.

6. It should also be tested if BRI2 affects the subcellular transport and localization of Trem2. Cytochemical detection of both proteins could be done to address the (co)localization of both proteins. Does BRI2 interact with immature and mature Trem2?
 7. The bands labeled as sTREM2 should be explained in more detail (Fig. 4A and C). Migration is indicated as ~15 kDa. This does not fit very well to the mw of mature full length Trem2 (about 30-50 kDa). Is immature trem2 secreted under these conditions?
 8. In transient expression systems (fig. 3 and 4) overexpression of BRI2 stabilizes full-length Trem2. However, levels of Trem2 were not altered by the deletion of *Itm2b* in vivo (Fig. 5B), and authors speculate about a potential compensatory mechanisms. What could be an explanation?
 9. Levels of Trem2 obtained by ELISA1 are given as A.U., that obtained by ELISA2 in pg. Authors should also provide data from ELISA1 in pg. Can authors detect full-length Trem2 and sTrem2 by western blotting (as done for the Trem2 CTF, fig. 5c)? To facilitate detection, samples could be deglycosylated (as done in fig. 3). It would be possible to determine the ratios of full-length Trem2 to sTREM2 and CTF in individual samples and further assess precursor-product relationships from in vivo samples. Such an approach could also be used to analyze the metabolism of Trem2 in primary microglia from mouse models generated in this study (microglia-specific deletion of *Itm2b*).
 10. It is mentioned that the FDD mouse model exhibits neuroinflammation. As Trem2 expression could be upregulated during neuroinflammation, authors should detect mRNA levels of Trem2 and also detect full-length Trem2 in order to assess precursor-product ratios (see above).
 11. Does the deletion or overexpression of *Itm2b* affect expression and overall activity of alpha-secretases (ADAM10/17)?
 12. It would also be important to test whether Trem2-specific signaling (e.g. phosphorylation of Dap12 and/or Syk) and microglial function (e.g. phagocytosis, cytokine secretion) is affected by the deletion *Itm2b*. Primary microglia cultures could be used for these experiments.
- Overall, while the study convincingly demonstrates a role of BRI2 in microglia, the proposed mechanism on Trem2 and functional implications should be analyzed in more detail to support the conclusion of the authors

Additional points:

- Authors should provide more information on the number of experiments and number of replicate samples for the individual data sets (e.g. Fig. 3-7)
- Heading of the last paragraph of the results section "FDD-KI mice show elevated CNS levels of Trem2-CTF and sTrem2 in mice increases CNS sTrem2 levels." reads confusing.

Reference list contains duplications

We thank the reviewers for the insightful comments. We have diligently addressed all questions to the best of our abilities, considering the constraints of available manpower and financial resources. Due to the extensive experiments conducted and the new insights gained, we have made substantial revisions to the manuscript. These revisions include changes to the title, abstract, results, discussion, and materials and methods sections. All modifications have been clearly indicated using red characters. Additionally, we have modified some of the original figures and introduced new figures and EV figures. This has necessitated a reordering and renumbering of some of the original figures. As a result of these changes, the new figures in the revised manuscript are as follows:

- 1) Figure 1 has been revised in accordance with the recommendations of Reviewer 2.
- 2) Figure 2 has been enhanced to include Volcano plots, addressing the recommendations of Reviewer 1.
- 3) In Figure 3, the schematic representations of TREM2 and BRI2 have been updated to provide clearer and more accurate depictions.
- 4) Figure 4 is a new addition to the manuscript, aimed at providing a more precise definition of the regions within TREM2 and BRI2 that mediate their interaction, in response to the query raised by Reviewer 3.
- 5) Figure 5 is also a new inclusion, demonstrating the direct interaction between the ectodomains of TREM2 and BRI2, in a cell-free system. This addresses the specific request made by Reviewer 3.
- 6) Figure 6 (formerly Figure 4 in the original submission) contains a new panel E in response to the query raised by Reviewer 3.
- 7) Figure 7 (formerly Figure 5 in the original submission) remains unchanged.
- 8) Figure 8 (formerly Figure 6 in the original submission) has been updated as per the suggestions of Reviewer 1. It now includes CD11b/CD45 staining, as requested. In addition, this figure includes an additional experiment (E) illustrating the reduced *Trem2* mRNA expression in microglia following *in vivo Itm2b* mRNA knockdown. Former panel E is now panel F.
- 9) Figure 9 (formerly Figure 7 in the original submission) remains unchanged.
- 10) Figures 10, 11, 12, 13 and 14 present a series of novel experiments conducted in both Wild-Type and *Itm2b*-KO primary microglia. These figures collectively aim to address various other requests made by reviewer 3.
 - Figure 10 delves into how Bri2 influences Trem2 processing, both under "basal" conditions and in response to microglial stimulation with *E. coli*. It also provides insights

suggesting that Bri2 may impact *Trem2* mRNA expression and the regulation of Trem2 protein via α -secretase-independent pathways.

- Figure 11 offers evidence that do not support the hypothesis that Bri2 deletion causes heightened α -secretase activity.
- Figure 12 analyzes the effects of Bri2 deletion on *E. coli*-induced intracellular signaling.
- Figures 13 and 14 investigate the role of Bri2 in *E. coli* phagocytosis and in the secretion of cytokines and chemokines, both under basal conditions and after *E. coli* stimulation.

11) Figure EV1 remains unchanged.

12) Figure EV2 has been updated to improve the presentation of UMAPs based on sex and genotype, addressing the suggestions from Reviewer 2. Moreover, we included additional data for improved cluster annotation, addressing the recommendations of Reviewer 1.

13) Figure EV3 is new and includes data supporting Figure 5 (A and B). In addition, Figure EV3C provides supplementary evidence supporting the interaction between endogenous Trem2 and Bri2. This figure, while important, is presented in the supplementary materials because: a) the experiment was conducted in macrophages, not microglia; b) additional controls, such as a similar experiment using *Trem2-KO* cells or an immunoprecipitation with an isotype-negative control, would be beneficial to establish the specificity of the interaction behind any reasonable doubt. Currently, we lack a functional anti-mouse Bri2 antibody to perform these additional controls.

14) Figure EV4 is new and is included to showcase a potential effect of BRI2-ECD on Trem2 processing and signaling in *Itm2b-KO* cells. Although this experiment was conducted using three biological replicates derived from six *Itm2b-KO* pups, it is presented as supplementary material due to the following reasons: a) recombinant BRI2-ECD is found in monomers, dimers and oligomers, warranting further investigation; b) to reinforce the data, we intend to replicate this experiment and to assess the effect of BRI2-ECD on WT microglia, as it may function as a dominant negative inhibitor; d) it would be advantageous to examine the effects of BRI2-BRICHOS and ideally, BRI2-ectodomain recombinant proteins that do not bind the TREM2 ectodomain for a comprehensive understanding of their interactions/function relationship.

15) Figure EV5 (formerly Figure EV3 in the original submission) remains unchanged.

Below is a point by point response to the reviewers' comments/criticism/suggestions.

Referee #1:

Major:

1) A) In Fig.2 the authors focus their attention on cluster 3, which appears de novo in *Itm2b*-KO. However, it seems that cluster 7 is partially reduced in *Itm2b*-KO and totally gone in *Trem2*-KO and *Itm2b*X*Trem2*-KO. In addition, cluster 1, 2 and 3 seem also to diminish in *Itm2b*-KO. These differences, if real, should be reported and carefully discussed.

Response: We appreciate the reviewer's comments and would like to clarify our approach to analyzing the scRNAseq data. As mentioned in Figure EV2, Cluster 7 was preponderant in WT controls but was mainly derived from a single WT control animal. Thus, the observed expansion of cluster 7 can be attributed to animal-specific factors rather than genotype-specific factors, as indicated in the UMAP plot in Figure EV2.

Regarding the other clusters, we have included in Figure EV2 additional data to improve our clusters annotation. Cluster 12 has been reannotated as a BAM cluster, cluster 14 as a ribosome biogenesis cluster.

The reviewer is correct that there are clusters where representation is affected by the genotype, as highlighted in the new Table1. We have excluded Cluster 7 for the reasons explained earlier. Based on this evidence, we have annotated clusters 1 as I/T-D1, 2 as I/T-D2, 3 as I/T-D3, 8 as I/T-D4, and 4 as *Itm2b*-D. However, as mentioned in the paper, we did not extensively use the scRNAseq data to characterize the types of microglia resident in the CNS of mice with different genotypes. This is because evidence published after we conducted the scRNAseq experiments (which were performed several years ago) has shown that the enzymatic dissociation method used to isolate microglia from the brain can activate microglia and alter their typology from when they were resident in the CNS. However, despite this technical limitation, the observation that Cluster 3 is predominant in microglia isolated from *Itm2b*-KO mice is still significant. It suggests that microglia from *Itm2b*-KO mice may respond differently to stimulations, even in an ex vivo context. Given this, we did not emphasize the significance of other clusters and instead chose to focus on the potential importance of Cluster 3, indicating a functional interaction between *Bri2* and *Trem2*. Therefore, we used this systems biology approach as a foundation for a molecular biology approach to investigate this interaction. We believe that the data presented in this manuscript justify this approach. However, we acknowledge that we passed over these other potentially interesting and important genotype-dependent changes in microglia clustering. These other changes are reported in the revised manuscript to provide a more comprehensive discussion of these findings.

B) The authors should re-analyze the data reducing the number of clusters (16 clusters for microglia only seems a little too excessive) and make sure that they have eliminated "wetting" artefacts in their scRNAseq dataset, a problem that might generate non-existing "cell" clusters.

Response: We appreciate the reviewer's suggestions, but we believe that reanalysis is unnecessary. First, we clustered the cells based on recommended principles

(https://satijalab.org/seurat/articles/pbm3k_tutorial.html), by applying a modularity optimization

technique (using the default Louvain algorithm) to iteratively group cells together. We implemented this procedure using the `FindClusters()` function, which contains a resolution parameter that determines the 'granularity' of the downstream clustering, with larger values leading to a greater number of clusters. Typically, setting this parameter between 0.4 - 1.2 returns good results for single-cell datasets of around 3000 cells with the optimal resolution often increasing for larger datasets. Object1 contains 363,653 cells (over 100 times larger) so our chosen resolution of 0.4 could be viewed as conservative. Additionally, we conducted a clustering tree analysis to demonstrate that most clusters are consistent across clustering resolutions ranging from 0.2 - 0.4, so lowering the resolution does not meaningfully alter the conclusions generated from our analysis. Clusters 3 and 4 at resolution 0.4 (blue) (which are up-regulated in *Itm2b-KO* mice) can be traced to a single cluster (cluster 2) at resolution 0.2 (red) but are also split at resolution 0.3 (green), before being more evenly distributed at

resolution 0.4. Thus, it seems reasonable to consider clusters 3 and 4 as distinct. Other primary clusters of interest (clusters 6, 10, and 12) are consistent across resolutions as depicted in the clustering tree.

Moreover, after reanalysis, cluster 12 has been identified as a BAMs cluster. Therefore, we now have a total of 15 microglia clusters.

Finally, the number of clusters in our study does not appear to be unusually high when considering the substantial number of microglia cells we analyzed. For reference, in a recent study published in *Cell* (Volume 186, Issue 20, 28 September 2023, Pages 4386-4403.e29), the authors analyzed 152,469 microglia nuclei and identified 13 microglia clusters. In another study also published in *Cell* (Volume 185, Issue 22, 27 October 2022, Pages 4153-4169.e19), the authors found 11 clusters when analyzing 89,615 single cells.

C) They should also show Volcano plots of genes differentially expressed by altered clusters.

Response: Thank you for the suggestion. Volcano plots of clusters I/T-D1, I/T-D2, - I/T-D3 and I/T-D4 are shown in Figure 2C.

2) In Fig.5E the authors show levels of Ab40 and Ab42. This is confusing. Ab40 and Ab42 generated in this context (if any) should be of mouse origin. However, in the Material and Methods the authors mention ELISA for human Ab peptides. Is the ELISA cross-reactive? If so, the authors should state this in the text.

Response: Thank you for noticing this mistake. The mistake has been corrected both in the result section as well as the Material and Methods section. We used a modified version of the Meso Scale Discovery A β ELISA kit, which has been adapted to detect rodent A β . The specificity of this modified Kit was previously reported in: Pham H, Yin T, D'Adamio L (2022) Initial assessment of the spatial learning, reversal, and sequencing task capabilities of knock-in rats with humanizing mutations in the A β -coding region of *App*. *PLoS One* 17: e0263546.

3) In Fig.6A they should add CD45 to CD11b, to distinguish microglia from BAMs.

Response: We have conducted the experiment and incorporated a new panel in Figure 8A (previously 6A). This panel confirms that EYP+ cells are CD11b+ and CD45^{low}, indicating their identity as microglia rather than BAMs. It is worth noting that the CD11b intensity may appear slightly lower than in the previous experiment. This difference is likely due to the use of the same aliquot of antibody, with the experiments being conducted approximately one year apart.

Minor:

1) page 5 of results line 16: "which suggest a reduction in Trem2 processing" should be probably an increase in Trem2 processing?

Response: Thank you for noticing this mistake. The mistake has been corrected.

2) In references: Yin et al. JBC is quoted twice.

Response: Thank you for noticing this mistake. The reviewer is correct. The mistake has been corrected.

Referee #2:

General: The manuscript under review lacks explicit elucidation regarding the normalization of mRNA levels with respect to GAPDH. This crucial methodological detail remains conspicuously absent from both the main body of the text and the accompanying figure legend. Furthermore, the specific analytical approach employed for Quantitative RTPCR, whether it be the delta delta Ct method, Efficiency-Corrected Delta Ct (E Δ Ct), Linear Models, Generalized Linear Models (GLMs), GeNorm, or NormFinder, is regrettably omitted from the presented work. Clarification and inclusion of these pertinent aspects would undoubtedly enhance the scientific rigor and comprehensiveness of the study.

Response: As detailed in the Materials and Methods section, we utilized LinRegPCR software (hartfaalcentrum.nl) for quantitative RTPCR analysis. To normalize the data, we calculated the ratio between the quantitative values of the target mRNA and those of *GAPDH* mRNA.

Detailed criticism:

Regarding Figure 1A (right panel), it is advisable to consider repositioning the labels currently situated at the uppermost part of the graph. A suggested improvement entails relocating these labels within an adjacent box, akin to the layout demonstrated in Figure 1A and 1B (left panel). This adjustment would likely contribute to the visual clarity and alignment consistency across the depicted panels.

Response: Thank you for the suggestion. The Figure has been modified accordingly.

Itm2b modulates microglial transcriptome in a Trem2-dependent manner:

2 independent experiments are done with un-equal groups:

Data 1:

1 male and 1 female WT (controls)

1 male and 1 female Itm2b-KO

1 male and 1 female Trem2- KO

1 male and 1 female Itm2b/Trem2-dKO

Data 2:

1 female WT

1 male and 1 female Itm2b-KO.

It is worth noting that the utilization of two separate datasets, namely Dataset 1 and Dataset 2, introduces a potential source of bias stemming from variations in biological variability.

Specifically, the presence of dissimilar group sizes across the datasets may lead to incongruities in variability. Should the smaller groups within one dataset exhibit heightened variability compared to the larger groups within the other dataset, the integrity of your conclusions may be compromised.

Response: As detailed in the Materials and Methods section, we conducted a joint analysis to account for potential bias originating from the use of separate datasets by utilizing a set of methods designed for scRNA-seq integration as described in Stuart et al, 2019:

<https://doi.org/10.1016/j.cell.2019.05.031>. In general, this technique serves to align shared cell populations across datasets. The integration process initially pinpoints pairs of cells across datasets that are in a corresponding biological state, termed 'anchors'. These anchors are instrumental in adjusting for technical discrepancies between datasets (batch effect correction) and facilitate comparative scRNA-seq analysis under various experimental conditions.

Figure 2b: In the initial assessment, it is recommended that special emphasis be placed on cluster number 6 to enhance its visibility and facilitate improved referencing. A simple alteration, such as modifying the color of the cluster number, should effectively achieve this goal.

Response: Is the reviewer referring to cluster number 3?

Additionally, it is advised to replace the term "high expression," currently utilized to delineate differential expression, with the more precise descriptors of "downregulation" or "up-regulation." This adjustment will contribute to the precision and clarity of the terminology employed.

Response: Thank you for the suggestion. We have revised accordingly.

Furthermore, a key clarification is warranted regarding the specific genotype to which the depicted graph pertains. Precisely identifying whether the graph corresponds to the WT control, Itm2b-KO, or Trem2-KO genotype is imperative for contextual understanding and accurate interpretation.

Response: We believe the genotypes are indicated where appropriate (Figure EV2B). However, we understand that it might be unclear which UMAPs correspond to each genotype. We have made efforts to enhance the clarity of this information in Figure EV2B.

In the text on page 6, the paragraph spanning from "We next" to "genotype-specific factors" articulates three concepts:

1. A significant majority of cells in cluster 3 exhibit Itm2b-KO (89%).

2. Itm2b registers as one of the most profoundly down-regulated genes within cluster
3. Among all four Itm2b-KO mice, there exists an ample presence of microglia assigned to cluster 3.

The composition of this paragraph raises concerns regarding potential "circular reasoning" wherein the ideas presented appear to reinforce one another in a circular manner.

Response: Thank you for the suggestion. The composition of this paragraph has been changed. Hopefully, these changes satisfy the reviewer's criticisms.

Concerning Figure 3A, an elucidation is warranted to account for the observed weak signal of TREM2. Given the current blotting data, this attenuated signal intensity might suggest a scenario wherein the extent of TREM2 engagement with BRI2 is relatively limited or minimal. Such an interpretation would contribute to a clearer understanding of the presented results and their potential implications.

Response: It is indeed accurate to note that not all TREM2 molecules are co-immunoprecipitated by the anti-FLAG antibody M2, indicating that not all the TREM2 molecules expressed in transfected cells are bound to BRI2. It is crucial to acknowledge that co-immunoprecipitations are conducted in the presence of detergents, such as 0.5% NP40, as in our study, to disrupt cellular membranes. Detergents can, in many cases, adversely affect protein-protein interactions, especially when dealing with transmembrane proteins. Moreover, variations between independent co-transfections (biological replicates) are not uncommon in complex experiments like this one. Factors such as differences in plasmid expression levels can contribute to this variability. Interestingly, in Figures 3C, D, and F, the co-precipitation appears more robust, but we are cautious about drawing strong conclusions regarding the interaction's strength based solely on overexpression experiments.

Turning attention to Figure 3C, it is noteworthy that the significance or implications of the asterisk symbol (*) remain unexplained within both the main text and the figure's legend. Additionally, the meaning behind the designation "(f.l)" lacks clarification.

Response: Thank you for noticing these omissions. They have been corrected.

In relation to Figure 3D, a concise clarification regarding the identity of the bands falling within the 100-150 kDa range under the NT2 condition would greatly enhance the figure's clarity and contextual understanding.

Response: Regarding the NT2 antibody, it is a Rabbit IgG monoclonal. The NT2 IPs were

analyzed by Western blotting using either M2 (upper panel), a mouse monoclonal antibody, which is detected by the secondary antibody Goat Anti-Mouse IgG(H+L), Human ads-HRP from Southern Biotech (catalog #1031-05), or Trem2-CT (lower panel), a Rabbit IgG monoclonal antibody, which is revealed using Mouse Anti-Rabbit IgG (Conformation Specific) (L27A9) mAb (HRP Conjugate) from Cell Signaling Technology (catalog #5127). In both cases, even though the primary and secondary antibodies come from different species, they both detect the same band >100 kDa. The size of this band does not correspond to reduced heavy or light chains (which could be detected by the secondary antibodies). It is conceivable that the band size aligns with non-reduced heavy chains that may cross-react with both secondary antibodies. However, we must acknowledge that we are unable to definitively confirm this interpretation (which is, anyway, farfetched) and we did not conduct tests with the secondary antibodies alone to investigate this further. However, it is important to emphasize that even though this unexpected.

Directing attention to Figure 4A, it is pertinent to note that the significance or interpretation associated with the asterisk symbol (*) remains undisclosed within both the main text and the figure's legend.

Response: Thank you for noticing this omission. It has been corrected.

Regarding Figure 5C, it is evident that the signal intensity of the Trem2 blot is notably weak. This observation should be acknowledged and addressed, possibly by providing an explanation for the diminished signal strength.

Response: Yes, the reviewer is correct. The Trem2-CTF signal is not very strong due to elevated background in this blot. Nevertheless, it is clear from the data that *Trem2-KO* cells lack, as predictable, a Trem2-CTF signal.

With regards to Figure 5D, while a discernible distinction is observable within the female group, the same differentiation is less apparent within the male group. This observation merits acknowledgement and consideration, particularly in terms of potential underlying factors that may account for the disparity in results between the male and female cohorts.

Response: Thank you for the suggestion. We have included a sentence in the results to acknowledge this difference.

Concerning Figure 5E, the observation of substantially reduced CNS A β 40 and A β 42 levels in

Trem2-KO mice indeed presents an intriguing contrast. This phenomenon appears counterintuitive given the established understanding that loss of TREM2 function is associated with compromised microglial uptake of A β and lipoproteins. The text acknowledges this potential discrepancy by noting, "This finding may appear contradictory to the notion that TREM2 mediates A β clearance." However, a more comprehensive analysis or discussion of this apparent contradiction, along with potential explanations or alternative mechanisms, would add depth to the interpretation of the results and enrich the scientific discourse.

Response: We have extended the discussion of this apparent contradiction in the results section: "*Trem2-KO* mice, on the other hand, had significantly lower CNS A β 40 and A β 42 levels (Fig 7E). This finding may appear contradictory to the notion that TREM2 mediates A β clearance (Lessard *et al*, 2018; Yeh *et al*, 2016). However, previous studies primarily investigated human A β , utilizing either transgenic models expressing human APP or *in vitro* oligomeric forms of human A β 42. It is noteworthy that the 3-amino acid differences between rodent and human A β greatly influence the propensity of A β to form oligomers. Human A β species are known to have a heightened tendency to aggregate compared to their rodent counterparts. If Trem2 primarily facilitates the clearance of oligomeric or aggregated A β species, while Trem2 deletion enhances the efficiency of clearing soluble A β , *Trem2 KO* mice might exhibit reduced A β levels, particularly if the majority of mouse A β forms are monomeric. Moreover, Trem2 deletion could conceivably lower A β levels by influencing A β generation, possibly through a trans-cellular mechanism. Although we currently lack data to definitively reconcile this apparent contradiction, these factors underscore the intricate role played by TREM2 in A β regulation."

The discussion presents compelling evidence for the interaction between BRI2 and TREM2, especially with respect to their involvement in microglial function. Consideration of potential downstream effects and regulatory mechanisms resulting from this interaction could enhance the depth of analysis. The discussion adequately delves into the expansion of this cluster in *Itm2b-KO* mice, but further interpretation of the functional significance of this expansion, particularly in the context of AD pathology, would be valuable. The potential roles of Trem2 processing and its modulation by BRI2 in shaping microglial phenotypes could be discussed more extensively. In consonance, while the paper discusses the implications of altered Trem2 processing in the context of FDD and AD, it would benefit from an exploration of potential downstream consequences and how these alterations may contribute to disease progression. Connecting the findings to Figure 7 A,B & C: BRI2 regulation of TREM2 processing in microglia:

implications for Alzheimer's and related dementias (Tao Yin* & Luciano D'Adamio*) 37 microglial dysfunction and neuroinflammation, which are central to AD pathogenesis, would strengthen the discussion's disease-related implications.

Response: Thank you for your suggestion. We have exercised caution in extending the discussion for two primary reasons. Firstly, speculative discussions can sometimes be misconstrued as an attempt to overstate the significance of our findings. Secondly, the relationship between neuroinflammation, pathogenic TREM2 variants, and dementia pathogenesis is quite complex and not fully understood. To simplify this complexity, one could hypothesize that conditions activating pro-inflammatory microglial functions, including phagocytosis and cytokine production, might serve as triggering pathogenic factors. Conversely, in the context of the Amyloid hypothesis, where extracellular A β is considered a primary pathogenic factor, the loss of microglial functionality like phagocytosis could be viewed as a contributing factor to AD. Microglial activation could also be seen as a compensatory response to a primary pathogenic factor, such as increased Amyloid, as the organism attempts to contain it by enhancing Amyloid uptake and phagocytosis, for instance. It is also plausible that both scenarios coexist, with microglial activation sometimes triggering dysfunctional changes that contribute to neurodegeneration and dementia, while in other cases, reduced microglial functionality results in an impaired clearance of amyloids, subsequently leading to neurodegenerative and dementia-related processes. This dual possibility underscores the intricate and multifaceted nature of the relationship between microglial activities and the pathogenesis of neurodegenerative diseases. The intricate nature of this relationship complicates our ability to clearly elucidate how our findings mechanistically connect to the process of neurodegeneration. In any case, we have extended and modified the discussion, and we hope this revised discussion aligns more closely with your expectations.

Referee #3:

1. A) Co-IP experiments were performed only with transient overexpression cell models (Fig. 3). Demonstration of co-IP from cells that endogenously express BRI2 and Trem2 (i.e. primary microglia) could support the conclusion on BRI2-Trem2 interaction.

Response: We had acquired data related to endogenous interactions, which were not included in the initial submission due to suboptimal quality and because the data were obtained using primary macrophages. This data is now incorporated into Figure EV3C in this revised version. It is important to note that generating effective antibodies against BRI2, both human and rodent, has historically been a challenging endeavor. The Western blot experiments featuring endogenous Bri2 in this paper, including the co-immunoprecipitation reported in this revised version in Figure EV3C, utilized a limited quantity of an anti-Bri2 rabbit monoclonal antibody produced by Dr. Richard W. Cho at Cell Signaling Technology. Unfortunately, despite being one of the most promising anti-Bri2 antibodies we have worked with, its quality was deemed insufficient by Dr. Richard W. Cho for commercialization by Cell Signaling Technology. Essentially, at present, we lack an antibody capable of detecting endogenous rodent Bri2. We have evaluated all commercially available antibodies, but none of them recognize a band of the expected size expressed in WT cells and absent in *Itm2b-KO* cells. Regrettably, none of the commercially available antibodies have undergone testing with the appropriate negative controls to establish specificity, such as cells lacking BRI2 expression.

B) Authors should explain the bands migrating above 100 kDa in the NT2 IP samples (fig. 3D).

Response: See response to reviewer #2.

C) In this panel (upper blot), an F-BRI2-NTF is indicated, but hardly visible on the blot, and apparently does not co-IP with Trem2. Authors should comment on the length of this fragment, and whether it would contain the putative APP interacting domain.

Response: These BRI2-NTF bands are thought to represent the membrane-bound products of BRI2 processing by ADAM10. The exact site of processing remains unknown to our knowledge, so we cannot precisely determine the amino acid composition of this BRI2 NTF. However, based on the comparison with BRI2-deletion mutants in Figure 3F and the new Figures 4 and EV3, we can approximate that this fragment may encompass amino acids 1-105 of BRI2. In light of our claim that both BRI2-1-105 and BRI2-1-93 interact with Trem2, one might expect to find this fragment in the Trem2 CT and NT1 immunoprecipitations shown in Figure 3D. However, it is

notable that this fragment is not detected in those immunoprecipitations. Additionally, it is worth mentioning that in this experiment, the presence of F-BRI2-NTF is not visible in lysates from cells co-transfected with F-BRI2 and Trem2. In any case, we detect this BRI2-NTF in the BRI2-IPs in Figure EV3A where human BRI2 and human TREM2 have been tested in interaction experiments.

2. The data in Fig. 3F indicate that the C-terminal half of BRI2 that contains the Brichos domain is mainly responsible for the interaction with Trem2. Thus, it should be tested whether the soluble ectodomain of BRI2 that is generated by alpha-secretase also interacts with Trem2 and affects the proteolytic processing.

Response: As mentioned earlier, the precise site of processing remains unknown to our knowledge. Nevertheless, we have investigated the interaction between two distinct BRI2 ectodomain recombinant proteins (new Figure 5). Furthermore, in Figure EV4, we present a pilot experiment (conducted once using three biological replicates obtained from six *Itm2b-KO* P2 pups) that provides preliminary support for the hypothesis that the BRI2 ectodomain can influence Trem2 shedding.

3. It is stated that the deletion constructs 1-131, 1-117, 1-105 bind with similar efficiency. However, the expression level of the different BRI2 constructs seem to vary (on the provided blot, Fig. 3F). Quantitative analysis could help to support the conclusion of the authors.

Response: We concur with the reviewer that our earlier statement regarding the equal efficiency of the BRI2 mutants was an overstatement, given the variability in the expression of the BRI2 mutants. We have rephrased this statement in the results section.

4. A) The Trem2 CTF co-precipitates with BRI2 (Fig.3C), raising the question which domains of Trem2 are involved in the interaction. Using deletion constructs of Trem2 could help to map interacting domains of Trem2 (as done for BRI2).

Response: Yes, this is correct and we should have commented on this finding. We have added a comment about this and have performed experiments with TREM2-deletion/mutant constructs (Figures 4C and D and EV3B).

B) Do disease-associated variants of Trem2 affect the interaction with BRI2?

Response: We concur with the reviewer that investigating whether any of the TREM2 variants linked to a higher risk of AD alter the strength of the interaction with BRI2 is an important

avenue for future research. It is plausible that some variants may impact the interaction while others do not. In this manuscript, we have tested interaction of BRI2 with the TREM2-W198Ter variant. This variant is linked to Frontotemporal Dementia. The data are shown in Figures 4C and D and EV3B. Nevertheless, we concur with the reviewer that assessing whether pathogenic TREM2 variants influence the interaction with BRI2 constitutes an essential follow-up to the study presented here.

C) As co-IP approaches from cell lysates do not rule out the involvement of additional proteins/factors, a direct interaction of the ectodomains of BRI2 and Trem2 could be tested in cell free systems with recombinant proteins.

Response: This is an excellent point. We have conducted the suggested experiment, and the results indeed indicate a direct interaction between the ectodomains of BRI2 and TREM2. However, further experiments will be necessary to precisely identify the specific region(s) of TREM2 and BRI2 involved in this interaction. The data are shown in the new Figure 5.

5. A) BRI2 also represents a protein substrate of alpha-secretase, and thus, it is possible that the overexpression of BRI2 (Fig. 4) could affect Trem2 processing by substrate competition. Overexpression of the Brichos domain could help to clarify whether the observed effects could result from direct interaction with Trem2 or substrate competition or both.

Response: The reasons why we do not think that BRI2 overexpression reduces Trem2 processing in transfected HEK293 cells via substrate competition are discussed in this revised manuscript in the results section: “Since BRI2 is an α -secretase substrate, BRI2 overexpression might influence Trem2 processing through substrate competition. Additionally, overexpression of BRI2 could impact the overall activity of α -secretases. However, prior evidence suggests that this is unlikely. BRI2 binds to APP but not to APP-Like Protein 1 (APLP1) and APP-Like Protein 2 (APLP2). Both APLP1 and APLP2 are also substrates for α -secretase, like APP (Scheinfeld *et al.*, 2002a). Overexpression of F-BRI2 had no discernible impact on the processing of APLP1 and APLP2 (Matsuda *et al.*, 2008). Nevertheless, when chimeric molecules of APLP1 and APLP2 were engineered, with the BRI2-binding domain of APP replacing the corresponding domains in APLP1 and APLP2, these chimeric proteins bound to BRI2 and were susceptible to processing inhibition by BRI2 (Matsuda *et al.*, 2008). These results suggest that the inhibition of α -secretase processing by BRI2 is primarily driven by BRI2-substrate”.

6. A) It should also be tested if BRI2 affects the subcellular transport and localization of Trem2.

Cytochemical detection of both proteins could be done to address the (co)localization of both proteins.

Response: We acknowledge the reviewer's suggestion regarding experiments to test the colocalization of endogenous Bri2 and Trem2 in primary microglia, which could provide valuable supporting evidence. Regretfully, conducting such experiments is currently not feasible for us due to certain limitations. Specifically, the availability of anti-mouse Bri2 antibodies is currently restricted, as mentioned above. Additionally, despite our efforts, the anti-Trem2 antibodies used in our study did not yield specific signals in immunofluorescence (IF) experiments in primary microglia. Our initial pilot experiments involved comparing IF signals in WT and *Trem2-KO* primary microglia under various conditions and with different antibodies. Unfortunately, in all tested conditions, the obtained IF signals were similar in both cell types, suggesting non-specificity.

B) Does BRI2 interact with immature and mature Trem2?

Response: This is indeed an important question for which we do not have a definitive answer. What we have observed from the data in Figure 3A, 3C, and 3F is that F-BRI2 appears to exhibit a preference for binding highly glycosylated Trem2, which implies that BRI2 may preferentially bind mature Trem2. Furthermore, the results presented in the new Figures 4C and D and EV3B provide additional support for this observation. We discuss the significance of these findings in the Results section.

7. A) The bands labeled as sTREM2 should be explained in more detail (Fig. 4A and C). Migration is indicated as ~15 kDa. This does not fit very well to the mw of mature full length Trem2 (about 30-50 kDa).

Response: Trem2 is expected to migrate at approximately 23.5 kDa without glycosylation, while non-glycosylated sTrem2 should migrate at around 15.5 kDa.

B) Is immature trem2 secreted under these conditions?

Response: The lack of bands detected by the Anti-C-terminal Trem2 antibody in the cell culture supernatants of transfected cells (new panel in Figure 6E) and primary mouse microglia (new Figure 10A) suggests that immature Trem2 is not secreted.

8. In transient expression systems (fig. 3 and 4) overexpression of BRI2 stabilizes full-length Trem2. However, levels of Trem2 were not altered by the deletion of *Item2b* in vivo (Fig. 5B),

and authors speculate about a potential compensatory mechanisms. What could be an explanation?

Response: We have discussed potential reasons for this apparent contradiction: “The study conducted in heterologous HEK293 cells effectively demonstrates that BRI2 diminishes α -secretase processing of TREM2 by providing clear evidence of increased levels of the TREM2 substrate, along with a simultaneous reduction in the cleavage products sTREM2 and TREM2-CTF. The correlation between the binding of BRI2 to TREM2 and the inhibition of TREM2 processing solidifies the notion that this observed effect is primarily a consequence of their direct interaction. One advantage of employing heterologous cells is the ability to isolate a specific biological effect, in this case, the impact of BRI2 binding to TREM2 on α -processing of TREM2. This isolation is essential because it enables to distinguish this effect from other potential influences stemming from various cellular components or indirect mechanisms that might be at play in cells where these pathways are naturally active. Indeed, we provide several lines of evidence suggesting other functional interactions between Bri2 and Trem2 in microglia. Firstly, we observed a decrease in *Trem2* mRNA expression in *Itm2b-KO* primary microglia and CNS-derived microglia, indicating a role of Bri2 in regulating *Trem2* gene expression. Furthermore, despite this reduction in *Trem2* mRNA and an increase in Trem2 processing, the overall levels of Trem2 full-length (f.l.) protein do not appear to be diminished in these models. This seeming contradiction could potentially be explained by compensatory mechanisms. For example, Bri2 might facilitate the clearance of Trem2 through alternative pathways, such as lysosomes and autophagosomes, or it could influence the synthesis and maturation of Trem2 protein. While these additional mechanisms are not explored in this study, they underscore the intricate interconnection between Trem2 and Bri2 in microglial biology”.

9. A) Levels of Trem2 obtained by ELISA1 are given as A.U., that obtained by ELISA2 in pg. Authors should also provide data from ELISA1 in pg.

Response: Due to the unavailability of recombinant full-length Trem2, the standards utilized for these measurements comprise cell lysates from HEK293 cells transfected with rat Trem2. As a negative control, lysates from mock-transfected HEK293 cells were used. Consequently, the results of ELISA1 cannot be expressed in pg.

Can authors detect full-length Trem2 and sTrem2 by western blotting (as done for the Trem2 CTF, fig. 5c)? To facilitate detection, samples could be deglycosylated (as done in fig. 3). It would be possible to determine the ratios of full-length Trem2 to sTREM2 and CTF in individual

samples and further assess precursor-product relationships from in vivo samples. Such an approach could also be used to analyze the metabolism of Trem2 in primary microglia from mouse models generated in this study (microglia-specific deletion of *Itm2b*).

Response: We conducted a comparison between ELISA and Western blot quantification of sTrem2 in conditioned media from WT and *Itm2b-KO* primary microglia (Figure 10A), revealing a strong concordance between these two measurement methods. Furthermore, we provide a rationale in the same paragraph explaining our preference for ELISA in subsequent sTrem2 quantification experiments. These experiments also involved the utilization of deglycosylation to quantify Trem2 and Trem2-CTF levels in WT and *Itm2b-KO* primary microglia (Figure 10D and 10H).

10. It is mentioned that the FDD mouse model exhibits neuroinflammation. As Trem2 expression could be upregulated during neuroinflammation, authors should detect mRNA levels of Trem2 and also detect full-length Trem2 in order to assess precursor-product ratios (see above).

Response: FDD patients do show neuroinflammation, correct. We agree with the reviewer's assessment that the observed increase in Trem2-CTF and sTrem2 levels in aged FDD-KI mice may be influenced by various factors. Unfortunately, we do not have access to additional material from the previously analyzed animals and do not possess 8-month-old mice to replicate the experiment. Consequently, we discuss the FDD-KI mouse results as follows: "The elevated levels of Trem2-CTF and sTrem2 in the CNS of FDD-KI mice provide evidence that pathogenic BRI2 mutations can impact the levels of TREM2 metabolites in the CNS. These changes may be attributed to neuroinflammatory effects resulting from this mutation, as seen in FDD patients. Neuroinflammation might indirectly affect Trem2-CTF and sTrem2 levels through various mechanisms, including alterations in microglia numbers and activation states, changes in the activity of secretases within microglia, fluctuations in Trem2 protein expression, and variations in the turnover rates of Trem2, Trem2-CTF, and sTrem2. Furthermore, this mutation might disrupt the functional interaction between BRI2 and TREM2, potentially due to the instability of the mutant BRI2 protein. Importantly, it should be noted that mature BRI2, which represents the functional form of BRI2, exhibits reduced levels in the CNS of both FDD-KI rodents (Tamayev *et al.*, 2010b; Yin *et al.*, 2021a; Yin *et al.*, 2021b) and FDD patients (Matsuda *et al.*, 2011b; Tamayev *et al.*, 2010a; Tamayev *et al.*, 2010b). These hypotheses are not mutually exclusive, and it is plausible that alterations in the BRI2-TREM2 functional interaction contribute to the neuroinflammation caused by the *ITM2B* Danish pathogenic mutation. Further studies are required to elucidate the

relative impact of these mechanisms on the observed neuroinflammation in FDD patients.” But we concur that the suggested experiments are of significant importance and should be conducted in future studies.

11. Does the deletion or overexpression of *Itm2b* affect expression and overall activity of alpha-secretases (ADAM10/17)?

Response: The effects of *BRI2* overexpression have been discussed answering a previous criticism. The experiments conducted to investigate the effects of *Itm2b* deletion question are presented in Figure 11. We did not observe an increase in the expression of *Adam10* and *Adam17* mRNA in *Itm2b-KO* primary microglia. In fact, there was a trend toward downregulation of both genes. Additionally, we detected comparable levels of Adam17 protein in primary microglia, both stimulated and unstimulated, from WT and *Itm2b-KO* mice (despite the potential mRNA downregulation). We were unable to detect Adam10 protein by Western blot.

As shown in Figure 10, *E. coli* stimulation increases Trem2 shedding and sTrem2 production in primary microglia. Notably, this increase was significantly higher in *Itm2b-KO* primary microglia compared to WT primary microglia cultures. In contrast, the levels of soluble TNF α , which is produced by the cleavage of mature TNF α by Adam10 and Adam17 upon microglial activation with *E. coli*, showed similar increases in both *Itm2b-KO* and WT microglia. While we cannot completely rule out the possibility that *Bri2* deletion might affect mature TNF α availability, the overall data presented in Figure 11 do not support the hypothesis that *Bri2* deletion increases the expression and overall activity of Adam10 and Adam17.

12. It would also be important to test whether Trem2-specific signaling (e.g. phosphorylation of Dap12 and/or Syk) and microglial function (e.g. phagocytosis, cytokine secretion) is affected by the deletion *Itm2b*. Primary microglia cultures could be used for these experiments.

Response: We have conducted these experiments as suggested. Regarding signaling, we did not observe an increase in Syk phosphorylation upon *E. coli* stimulation of primary microglia, both in control WT and *Itm2b-KO* primary microglia (Figure 12). Unfortunately, we encountered difficulties in finding a phosphor-Dap12 antibody that provided specific signals. However, we did identify reduced phosphorylation of p38 in *Itm2b-KO* primary microglia compared to control cells, implying potential impairment in the p38 pathway due to *Bri2* deletion. The functional implications of this deficit will require further exploration.

These signaling assays were conducted at a single time point, specifically 2 hours after *E. coli* stimulation. Several studies indicate that microglial activation by *E. coli* should result in

increased Syk phosphorylation at this time point. We are uncertain why we did not detect it; nevertheless, we cannot rule out the possibility that Bri2 deletion may alter the kinetics of Syk phosphorylation. To address this issue, additional experiments examining pSyk at earlier time points will be necessary.

We conducted phagocytosis experiments and observed that Bri2 is essential for effective *E. coli* phagocytosis (Figure 13). This finding is particularly intriguing in light of the fact that the pathogenic TREM2 p.H157Y mutation, which enhances the shedding of mutant TREM2, is also associated with a significant decrease in *E. coli* phagocytosis capacity.

As for cytokines secretion, we analyzed secretion of 10 cytokines and chemokines in primary microglia (Figure 14), and found evidence of a substantial impact of Bri2 deletion on both *E. coli*-induced and "basal" cytokine and chemokines secretion .

Additional points:

Authors should provide more information on the number of experiments and number of replicate samples for the individual data sets (e.g. Fig. 3-7)

Response: Thank you for noticing these omissions. They have been corrected.

Heading of the last paragraph of the results section "FDD-KI mice show elevated CNS levels of Trem2-CTF and sTrem2 in mice increases CNS sTrem2 levels." reads confusing.

Response: Thank you for noticing this mistake. The heading has been corrected.

Reference list contains duplications

Response: Thank you for noticing this mistake. Duplications have been eliminated.

Dear Luciano,

Thank you for your patience while your revised manuscript was re-reviewed. We have now received the enclosed reports from all referees. Referees 2 and 3 still have a few more minor suggestions that I would like you to incorporate before we can proceed with the official acceptance of your manuscript. Please co-submit a point by point response to all comments with your final ms.

Some editorial requests will also need to be addressed:

- Please reduce the number of keywords to 5.
- Please remove the author credits from the ms file. All credits need to be entered in our online ms submission system.
- The figure callout "1B" should be corrected to "Fig 1B"; please add a callout for Fig 10I.
- Please upload the Reagents table as a separate file and remove it from the ms file. Please use the table format that is available in our Author Guidelines online.
- The following Figures seem to be in landscape format: 1, 3, 4, 6, 7, 8, 10, 11, 12, 14 and EV5. The format needs to be correct to portrait and the figure should fit on one regular page.
- The abstract needs to be written in present tense with respect to the new findings. Please correct.
- Please address these comments from our data editors:

1. Please note that the figure 6a; 9c; 11c-d; does not contain any quantification graph, kindly rectify the statistics related information in the figure legend appropriately.
2. Please define the annotated p values **** in the legend of figure 1d; as appropriate.
3. Please indicate the statistical test used for data analysis in the legend of figure 2c.
4. Please note that in figures 6b, d; 7b, d-e; 11a-b, e-f; 12b, d; there is a mismatch between the annotated p values in the figure legend and the annotated p values in the figure file that should be corrected.
5. Please note that information related to n is missing in the legends of figures 1a-b; 2c-d.
6. Please note that the error bar is not defined in the legend of figure 2d.
7. Please note that the asterisk "*" is not defined in the legend of figure 6a; 7c; 12a. This needs to be rectified.
8. Please note that the asterisks "**"/**** are not defined in the legend of figure EV 3b. This needs to be rectified.

EMBO press papers are accompanied online by A) a short (1-2 sentences) summary of the findings and their significance, B) 2-3 bullet points highlighting key results and C) a synopsis image that is exactly 550 pixels wide and 200-600 pixels high (the height is variable). You can either show a model or key data in the synopsis image. Please note that text needs to be readable at the final size. Please send us this information along with the final manuscript.

Referee #1:

The authors have addressed my concerns.

Referee #2:

The majority of the comments has been addressed and the papers has been substantially improved.

2 minor errors have been found.

2 additional suggestions as follows:

"Cluster 12 was identified as brain-associated macrophages (BAMs), distinguished by the up-regulation of Mrc1, Cd163, and Lyve1 (Fig EV2C). Similarly, cluster 14 exhibited strong resemblances to a recently characterized ribosome biogenesis cluster (Sun et al, 2023), marked by the up-regulation of ribosomal genes (Fig EV2C)."

Comment:

In the text, cluster 12 and 14 are mentioned. With this graph, unless previous clusters are excluded, it is impossible to visually appreciate what is up-regulated or down-regulated in clusters 12 and 14. An effective and straightforward way to solve this would be to change the columns width. Considering that "Doheatmap" function returns a ggplot object, can be easily done → <https://stackoverflow.com/questions/75007545/how-to-adjust-the-cluster-size-column-in-doheatmap-plot-in-r>

"Therefore, the observed expansion of cluster 7 is attributed to animal-specific factors rather than genotype-specific factors. Several of these 10 clusters exhibited distinct representations across different genotypes. Cluster 4 displayed overrepresentation in the *Itm2b*-KO samples (and, to a somewhat lesser extent, in *Itm2b/Trem2*-dKO samples) and has thus been designated as the *Itm2b*-dependent (*Itm2b*-D) cluster."

Comment:

It would be helpful for the reader to state in the text that differences in representation can be observed in "Table 1".

Minor:

In figure EV2 and legend, D and E letters designating the figures are swapped. The description in the figure legends therefore is not correct.

"Thus, we transfected HEK293 and N2a cells with constructs coding for BRI2 FLAGtagged at the NH2-terminal cytoplasmic tail (F-BRI2) and rat Trem2 (Trem2-Mi isoform, UniProtKB - A0A6G8MV71)(Tambini & D'Adamio, 2020b), and then immunoprecipitated the lysates with an anti-FLAG antibody to pull down BRI2."

In the previous version, this reference was included.

Referee #3:

In the revised version of the manuscript, authors provide substantial additional data that further support for the relevance of BRI2 in microglia and the interaction with TREM2. The manuscript improved by the addition of relevant controls, and appropriate discussion of the provided data. Overall, the study provides novel insight into the role of BRI in microglia. However, the following minor points should be considered:

1. the indicated intraluminal/extracellular domains of the proteins TREM1 and BRI2 (blue boxes) should be placed more accurately (Fig. 3B, E). These domains span to the transmembrane domains.
2. The sentence „This confirms that BRI2 and Trem2 interact with each other, suggesting a functional interaction between the two proteins" in the results section reads bit awkward, and should be rephrased.
3. The new statement „ There is a notable trend suggesting that F-BRI2 may preferentially interact with glycosylated Trem2, ..." is somewhat misleading. Trem2 undergoes co-translational N-glycosylation. When expressed in cells, this protein should (always) be glycosylated, but can be present in immature and mature form. The difference in migration results from further glycosylation during transport/maturation of the protein. The sentence should be rephrased (e.g. There is a notable trend for the interaction of F-BRI2 with mature Trem2). This should also be considered when describing the results obtained with the W198Ter TREM2 variants.
4. Regarding the previous point on the nature of the band labeled as sTREM2 in Fig. 6 (previous Fig. 4), authors write in their response letter that „Trem2 is expected to migrate at approximately 23.5 kDa without glycosylation, while non-glycosylated sTrem2 should migrate at 15.5 kDa." As mentioned above, it is unlikely that cells produce non-glycosylated Trem2 or sTrem2. Thus, the band migrating at about 15 kDa (fig. 6a, c) likely does not represent non-glycosylated sTrem2. Deglycosylation experiments (as done for the purified TREM2-ECD and sTREM2 proteins from CHO-S cells, fig. 5) could address this question. Is it possible that the transiently transfected cells in their experimental system rather secrete immature (partially glycosylated) sTrem2? A similar finding for transiently overexpressed Trem2 has been described recently (Ibach et al. 2021). Actually, the recombinant sTrem2/ECD proteins used for interaction studies (fig. 5) have very different migration as a smear between 25-50 kDa. At least, the possibility that the sTrem2 band at about 15 kDa (fig. 6) represents immature Trem2 should be discussed.

5. Authors should not write „mutant experiments“, rather experiments with mutant proteins.
6. The bicistronic constructs should be described in more detail or a reference should be provided. Does it contain an IRES?

The manuscript would strongly benefit from careful proofreading.

Dear Editor,

Thank you for your message.

Our responses to all requests/comments are in **BOLD**.

Please reduce the number of keywords to 5.

Done.

- Please remove the author credits from the ms file. All credits need to be entered in our online ms submission system.

Done.

- The figure callout "1B" should be corrected to "Fig 1B"; please add a callout for Fig 10I.

Done.

- Please upload the Reagents table as a separate file and remove it from the ms file. Please use the table format that is available in our Author Guidelines online.

Done.

- The following Figures seem to be in landscape format: 1, 3, 4, 6, 7, 8, 10, 11, 12, 14 and EV5. The format needs to be correct to portrait and the figure should fit on one regular page.

After email exchange, it was decided that the Figures' sizes are acceptable.

- The abstract needs to be written in present tense with respect to the new findings. Please correct.

Done.

- Please address these comments from our data editors:

1. Please note that the figure 6a; 9c; 11c-d; does not contain any quantification graph, kindly rectify the statistics related information in the figure legend appropriately.

Done. However, Fig 11 c and d contain a quantification graph (Adam17).

2. Please define the annotated p values **** in the legend of figure 1d; as appropriate.

Done.

3. Please indicate the statistical test used for data analysis in the legend of figure 2c.

Done.

4. Please note that in figures 6b, d; 7b, d-e; 11a-b, e-f; 12b, d; there is a mismatch between the annotated p values in the figure legend and the annotated p values in the figure file that should be corrected.

Done.

5. Please note that information related to n is missing in the legends of figures 1a-b; 2c-d. The data set in Fig 1a

Done. However, note that these figures use publicly available data and the detailed information about the n are also found in the cited source papers.

6. Please note that the error bar is not defined in the legend of figure 2d.

Done.

7. Please note that the asterisk "*" is not defined in the legend of figure 6a; 7c; 12a. This needs to be rectified.

Done.

8. Please note that the asterisks "**/****" are not defined in the legend of figure EV 3b. This needs to be rectified.

The asterisks have been removed. Should not have been there.

Referee #1:

The authors have addressed my concerns.

Referee #2:

"Cluster 12 was identified as brain-associated macrophages (BAMs), distinguished by the up-regulation of Mrc1, Cd163, and Lyve1 (Fig EV2C). Similarly, cluster 14 exhibited strong resemblances to a recently characterized ribosome biogenesis cluster (Sun et al, 2023), marked by the up-regulation of ribosomal genes (Fig EV2C)."

Comment:

In the text, cluster 12 and 14 are mentioned. With this graph, unless previous clusters are excluded, it is impossible to visually appreciate what is up-regulated or down-regulated in clusters 12 and 14. An effective and straightforward way to solve this would be to change the column width. Considering that "Doheatmap" function returns a ggplot object, can be easily done →

<https://stackoverflow.com/questions/75007545/how-to-adjust-the-cluster-sizecolumn-in-doheatmap-plot-in-r>

We have added a panel with the column width of clusters 8-15 is enlarged (Fig EV2E).

"Therefore, the observed expansion of cluster 7 is attributed to animal-specific factors rather than genotype-specific factors. Several of these 10 clusters exhibited distinct representations across different genotypes. Cluster 4 displayed overrepresentation in the Itm2b-KO samples (and, to a somewhat lesser extent, in Itm2b/Trem2-dKO samples) and has thus been designated as the Itm2b-dependent (Itm2b-D) cluster."

Comment:

It would be helpful for the reader to state in the text that differences in representation can be observed in "Table 1".

Reference to Table 1 has been added according to the suggestion.

Minor:

In figure EV2 and legend, D and E letters designating the figures are swapped. The description in the figure legends therefore is not correct.

The inversion has been corrected.

"Thus, we transfected HEK293 and N2a cells with constructs coding for BRI2 FLAGtagged at the NH2-terminal cytoplasmic tail (F-BRI2) and rat Trem2 (Trem2-Mi isoform, UniProtKB - A0A6G8MV71)(Tambini & D'Adamio, 2020b), and then immunoprecipitated the lysates with an anti-FLAG antibody to pull down BRI2."

In the previous version, this reference was included.

The reference is redundant since we indicate the UniProtKB number.

Referee #3:

1. the indicated intraluminal/extracellular domains of the proteins TREM1 and BRI2 (blue boxes) should be placed more accurately (Fig. 3B, E). These domains span to the transmembrane domains.

This has been corrected.

2. The sentence „This confirms that BRI2 and Trem2 interact with each other, suggesting a functional interaction between the two proteins" in the results section reads bit awkward, and should be rephrased.

This confusing sentence has been eliminated.

3. The new statement „ There is a notable trend suggesting that F-BRI2 may preferentially interact with glycosylated Trem2, ..." is somewhat misleading. Trem2 undergoes co-translational N-glycosylation. When expressed in cells, this protein should (always) be glycosylated, but can be present in immature and mature form. The difference in migration results from further glycosylation during transport/maturation of the protein. The sentence should be rephrased (e.g. There is a notable trend for the interaction of F-BRI2 with mature Trem2). This should also be considered when describing the results obtained with the W198Ter TREM2 variants.

The sentence has been changed according to the suggestion.

4. Regarding the previous point on the nature of the band labeled as sTREM2 in Fig. 6 (previous Fig. 4), authors write in their response letter that „Trem2 is expected to migrate at approximately 23.5 kDa without glycosylation, while non-glycosylated sTrem2 should migrate at 15.5 kDa." As mentioned above, it is unlikely that cells produce non-glycosylated Trem2 or sTrem2. Thus, the band migrating at about 15 kDa (fig. 6a, c) likely does not represent non-glycosylated sTrem2. Deglycosylation experiments (as done for the purified TREM2-ECD and sTREM2 proteins from CHO-S cells, fig. 5) could address this question. Is it possible that the transiently transfected cells in their experimental system rather secrete immature (partially glycosylated) sTrem2? A similar finding for transiently overexpressed Trem2 has been described recently (Ibach et al. 2021). Actually, the recombinant sTrem2/ECD proteins used for interaction studies (fig. 5) have very different migration as a smear between 25-50 kDa. At least, the possibility that the sTrem2 band at about 15 kDa (fig. 6) represents immature Trem2 should be discussed.

We employ the Protein Deglycosylation Mix II (New England Biolabs, P6044s), which removes all N-linked and several common O-linked glycans from glycoproteins. In Figure 6A and 6C, the band that migrates at approximately 16 kDa (we assume the reviewer is referring to the lower blots of conditioned media) and is labeled as sTrem2 corresponds to deglycosylated Trem2 polypeptides. This is because prior to loading the samples onto PAGE and conducting western blot analysis with the Trem2 NT1 antibody, the conditioned media underwent deglycosylation treatment. The information regarding the deglycosylation of these samples is clearly indicated on the figure itself, positioned to

the left of the MWM labeling, as well as in the Figure legend. Indeed, the deglycosylated recombinant sTREM2 (Figure 5E, lower panel), exhibits an apparent MW that closely resembles, or possibly slightly exceeds, the MW of the sTrem2 bands seen in Figure 6A and 6C. This minor variation in apparent MW could be attributed to the presence of six additional His residues at the COOH-terminus of the recombinant sTREM2 protein, as well as potential species-specific differences.

5. Authors should not write „mutant experiments“, rather experiments with mutant proteins.
The sentence has been changed according to the suggestion.

6. The bicistronic constructs should be described in more detail or a reference should be provided. Does it contain an IRES?

Yes, they contain an IRES. We have included this information in the paper. It is important to note that readers who are interested in the details of these constructs can refer to the Reagents and Tools Table, which provides information to access the complete features, including the full DNA sequence of the constructs, amino acid sequences of the expressed proteins, schematics encompassing all components of the constructs, and more.

The manuscript would strongly benefit from careful proofreading.

Dr. Luciano D'Adamio
New Jersey Medical School, Rutgers University
205 South Orange Avenue
Cancer Building, Room H1214
Newark, USA, NJ 07103 07103
United States

Dear Luciano,

I am very pleased to accept your manuscript for publication in the next available issue of EMBO reports. Thank you for your contribution to our journal.

I slightly modified your short summary, please let me know whether you agree with this:

Transcriptomic and biochemical analyses of microglia uncover that Bri2 inhibits Trem2 cleavage by α -secretase. Bri2 deletion increases Trem2 cleavage, reduces E. coli phagocytosis, and alters patterns of cytokine/chemokine secretion.

- Single-cell transcriptomics of microglia unveils an epistatic interaction between Trem2 and Bri2
- BRI2 binds TREM2 and inhibits basal and E. coli-induced TREM2 cleavage by α -secretase.
- Bri2 deletion diminishes E. coli phagocytosis and alters microglial cytokine/chemokine secretion patterns.
